  SciPost Phys. Lect.Notes 25 (2021)

# A beginner's guide to non-abelian iPEPS for correlated fermions

Benedikt Bruognolo[1], Jheng-Wei Li[1], Jan von Delft[1] and Andreas Weichselbaum[1,2]*

**1** Arnold Sommerfeld Center for Theoretical Physics, Center for NanoScience,
and Munich Center for Quantum Science and Technology,
Ludwig-Maximilians-Universität München, 80333 Munich, Germany
**2** Department of Condensed Matter Physics and Materials Science,
Brookhaven National Laboratory, Upton, NY 11973-5000, USA

* weichselbaum@bnl.gov

## Abstract

Infinite projected entangled pair states (iPEPS) have emerged as a powerful tool for studying interacting two-dimensional fermionic systems. In this review, we discuss the iPEPS construction and some basic properties of this tensor network (TN) ansatz. Special focus is put on (i) a gentle introduction of the diagrammatic TN representations forming the basis for deriving the complex numerical algorithm, and (ii) the technical advance of fully exploiting non-abelian symmetries for fermionic iPEPS treatments of multi-band lattice models. The exploitation of non-abelian symmetries substantially increases the performance of the algorithm, enabling the treatment of fermionic systems up to a bond dimension $D = 24$ on a square lattice. A variety of complex two-dimensional (2D) models thus become numerically accessible. Here, we present first promising results for two types of multi-band Hubbard models, one with 2 bands of spinful fermions of $SU(2)_{\text{spin}} \otimes SU(2)_{\text{orb}}$ symmetry, the other with 3 flavors of spinless fermions of $SU(3)_{\text{flavor}}$ symmetry.

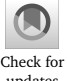

# 1 Introduction

Ever since the discovery of high-$T_c$ superconductivity, there is a great need for developing and improving numerical approaches for studying one-band and multi-band fermionic many-body systems in two spatial dimensions. Quantum Monte-Carlo (QMC) is an excellent candidate for this challenge [1]. However, the presence of the fermionic sign problem in these systems at finite doping often restricts the applicability of QMC to special points in the phase diagram close to half filling.

Tensor network techniques represent a promising alternative to QMC to successfully deal with complex systems of itinerant fermions. In particular, the density matrix renormalization group (DMRG) applied to two-dimensional clusters has provided us with some remarkable insights. Examples include the discovery of the spin-liquid ground state of the Kagome Heisenberg model [2,3] or the first observation of stripe states in the hole-doped $t$-$J$ model [4]. More

recently, also the infinite projected entangled pair state (iPEPS) approach was successfully used for a detailed study of the *t-J* model [5,6], as well as for clarifying the spin-liquid nature of the spin-half Kagome Heisenberg model [7, 8]. In addition, a combined iPEPS and DMRG study, supported by other numerical methods, led to a consensus regarding the existence of stripe order in the hole-doped Hubbard model [9].

A PEPS can be constructed by considering a lattice system where the entanglement of each site to the rest of the system is encoded via virtual degrees of freedom (entangled pairs) associated with the lattice bonds connecting that site to its neighbors. Projecting all virtual degrees of freedom associated with a given site to the physical Hilbert space of that site generates a PEPS tensor for that site [10]. Such a tensor network representation can be considered as a generalization of Affleck, Kennedy, Lieb and Tasaki (AKLT) states or tensor product states, which date back to even earlier literature [11–14]. In short, many tensor network algorithms to simulate many-body states in 2D are based on the PEPS representation, including the tensor renormalization group (TRG) [15, 16], the second renormalization group (SRG) [17], the higher-order tensor renormalization group [18], tensor network renormalization [19, 20], DMRG-like ground-state optimization [21, 22] and promising extensions to excited states by means of tangent space methods [23].

Despite many interesting developments, PEPS has not yet reached its full potential in application to frustrated and fermionic 2D systems. This is mostly due to the technical complexity of the algorithm, especially when dealing with fermionic signs [24] and when implementing symmetries explicitly [25–33]. Nevertheless, PEPS has recently proven its competitiveness and, for instance, provided new insights for underdoped Hubbard model [9, 34, 35] and *t-J* models [5, 6, 36], for spin-$\frac{1}{2}$ [7, 8] and spin-1 Kagome-Heisenberg models [37], as well as for the Shastry-Sutherland model [38, 39]. At the same time, PEPS is still in its infancy and there is much room for technical progress boosting the performance of the method [40–42].

In this work, we consider the PEPS method applied to translationally invariant systems, the so-called iPEPS ansatz [43], and focus on an aspect where further technical progress is certainly possible – the exploitation of symmetries. If the Hamiltonian is invariant under some symmetry group, its energy eigenstates can be grouped into multiplets transforming as irreducible representations (irreps) under symmetry transformations. Correspondingly, a tensor network for such a system can be constructed from tensors whose legs (both physical and virtual) carry irrep labels. Keeping track of this multiplet structure can reduce computational costs tremendously, since tensors acquire block substructures. Moreover, for non-abelian symmetries the relevant bond dimension is reduced from $D$, the number of individual states per bond, to $D^*$, the number of multiplets per bond. Computational costs scaling as $D^\alpha$ can thus effectively be reduced by a factor of $(D/D^*)^\alpha$. Also, memory requirements, the primary bottleneck for iPEPS calculations, can be significantly reduced. However, the tensor block structure entails overhead in the code complexity and performance, which requires some special care, specifically so if many, individually small blocks arise. With $\alpha \gtrsim 12$ for iPEPS and $D/D^* \simeq 3$ for SU(2) symmetry or larger for SU($N > 2$), the potential benefits of exploiting symmetries are evidently enormous. In practice, however, keeping track of symmetry labels requires codes with an additional layer of complexity, in particular for symmetry groups having outer multiplicity $> 1$, such as SU($N > 2$). While the exploitation of abelian symmetries in PEPS codes is becoming fairly routine by now, the number of applications of non-abelian PEPS can still be counted on one hand [33, 37, 44], all involving SU(2) symmetry.

Believing that non-abelian PEPS nevertheless holds great promise, we devote this tutorial review to a detailed exposition of its key ingredients. We offer a pedagogical review of the most important aspects of the PEPS representation and the iPEPS algorithm, mainly following the work of Philippe Corboz and coworkers [5, 6, 24, 45–47]. In particular, we discuss how to perform contractions [Sec. 3.3], how to keep track of fermionic minus signs, and how to per-

form tensor optimization via imaginary-time evolution [Sec. 3.5], including the gauge fixing for iPEPS [47, 48]. Additionally, we go beyond the scope of Corboz' work by explaining how arbitrary non-abelian symmetries can explicitly incorporated in the fermionic iPEPS ansatz in a generic manner, based on the QSpace [30] tensor library. A pedagogical discussion of SU(2) iPEPS was recently given in Ref. [49], with benchmarking computations on spin systems reported in Ref. [50]. Our treatment of symmetries represents a fully alternative approach to theirs, which permits us to deal with non-trivial outer multiplicities (OM) on a general footing. While OM is not present for SU(2) for rank-3 tensors, it already also occurs for SU(2) for tensors of rank $r > 3$. For larger symmetries, such as SU($N > 2$), OM already occurs generically even at the elementary level of rank-3 tensors.

A first application of our non-abelian fermionic iPEPS code, published concurrently with this tutorial review, is a study of the 2D fermionic $t$-$J$ model [51] – by exploiting either U(1) or SU(2) symmetry to allow or forbid spontaneous spin symmetry breaking, we elucidate the interplay between antiferromagnetic order, stripe formation and pairing correlations. In the present work, we further illustrate the power of non-abelian iPEPS by presenting some exemplary results for two 2D fermionic Hubbard models of higher complexity: a model with two degenerate bands of spinful fermions, featuring SU(2)$_{\text{spin}} \otimes$ SU(2)$_{\text{orb}}$ symmetry, and a model with three degenerate bands of spinless fermions, featuring SU(3)$_{\text{flavor}}$ symmetry.

## 2   Tensor network diagrams and convention

As implied by their name, tensor network techniques typically involve a large number of tensors of various rank that are iteratively manipulated. These manipulation steps may vary in their complexity and, for example, include matrix multiplication, or decomposition techniques such as singular value or eigenvalue decompoitions. In order to simplify the lengthy mathematical expressions which describe these steps and typically involve large sums over multiple indices, we heavily rely on using a diagrammatic representation for tensor network states. Analogous to the role of Feynman diagrams in quantum field theories, these tensor network diagrams are pictorial representation of mathematical expressions and help a great deal grasping the essence a TN algorithm. Since we extensively employ this pictorial language in this review, we here give a brief summary of our conventions together with an explanation on how to understand these diagrams in the following.

Each TN diagram consists of one or multiple extended objects (squares, circles, ...), which are connected by lines. Objects and lines represent tensors and indices, respectively. In the following, we give a few simple examples. For instance, a matrix or rank-2 tensor $A$ has two indices $\alpha, \beta$,

$$A_{\alpha\beta} = \underset{\alpha}{\quad} \overset{}{\underset{}{A}} \underset{\beta}{\quad}. \tag{1}$$

The number of values that an index can take is called its dimension.

The next expression, illustrating a matrix multiplication

$$\sum_{\beta} A_{\alpha\beta} B_{\beta\gamma} = \underset{\alpha}{\quad} A \underset{\beta}{\quad} B \underset{\gamma}{\quad}, \tag{2}$$

involves the sum over the common index $\beta$ of $A$ and $B$. This contraction is indicated by a line connecting $A$ and $B$.

In addition to the simple expressions shown above, we often have to deal with diagrams containing multiple sums and open indices, such as

$$\sum_{\alpha,\gamma} A_{\alpha\delta} B_{\alpha\beta\gamma} C_{\gamma\epsilon} = \qquad\qquad . \tag{3}$$

It holds generally true, that the diagrammatic representation becomes more beneficial, the more complex the expression and the larger the number of tensors involved since the logic of reading and understanding these diagrams remains the same.

For more evolved topics, such as fermionic TN descriptions and symmetric TNs, the diagrams will contain extra features. We will introduce these features in detail at the appropriate parts of this review.

# 3 Infinite projected entangled pair states

Projected entangled pair states (PEPS) present the natural generalization of the well-known MPS ansatz to higher spatial dimensions [10]. Analogously to their 1D counterpart, a PEPS consists of a set of high-ranked tensors which are connected by virtual bonds along the physical directions of the corresponding lattice system. In addition, PEPS satisfy the area law of the entanglement entropy in two dimensions [52], thus being able to faithfully represent physical states in gapped lattice models.

In this section, we give a pragmatic introduction to the PEPS construction from the point of view of numerical practitioners. To this end, we only briefly elaborate the ansatz and its properties before discussing numerical details of contraction, optimization, fermionic systems, and the implementation of symmetries.

## 3.1 PEPS ansatz and properties

To give a practical example, we consider a generic many-body wavefunction $|\psi\rangle$ on a $3 \times 3$ cluster. In its most general form, the wavefunction can be expressed in terms of the rank-9 coefficient tensor $\Psi_{\sigma_1^1 \sigma_2^1 \dots \sigma_3^3}$ acting in the local Fock space $|\sigma_y^x\rangle$,

$$|\psi\rangle = \sum_{\sigma_1^1 \sigma_2^1 \dots \sigma_3^3} \Psi_{\sigma_1^1 \sigma_2^1 \dots \sigma_3^3} |\sigma_1^1\rangle |\sigma_2^1\rangle \dots |\sigma_3^3\rangle, \tag{4}$$

where the integer indices $x$ and $y$ enumerate sites in the horizontal and vertical direction. The local or physical index $\sigma_y^x \in 1, \dots, d$ labels states in the local Hilbert space at site $\boldsymbol{r} = (x, y)$. Obviously, this generic representation suffers from an exponential system-size scaling, which is reflected in the fact that the number of elements of $\Psi$ is equal to the total Hilbert space $d^N = d^9$. Here $N$ denotes the total number of sites and the local dimension $d$ describes the total number of quantum states per site. Typical values are $d = 2$ for a spin-$\frac{1}{2}$ system or spinless fermions, $d = 3$ for spin-1, and $d = 4$ for spinful fermions.

The key idea of the PEPS construction is to circumvent the exponential scaling in system size by decomposing $\Psi$ into a set of high-ranked tensors (in the following denoted $M$ tensors).

A PEPS representation for the wavefunction in Eq. (4) requires a total of nine $M$ tensors,

$$|\psi\rangle = \sum_{\substack{\sigma_1^1 \sigma_2^1 \dots \sigma_3^3 \\ \alpha_1 \alpha_2 \dots \alpha_6 \\ \gamma_1 \gamma_2 \dots \gamma_6}} M_{\alpha_1 \gamma_1}^{\sigma_1^1} M_{\alpha_2 \gamma_1 \gamma_2}^{\sigma_2^1} \dots M_{\alpha_6 \gamma_6}^{\sigma_3^3} |\sigma_1^1 \sigma_2^1 \dots \sigma_3^3\rangle. \tag{5}$$

Each tensor $M$ has a set of virtual indices, $\alpha_i$ for horizontal bonds and $\gamma_i$ for vertical bonds, connecting each $M$ to its counterparts on up to four neighboring sites, according to the lattice geometry. Following Sec. 2, the diagrammatic representation can be easily derived by introducing the diagram for a rank-5 "bulk" tensor

$$M_{\alpha\beta\gamma\rho}^{\sigma_y^x} = \qquad \tag{6}$$

The boundary tensors of a finite-size PEPS contain fewer legs. Since we focus on the translationally invariant formulation of PEPS in the following, we refrain from a detailed discussion of various boundary conditions and the corresponding tensors [48].

In general, the number of $M$ tensors in the PEPS representation is equal to the number of sites in the system, e.g., $N = L \times L$ tensors for a square lattice of $L \times L$ sites. Starting from Eq. (6), the diagrammatic representation of the full wavefunction $|\psi\rangle$ in Eqs. (4) and (5) follows immediately,

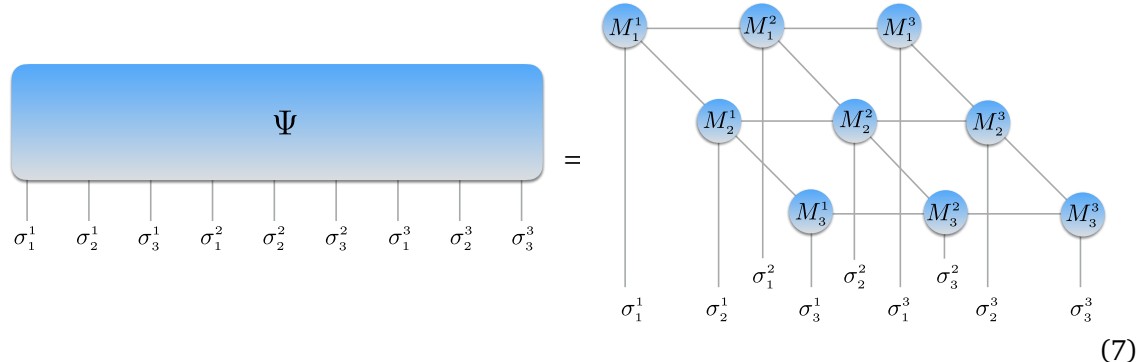

$$\tag{7}$$

In principle, one can perform such a decomposition exactly for any arbitrary many-body wavefunction. For larger systems, however, the dimension of the virtual indices has to be increased exponential which, for numerical purposes, is not practicable. Therefore, one limits the dimension of the virtual bonds of each PEPS tensor to some upper cutoff $D$ [53]. Thus adding an additional site (or row/column of sites) only leads to a polynomial increase in the number of coefficients of the wavefunction. In numerical practice, $D$ is used as a control parameter for the numerical accuracy. It is typically restricted to $D \leq 8\text{-}16$, depending on the model and lattice geometry, because for larger values the numerical costs become unfeasibly high.

Restricting the bond dimension of the $M$ tensors comes at a price: only a subset of states can efficiently be represented by a PEPS, since $D$ also limits the maximum amount of entanglement that can be captured by the construction. Fortunately, this is perfectly in line with the area law of the entanglement entropy in 2D, which is fully satisfied by a PEPS representation. Hence, PEPS are ideally suited to approximate low-energy states, including the ground state of local gapped Hamiltonians in two dimensions. Although this statement cannot yet be put on such a mathematically rigorous foundation as 1D, it is strongly supported by numerical evidence [54].

Moreover, the PEPS representation has the remarkable property that, in contrast to MPS, it is capable of faithfully representing algebraically decaying correlation functions, which are

characteristic for gapless models. This can easily be shown for the example of the partition function of the 2D Ising model [52]. Therefore, the PEPS ansatz is in principle able to also treat critical ground state wavefunctions. In practice, however, this does not help substantially in the context of 2D quantum criticality (the above mentioned example deals with classical and not quantum criticality). Based on the quantum-to-classical correspondence, one would require a 3D PEPS construction to faithfully approximate a critical 2D quantum system. Thus, in reality PEPS faces the same challenges in the context of gapless 2D systems as MPS treating critical 1D models: Both TN frameworks may yield results ranging from excellent to moderate quality depending on the "severeness" of the area-law violation in a particular system [53].

## 3.2 iPEPS

For finite-size PEPS simulations, each $M$ tensor is typically chosen to be different (similar to MPS applications for finite systems). Alternatively, it is possible to exploit the translational invariance of a system and directly work in the thermodynamic limit (of course, this approach also works for MPS [55]). In this way, finite-size and boundary effects can be completely eliminated.

In order to construct an infinite PEPS (iPEPS) [43], we first choose a fixed unit cell of a certain size, and repeat it periodically to cover the entire infinitely large lattice. The size of the fundamental unit cell directly translates into the number of different $M$ tensors required for the iPEPS representation. For instance, one can impose strict translational invariance and choose a unit cell of size $1 \times 1$,

$$|\psi\rangle = \qquad . \tag{8}$$

The resulting iPEPS representation of $|\psi\rangle$ then requires only a single $M$ tensor.

However, ordered ground states often break translational invariance to some degree. An iPEPS ansatz of type (8) cannot capture this behavior. Therefore, it is advisable to relax the translational invariance to some extent by choosing a larger unit cell. For example, the following ansatz is fully compatible with a antiferromagnetic ground-state order using two different $M$ tensors in a $2 \times 2$ unit cell:

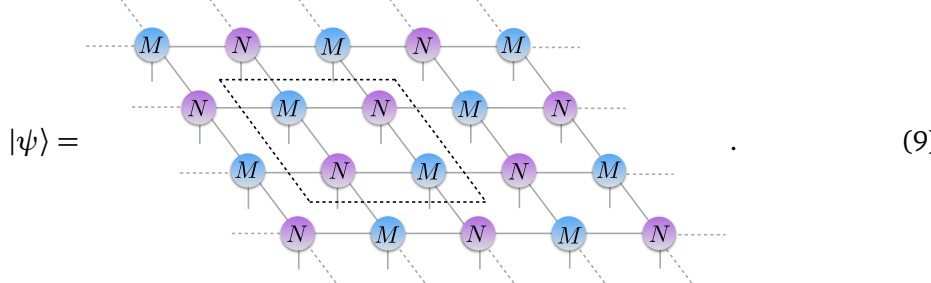

$$|\psi\rangle = \qquad . \tag{9}$$

In principle, unit-cells of arbitrary size can be considered, e.g.,

$$|\psi\rangle =$$ 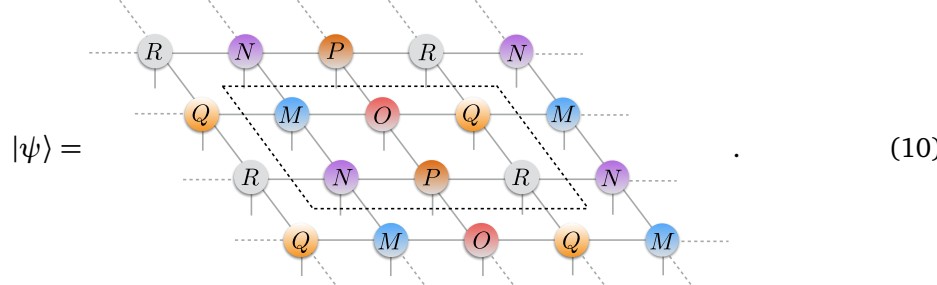 . (10)

The numerical costs scale linearly with the number of tensors in the unit cell, meaning that large unit cells become numerically expensive. A natural guideline to evaluate which unit-cell sizes should be considered in a simulation is to remember that the unit cell should be compatible with the actual ground-state order. Otherwise, one does not obtain the actual ground state from an iPEPS calculation. Instead, one ends up with the lowest-energy state for the system constrained to the corresponding unit-cell geometry and, therefore, is restricted to specific orders.

When studying systems with competing low-energy orders, the flexible unit-cell setup of the iPEPS algorithm actually becomes a big advantage. By probing different unit cells, it is possible to stabilize wavefunctions with competing orders independently. Comparing the energies obtained from the corresponding simulations, one may then determine which order survives in the ground state of the system [5, 6].

## 3.3 Contractions

To extract local observables, perform overlaps, or to actually optimize the tensors, the (i)PEPS framework requires contracting an (infinitely) large tensor network. This turns out to be much more challenging than in context of MPS where, for example, overlaps can be evaluated exactly with only polynomial costs in system size. For a PEPS tensor network, however, the calculation of an exact overlap represents an exponentially hard problem [56] and cannot be performed efficiently. Fortunately, there exist a variety of approximate schemes to deal with this issue.

In this review, we focus on the corner transfer matrix method (CTM) [57, 58], which is particularly well suited for iPEPS applications on square-lattice geometries. Alternatively, it is also possible to rely on an infinite MPS technique for the purpose of this work [43, 59, 60]. Other contraction schemes based on renormalization ideas, such as the tensor renormalization group [15, 16], or tensor network renormalization [19, 20], do have some technical disadvantages (e.g., environmental recycling [47, 61] is not possible, and difficulties arise when calculating longer-ranged correlators, ect.), rendering them unsuitable for our purposes.

Before discussing the details of the CTM scheme for evaluating the scalar product $\langle\psi|\psi\rangle$, we first introduce the corresponding diagram of $\langle\psi|$ for the $3 \times 3$ square-lattice toy example,

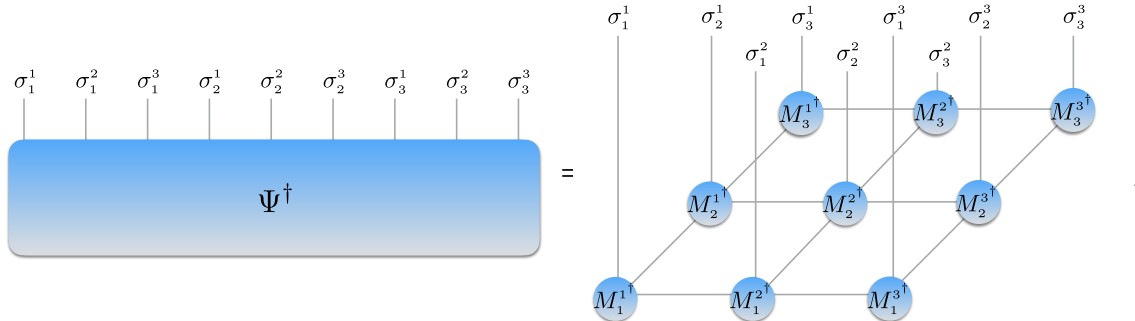

,

$$(11)$$

which is a mirror image of Eq. (7). The contraction of $\langle\psi|\psi\rangle$ can be done site by site using so-called reduced tensor $m = M_y^{x\dagger}M_y^x$, which is obtained by tracing over the joint physical index of $M_y^{x\dagger}M_y^x$,

$$m^x_{y(\alpha\alpha')(\beta\beta')(\gamma\gamma')(\rho\rho')} = \sum_{\sigma^x_y} M^{\sigma^x_y\dagger}_{\rho\gamma\beta\alpha} M^{\sigma^x_y}_{\alpha'\beta'\gamma'\rho'} =$$

$$= \qquad\qquad = \qquad\qquad , \qquad (12)$$

where the double indices (e.g., $(\alpha\alpha')$) have dimension $D^2$, as indicated by their increased line thickness. In the second line, we redrew the lines representing indices $\gamma$ and $\rho$ in such a way that pairs of corresponding primed and unprimed indices match up. This diagrammatically performed "index bending" exploits the non-uniqueness of the graphical representation for a tensor network [45]. This modification is completely trivial for bosonic iPEPS but will add additional complications in the context of fermions [see Sec. 4].

To reduce the complexity of the TN diagrams appearing in the following, we introduce a modified version of the conjugate tensor that automatically accounts for the index bending discussed in Eq. (12):

$$= \qquad\qquad . \qquad (13)$$

This distinction may seem unnecessary at this point, since $\bar{M}_y^{x\dagger}$ and $M_y^{x\dagger}$ are mathematically equivalent objects in the context of bosons. However, this is not the case for fermionic systems [c.f. Eq. (61)]. Therefore, we emphasize the importance of this modification already here.

The scalar product $\langle\psi|\psi\rangle$ for this simple example is obtained by contracting all physical and virtual index of the nine $m$ tensors,

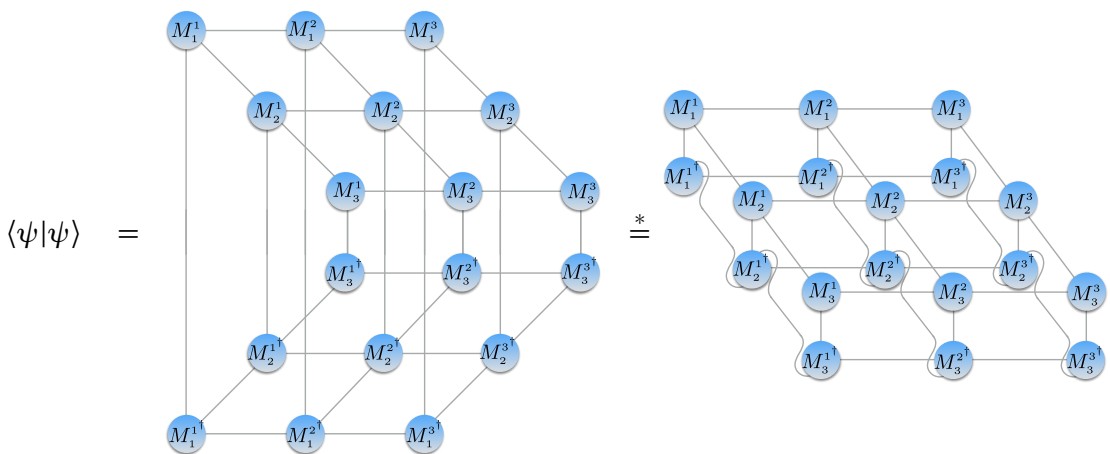



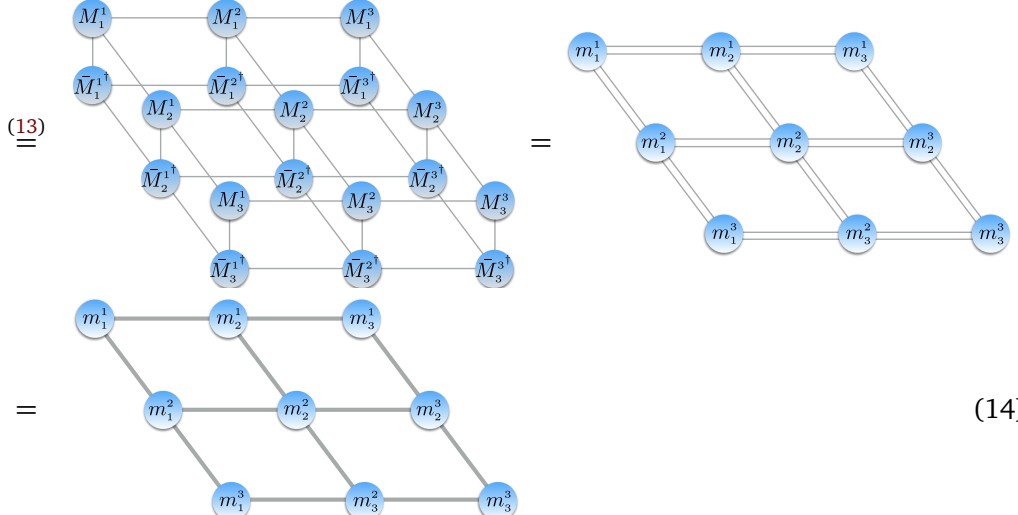

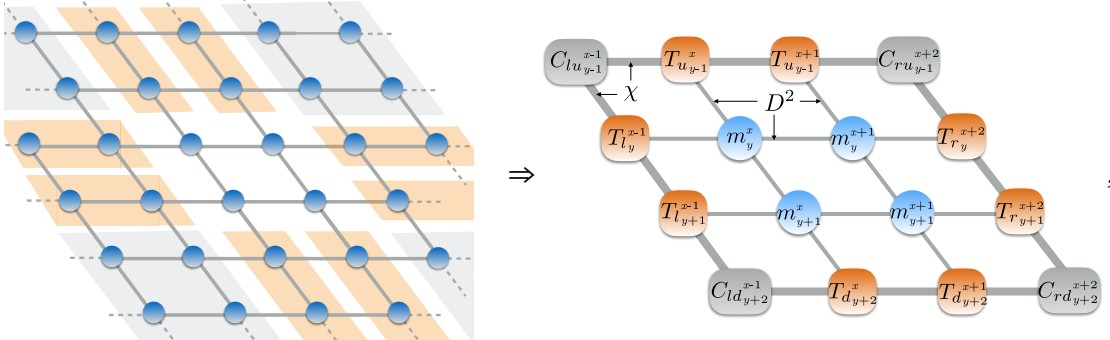

Note that the second step ($\stackrel{*}{=}$) also exploits the non-uniqueness of the diagrammatic representation by employing a number of so-called "jump-moves" [45]. In these operations, it is possible to drag a line over a tensor without changing the corresponding TN. For example, the line connecting $M_3^2$ and $M_3^3$ was dragged downward acroos $M_2^{3^\dagger}$. Again, this modification is trivial in context of bosonic PEPS, but nontrivial for fermionic PEPS [see Sec. 4].

Studying the small tensor networks in Eq. (14), it becomes obvious that the exact contraction of the expression scales exponential with system sizes. No matter in which order one decides to contract the tensors, i.e., which "contraction pattern" one uses, one always generates an object with a number of open indices scaling with $L$ (here $L = 3$).

### 3.3.1 Corner transfer matrix scheme

Since it is not possible to perform the exact calculation of a scalar product efficiently in the PEPS nor in the iPEPS framework, one has to rely on approximate approaches. A particularly powerful contraction scheme is based on ideas of the corner transfer matrix (CTM) renormalization group proposed by Nishino and Okunishi [57]. Their idea was later adapted by Orús and Vidal [58] in the context of quantum systems to efficiently evaluate an iPEPS tensor network contraction.

The key insight of the approach is to represent the infinitely large tensor network by a small number of tensors, zooming into a $1 \times 1$ or $2 \times 2$ window of sites (in general, this might be only a subset of the full unit cell, which in general has the size $L_x \times L_y$). The rest of the system, the so-called "environment", is represented by a set of corner matrices $C$ and transfer tensors $T$. For the $2 \times 2$ subset embedded in the environment, this takes the form

where the environmental tensor network is represented by a set of four corner matrices ($C_{lu}, C_{ld}, C_{ru}, C_{rd}$ with subscripts denoting the spatial location, i.e., $l, r, u, d$ stand for left,

right, up, down, respectively), and eight transfer tensors (two tensors for each direction, $T_l, T_r, T_u, T_d$, respectively). In this representation, a new set of virtual indices is introduced connecting tensors of the environment only. As we discuss below, the dimension $\chi$ of these indices acts as additional parameter controlling the accuracy of the environmental approximation (reasonable choices are $\chi \gtrsim D^2$).

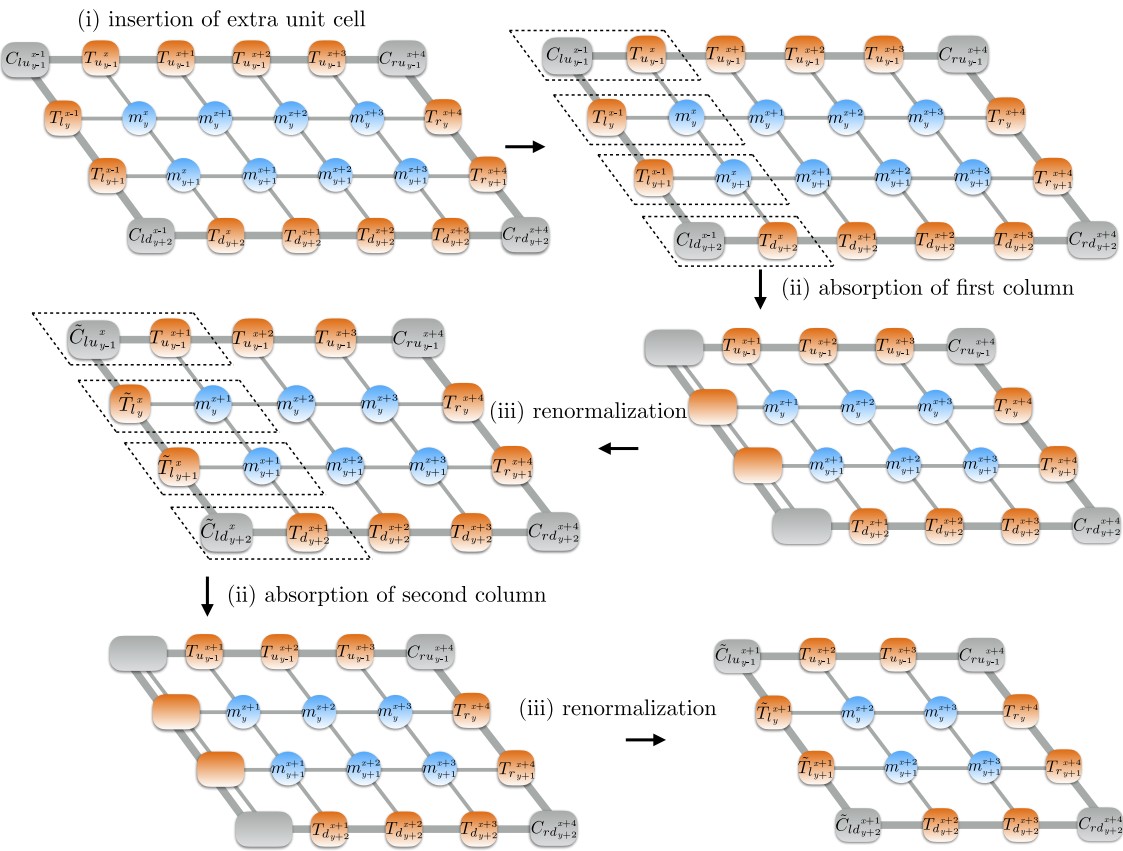

Figure 1: CTM coarse graining move to the left lattice direction: (i) extra unit cell is first inserted, and then column-wise integrated into the left part of the environment by performing two subsequent (ii) absorption and (iii) renormalization steps.

*CTM protocol.*– The environmental tensors are obtained by performing directional coarse graining moves in each direction of the lattice. Each coarse graining move consists of three different steps: (i) *insertion* of an extra unit cell; (ii) *absorption* of a single row or column of the unit-cell tensors into the set of environmental tensors in one lattice direction, leading to an enlarged environmental bond dimension $\chi D^2$; (iii) *renormalization* (or truncation/compression) of the enlarged environmental tensors to their original size. Steps (ii) and (iii) are repeated until the inserted unit cell has been fully absorbed into the set of environmental tensors in the one particular direction. Next, an additional unit cell is inserted next to the original unit cell in one of the other directions, and the move is carried out with respect to another direction of the lattice. A full coarse graining step is completed after one move in each of the four lattice directions (left, right, top, bottom) has been performed.

In the following, we illustrate this procedure for an iPEPS representation with a $2 \times 2$ unit cell, using four $M$ tensors that all have the property $M_y^x = M_y^{x+2} = M_{y+2}^x = M_{y+2}^{x+2}$ (as in Eq.9). A directional move to the left then includes the steps illustrated in Fig. 1. Note that

the extra unit cell has been inserted horizontally on the left (this is also the case for a move to the right). Moreover, two absorption and renormalization steps are carried out, at the end of which the inserted unit cell has been fully integrated into the left part of the environment. This set of operations yields an updated set of environmental tensors for the direction of the coarse graining step.

We also sketch in Fig. 2 a coarse graining move towards the top of the lattice. In this case,

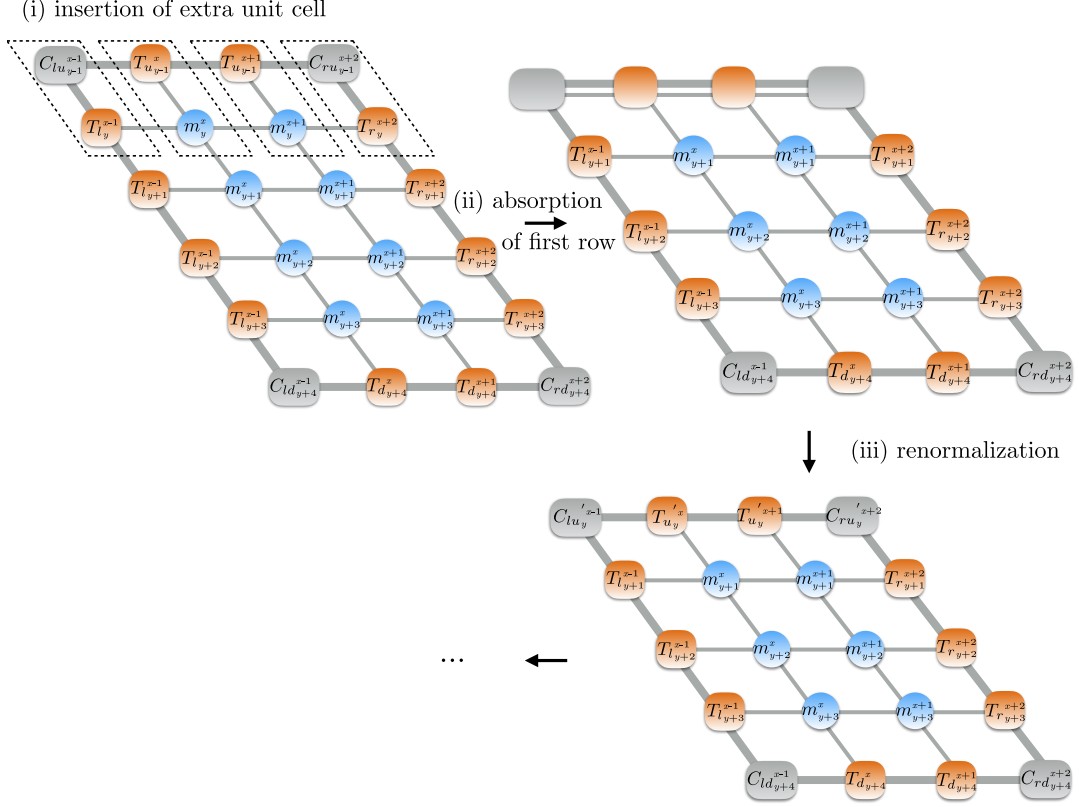

Figure 2: CTM coarse graining move to the top of the lattice: (i) extra unit cell is first inserted, and then row-wise integrated into the upper part of the environment by performing two subsequent (ii) absorption and (iii) renormalization steps (only first step is shown).

the unit cell is inserted vertically. Then we follow the same protocol as for the left move. Only the direction of the absorption and renormalization steps differs. After also carrying out these coarse graining moves with respect to the other two lattice directions, a full coarse graining step has been completed. The full cycle is typically repeated multiple times depending on the correlation length in the system. For example, for a gapped system a few ($\sim 10$) steps may be sufficient to obtain converged results. However, for a critical system, due to the absence of the energy gap, the number of steps required to reach convergence in local observables can be significantly larger, up to $\gtrsim 100$ steps.

*Renormalization.*– In addition to the number of steps performed, the convergence of the results also strongly depends on the implementation of the renormalization step, which truncates the environmental tensors after the absorption step. The renormalization is crucial for the performance of the CTM scheme. However, its implementation details are not very straightforward, and currently there seems to be ample room for future improvement. The ambiguity of implementation details is mostly caused by the lack of an exact canonical representation for

a PEPS TN, which implies that there is no obvious optimal way of performing the truncation (in contrast to an MPS tensor network, which can be truncated optimally even in the context of translationally invariant systems [60]).

We list and comment on a number of different renormalization schemes. One corresponds to the directional updated scheme proposed by Orús and Vidal in Ref. [58], which we found to work well only in the context of very homogenous wavefunctions. This method takes only small subsets of the environment into account and implicitly assumes full translational invariance when generating the projectors (or isometries) to perform the truncation. This ultimately yields a very biased truncation pattern for inhomogeneous systems, where this method is bound to fail. The second approach is based on the original CTMRG of Nishino and Okunishi [57] and was first employed by Corboz, Jordan and Vidal Ref. [24] in the context of iPEPS. In this case, the full environment is taken into account in each truncation step, which presents a crucial advantage for simulating inhomogeneous states. On the other hand, it is severely limited by machine precision, making it unstable for large values of environmental bond dimension $\chi$. This is far from ideal since it is desirable to use $\chi$ as additional control parameter. To overcome these shortcomings, Corboz, Rice and Troyer Ref. [6] introduced a third CTM variant that shows strongly improved convergence properties in comparison to the original CTMRG scheme and, at the same time, overcomes the inhomogeneity issues of the directional updated scheme. In the following, we sketch how to obtain the projectors used to reduce the sizes of the environmental tensors after an absorption step in the left direction, following Ref. [6]. The protocol works similarly for the other spatial directions of the lattice.

In the first step, we enforce two cuts in the tensor network consisting of the $2 \times 2$ unit-cell subset embedded in the effective environment as follows

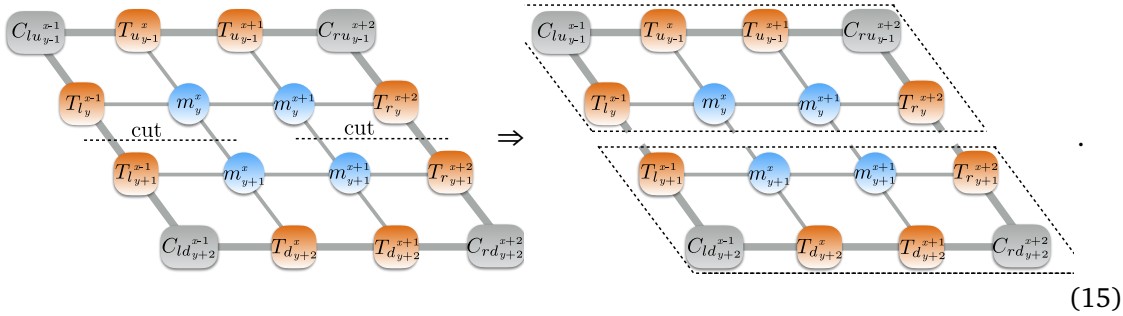

$$(15)$$

Our goal is to obtain projectors (or isometries) that are inserted after an absorption step at a specific bond to "project" (or truncate/compress) the enlarged environmental Hilbert space $D^2\chi$ back to its original size $\chi$. In this example, we specifically aim for the projectors to be inserted into the two bonds split by the left cut.[1] To this end, we contract the two upper and lower parts of the tensor network, leading to rank-4 tensors $Q_u$ and $Q_d$. By applying a singular value (or QR) decomposition to both of these tensors, we obtain

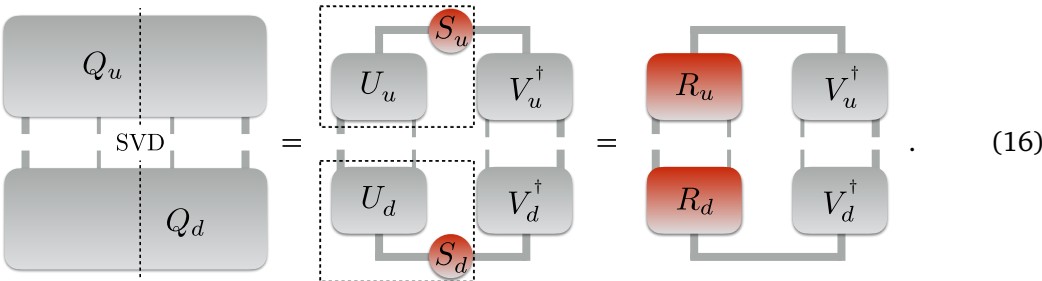

$$(16)$$

---

[1]Analogously, we could use (15) to obtain the projectors for the two split bonds on the right. This becomes necessary when performing a CTM move into the right direction of the lattice.

The product $R_u R_d$ is then subjected to an additional SVD where only the $\chi$ largest singular values are kept,

$$
\begin{array}{c}
\boxed{\begin{matrix} R_u \\ R_d \end{matrix}} = \text{SVD} \approx \begin{matrix} U & \leftarrow \chi D^2 \\ S \;\chi \\ V^\dagger & \chi D^2 \rightarrow \end{matrix} \Rightarrow \begin{matrix} R_d^{-1} \\ R_u^{-1} \end{matrix} \approx \begin{matrix} V \\ S^{-1} \\ U^\dagger \end{matrix} \, . \quad (17)
\end{array}
$$

Using the inverse matrices $R_u^{-1}$ and $R_d^{-1}$, we generate the projectors $P_y^x$, $\tilde{P}_y^x$ that are inserted at the left cut of the tensor network (15):

$$
\mathbb{I} = \begin{matrix} R_d \\ R_d^{-1} \\ R_u^{-1} \\ R_u \end{matrix} \approx \begin{matrix} R_d \\ V \\ \sqrt{S^{-1}} \\ \hline \sqrt{S^{-1}} \\ U^\dagger \\ R_u \end{matrix} = \begin{matrix} \chi \rightarrow \tilde{P}_y^x \leftarrow D^2 \\ \chi \\ \chi \rightarrow P_y^x \leftarrow D^2 \end{matrix} \, . \quad (18)
$$

The protocol is repeated for the entire row of the unit cell to be absorbed into the environment during this particular coarse graining step (i.e., $L_y$ times). In our example of an $2 \times 2$ unit cell, we therefore also obtain $P_{y+1}^x$ and $\tilde{P}_{y+1}^x$ (or alternatively $P_{y-1}^x$ and $\tilde{P}_{y-1}^x$ due to translational invariance) by considering the tensor network and repeating the procedure sketched above,

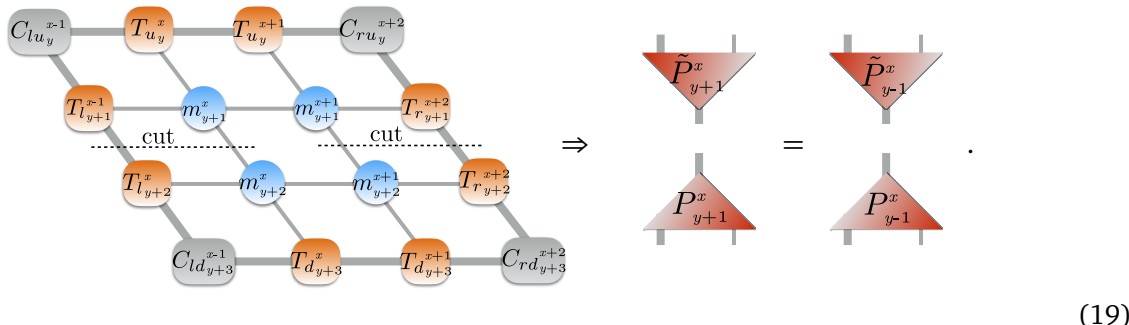

$$(19)$$

Now we are fully equipped to renormalize the entire set of environmental tensor which are

subject to truncation during an absorption step to the left,

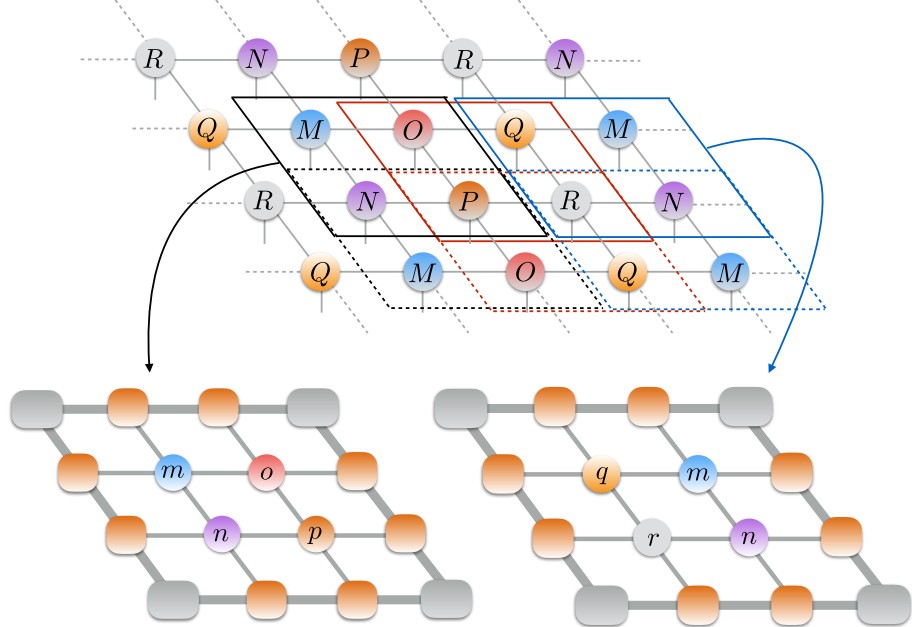

$$
\quad = \quad . \qquad (20)
$$

What has been achieved is a scheme that compressed the bond dimensions of the environmental tensors along the left row in a way that encodes information from the full environment. Thus we can appropriately deal with translational symmetry breaking in the iPEPS wavefunction. At the same time, this procedure leads to numerically stable results since we can eliminate spurious parts of the SVD spectrum during the intermediate SVD decompositions in Eq. (16) by discarding very small singular values (e.g., $< 10^{-7}$). This helps to reduce the influence of numerical noise in the subsequent steps.

**Figure 3:** A unit cell of size $3 \times 2$ consists of six different $M$ tensors (here denoted $M, N, O, P, Q$, and $R$). For each of the six relative coordinates in the unit cell, we have to obtain a $2 \times 2$ CTM representation (indicated by the solid and dashed squares, and explicitly illustrated for two examples). Therefore, the CTM scheme here requires storing 24 corner matrices and 24 transfer tensors in total.

*Larger unit cells.–* The CTM scheme can also deal with rectangular unit cells of arbitrary

sizes containing $L_x \times L_y = N$ different $M$ tensors, where the relative position of each tensor in the unit cell is labeled by its coordinate $\mathbf{r} = (x, y)$. To this end, we assign one set of corner matrices and transfer tensors to *each* coordinate, requiring a total number of $4N$ corner matrices and $4N$ transfer tensors to be stored independently. We illustrate this approach for a $3 \times 2$ unit cell in Fig. 3. After initialization (see below), the environmental tensors are then iteratively updated by performing coarse graining moves in all four lattice directions, as outlined above. However, an entire CTM cycle now includes $L_x$ coarse graining steps to the left and right, respectively, as well as $L_y$ coarse graining steps to the top and $L_y$ to the bottom of the lattice. Note that using a larger zooming window is not an option, since the numerical costs quickly become unfeasible.

*Initialization.*– While covering the coarse graining procedure to obtain the converged environmental tensors, we have not yet discussed the initialization of the CTM scheme. In principle, one could start from an arbitrary set of corner matrices and transfer tensors. However, choosing a completely random set can significantly increase the number of coarse graining steps required for obtaining a stable environment TN and sometimes even cause numerical instabilities. In practice, we found that optimal convergence is achieved by starting from an environmental tensor set formed by the corresponding $M_y^x$ tensors and their conjugates, which previously have been generated by means of ground-state optimization [see Sec. 3.5]. We illustrate this initialization procedure for two examples,

$$
C_{ld_y^x} \;=\; \cdots \;,\qquad T_{l_y^x} \;=\; \cdots \;. \tag{21}
$$

*Effective contraction pattern.*– The numerical costs of implementing the square-lattice CTM scheme presented above scales as $\mathcal{O}(D^6 \chi^3)$, with iPEPS bond dimension $D$ and environmental bond dimension $\chi$. Note that these costs are equivalent to those of the infinite MPS method from Ref. [43]. Assuming that $\chi = \mathcal{O}(D^2)$, we end up with a total cost scaling of $\mathcal{O}(D^{12})$ for the iPEPS algorithm. The underlying assumption behind this cost scaling is that all contractions are carried out as efficiently as possible, which forces us to pay some attention to the contraction patterns. In particular, we cannot directly work with the reduced tensors $m_y^x$, but rather need to perform contractions involving $M_y^x$ and its conjugate $M_y^{x\dagger}$ sequentially.

This is illustrated below for contracting a part of the diagram in Eq. (15). First consider the case explicitly using the reduced tensor $m_y^x$,

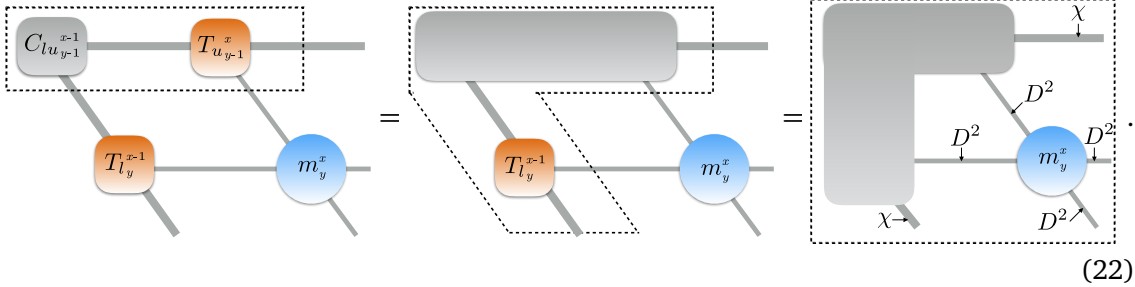

$$\tag{22}$$

Counting the involved indices in the dashed box, it becomes clear that the last contraction step scales rather unfavorably as $\mathcal{O}(D^8 \chi^2)$.

If we want to achieve the optimal scaling $\mathcal{O}(D^6 \chi^2 d)$ in this step, we have to contract over $M_y^x$ and $M_y^{x\dagger}$ sequentially,

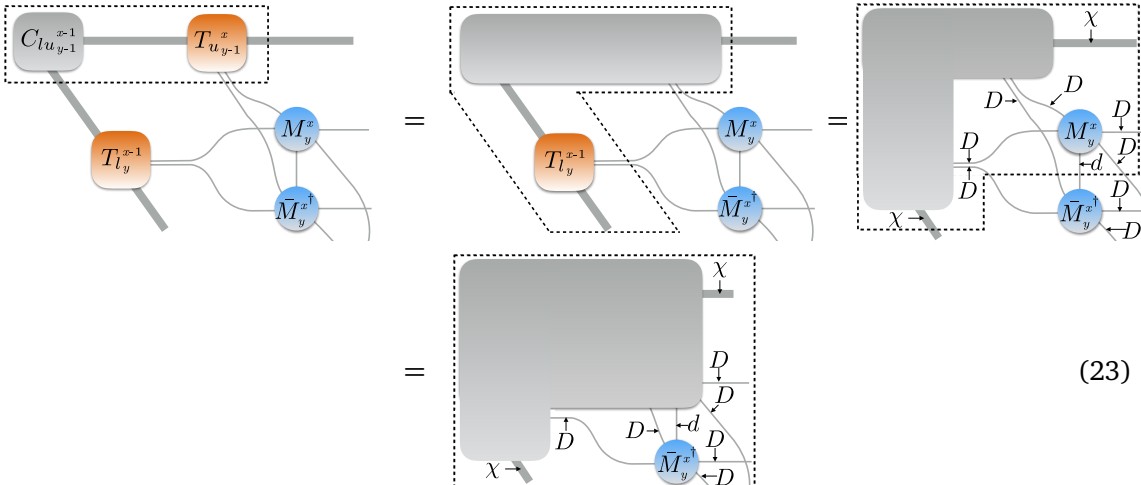

$$\tag{23}$$

The same applies to contraction orders of other TN such as, for example, the one shown in Eq. (20) and many others. It pays off to constantly pay attention and ensuring that the optimal contraction pattern is used when implementing an iPEPS algorithm. Otherwise, the backlash of an inefficient iPEPS implementation will quickly become apparent, since simulations with moderate to large $D$ will not be feasible. Note that the most expensive steps of the CTM algorithm occur when generating the projectors. To obtain the tensor $Q_u$ in Eq. (16), for instance, one has to perform the contraction,

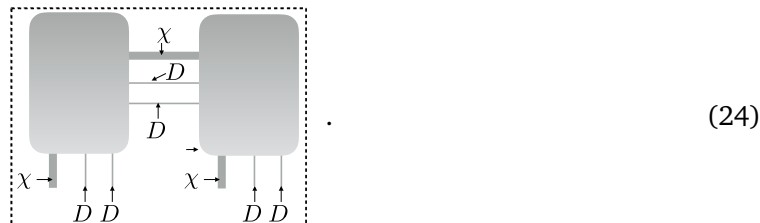

$$\tag{24}$$

This always yields a cost scaling of $\mathcal{O}(\chi^3 D^6)$ which cannot be reduced further.

## 3.4 Expectation value

The CTM scheme enables us to evaluate observables within the iPEPS framework. For this case, we consider a simple two-site observable $\hat{O}_{(x,y)}^{(x+1,y)}$ which, for example, represents a spin-spin correlation function involving two neighboring sites. To compute an approximation for the expectation value $\langle \hat{O}_{(x,y)}^{(x+1,y)} \rangle = \langle \psi | \hat{O}_{(x,y)}^{(x+1,y)} | \psi \rangle / \langle \psi | \psi \rangle$, we represent the environment of the two contiguous sites $\boldsymbol{r} = (x, y)$ and $\boldsymbol{r}' = (x, y+1)$ in terms of the corner matrices and transfer tensors encountered in the last section,

$$\langle \psi | \hat{O}_{(x,y)}^{(x+1,y)} | \psi \rangle_\chi \quad = \quad$$ 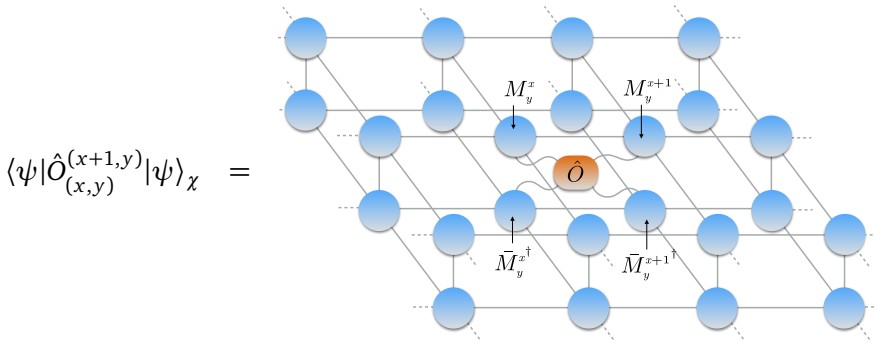

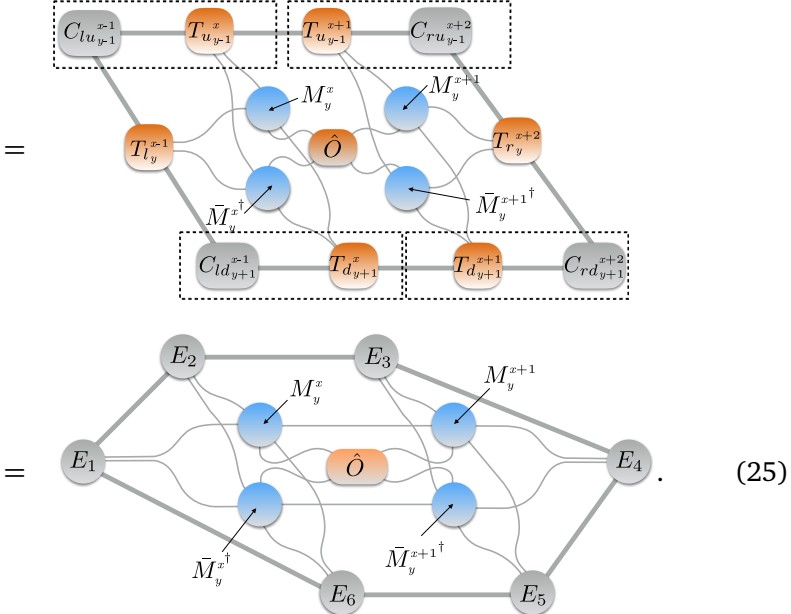

$$= \qquad . \tag{25}$$

The contraction of the final tensor network, consisting of the six environmental tensors $E_1, \dots, E_6$, the two $M$ tensors, their conjugates, and the operator $\hat{O}$ can be carried out efficiently. It produces an approximation of $\langle\psi|\hat{O}_{(x,y)}^{(x+1,y)}|\psi\rangle \approx \langle\psi|\hat{O}_{(x,y)}^{(x+1,y)}|\psi\rangle_\chi$ which is generally expected to deviate from the exact value due to the non-exact representation of the full tensor network. The correct value of $\langle\psi|\hat{O}_{(x,y)}^{(x+1,y)}|\psi\rangle/\langle\psi|\psi\rangle \approx \langle\psi|\hat{O}_{(x,y)}^{(x+1,y)}|\psi\rangle_\chi/\langle\psi|\psi\rangle_\chi$ should be recovered in the limit $\chi \to \infty$. In practice, one evaluates Eq. (25) for a number of different values of $\chi = 10, 20, \dots, 100, 150, \dots$ until the observable shows no more significant dependence on $\chi$. The required value for $\chi$ to obtain converged results strongly varies depending on the physical properties of the corresponding system and the employed iPEPS bond dimension $D$. If one is already well within the relevant low-energy critical regime, it can therefore be useful to extrapolate observables towards $1/\chi \to 0$ and $1/D \to 0$ [62, 63]. A theoretical justification for such an approach is based on the theory of finite entanglement scaling, which has been well analyzed in the one-dimensional scenario [64–67].

## 3.5 Ground state search

An iPEPS is an approximate representation for the ground-state wavefunction of a local Hamiltonian on a two-dimensional lattice. Having addressed the contraction issue by means of the CTM scheme [see previous Sec. 3.3], the remaining open question concerns finding the ground-state iPEPS representation, given some Hamiltonian $\hat{H}$ with only nearest-neighbor interactions. (Albeit technical more complicated, iPEPS can also treat longer-ranged interactions, for more details see Ref. [24, 38].)

Here we follow the strategy proposed in the original iPEPS formulation by Jordan, Orús, Vidal, Verstraete and Cirac [43], and use the imaginary time evolution to target the ground state,

$$|\psi_0\rangle = \lim_{\tau\to\infty} \frac{e^{-\tau\hat{H}}|\psi\rangle}{\left|\left|e^{-\tau\hat{H}}|\psi\rangle\right|\right|}. \tag{26}$$

The time-evolution operator $e^{-\tau\hat{H}}$ is further decomposed by Suzuki-Trotter decomposition,

$$e^{-\hat{H}\tau} \approx \prod_{j=1}^{N_b} e^{-\hat{h}_{y,y'}^{x,x'}\tau} + \mathcal{O}(\tau^2), \tag{27}$$

where $\hat{h}^{x,x'}_{y,y'}$ describes the local interaction terms acting on a pair of nearest-neighbor sites in the unit cell, and $\hat{H} = \sum_{\langle (x,y),(x',y')\rangle} \hat{h}^{x,x'}_{y,y'}$. The two-site gates, $e^{-\hat{h}^{x,x'}_{y,y'}\tau}$, are subsequently applied to the corresponding pairs of $M$ tensors, $M^x_y$ and $M^{x'}_{y'}$. As in the case of MPS, the resulting tensor has to be truncated accordingly to restore the original form of the iPEPS representation.

In the MPS framework, the truncation can be implemented in an optimal way using the canonical form of the MPS and employing a single singular value decomposition. In the context of iPEPS, this step turns out to be more evolved. Due to the lack of an exact canonical form for the iPEPS, one has to rely on approximate techniques to account for the effects of the environment when employing the truncation. This can be done using several different optimization schemes, such as the *simple update* [68] and the *full update* [43]. We discuss both of these approaches extensively in the rest of this section.

Although not employed in the context of this review, we also note that two groups recently introduced alternative optimization schemes, which do not rely on imaginary time evolution [21, 22]. Instead, they implement a variational update method,

$$\min_{\{M^x_y\}}\left[E_0\right] = \frac{\langle\psi_0|\hat{H}|\psi_0\rangle}{\langle\psi_0|\psi_0\rangle}. \tag{28}$$

The major technical challenge of these newly developed schemes is to find an approximate, yet accurate, representation for the full Hamiltonian $\hat{H}$. Corboz [21] achieves this based on a modified CTM scheme, while Vanderstraeten, Haegeman, Corboz and Verstraete [22] build on MPS techniques. In addition, it is still unclear how to optimally translate the local update performed on a pair of tensors to the iPEPS representation in the infinite system. Despite these issues, both variational optimization techniques already obtain very impressive results, illustrating that the iPEPS formalism will continuously improve and become more competitive in the near future.

### 3.5.1 Bond projection

In this work, we only consider the optimization via imaginary-time evolution based on two-site Trotter gates, which implies that we constantly have to update two neighboring $M$ tensors at once (i.e., there is no one-site version of this algorithm). Hence, it is essential to perform the tensor updates as efficiently as possible. Treating the full $M$ tensors in this context turns out to be numerically very inefficient (i.e., numerical costs of $\mathcal{O}(D^{12})$ in the context of the full update). Instead, it is always advisable to perform the tensor update on two subtensors with lower rank which are easily obtained by a bond projection [69], leading to a significant cost reduction (i.e., $\mathcal{O}(D^6 d^3)$ [47]. Note that this scheme does not introduce further approximations since the two-site Trotter gate only changes properties of the corresponding bond but leaves the remaining bonds of the iPEPS tensors unchanged.

The bond projection is obtained by performing two exact SVD (or QR) decompositions:

$$\tag{29}$$

The tensor optimization now only affects the subtensors $v_y^x$ and $w_y^{x+1}$, whereas the remaining bonds are shifted into the subtensors $X_y^x$ and $Y_y^x$, which can be treated as parts of the environment tensor network during the optimization.

Each tensor update is initialized by applying the corresponding Trotter gate in the bond projection,

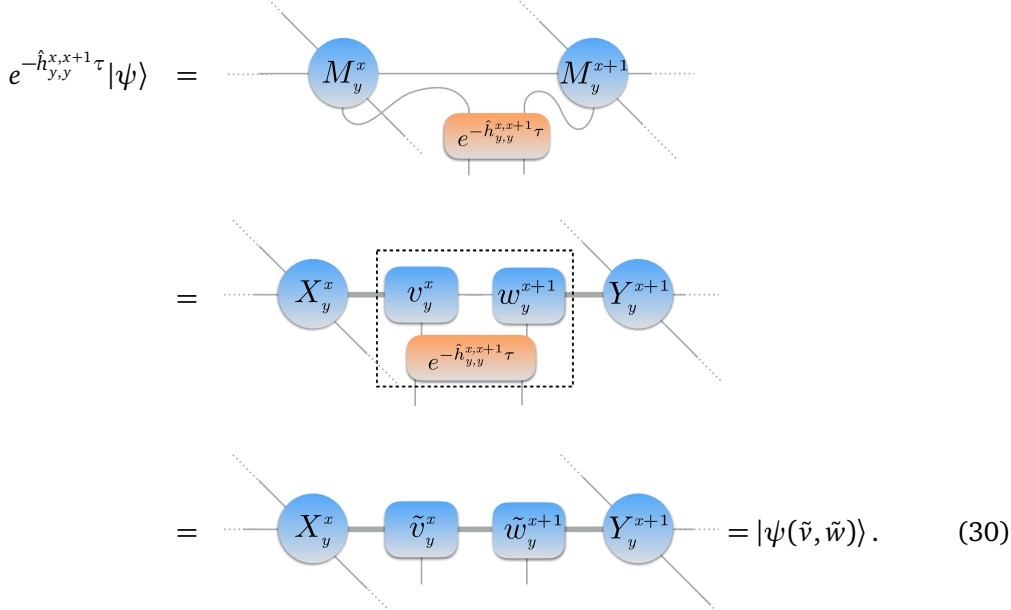

$$\tag{30}$$

The Trotter gate increases the initial bond dimension $D$ of the subtensors $v_y^x$ and $w_y^{x+1}$. Restoring the original representation exactly yields a pair of enlarged subtensors $\tilde{v}_y^x$ and $\tilde{w}_y^{x+1}$ with bond dimension $dD$ (illustrated by the increased line thickness in Eq. (30)). In a next step, we have to find an appropriate truncation scheme to obtain a pair of subtensors $v'^x_y$ and $w'^{x+1}_y$ with the original bond dimension $D$ to prevent an exponential blowup of the iPEPS tensors.

In the following, we present two different truncation methods: (i) the simple update [68], a numerically very efficient and fast approach which, however, relies on a strong simplification of the environmental tensor network and thus carries out the truncation in a suboptimal way; (ii) the full update scheme [43] which leads to an optimal truncation by incorporating the effects of the entire wavefunction appropriately. However, the full update comes at the price of requiring significantly more numerical resources.

### 3.5.2 Simple update

The simple update, introduced by Jiang, Weng and Xiang Ref. [68] is formulated in a slightly modified iPEPS representation. So far, we only dealt with $M$ tensors located directly at sites of the lattice. For the simple update we put an extra set of tensors on the bonds of the iPEPS tensor network. These tensors, here labeled $\lambda_y^x$ for horizontal and $\tilde{\lambda}_y^x$ for vertical bonds, are diagonal matrices similar to those used in Vidal's TEBD and iTEBD formulation for time-evolving matrix product states [59,70].

Starting from the standard iPEPS representation that has been adopted in this review, so

far, it requires only a minor adaption to translate into this modified representation,

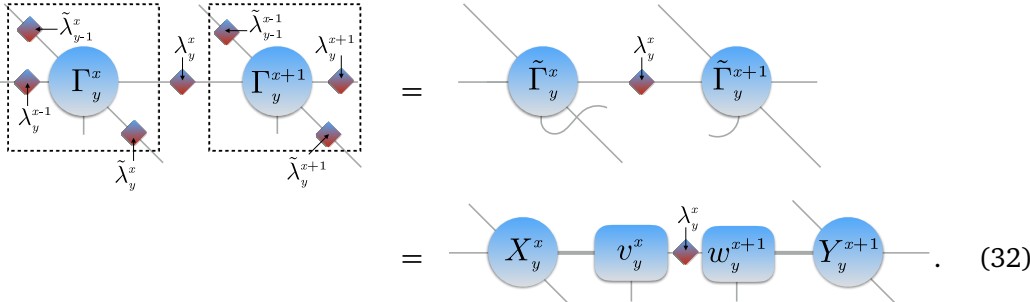

$$(31)$$

where $\Gamma^x_y$ in combination with the roots of all four bond tensors yields the original $M^x_y$ tensor. The key idea of the simplified update is to approximate the full environment of two neighboring sites, $r = (x, y)$ and $r' = (x+1, y)$, by only the diagonal tensors surrounding this pair of sites. This procedure is adopted from MPS-based time evolution via the iTEBD algorithm.

To perform the simple update explicitly, we switch first into the bond projection to carry out the optimization more efficiently. We illustrate the projection here explicitly since different tensors are involved in the modified iPEPS representation,

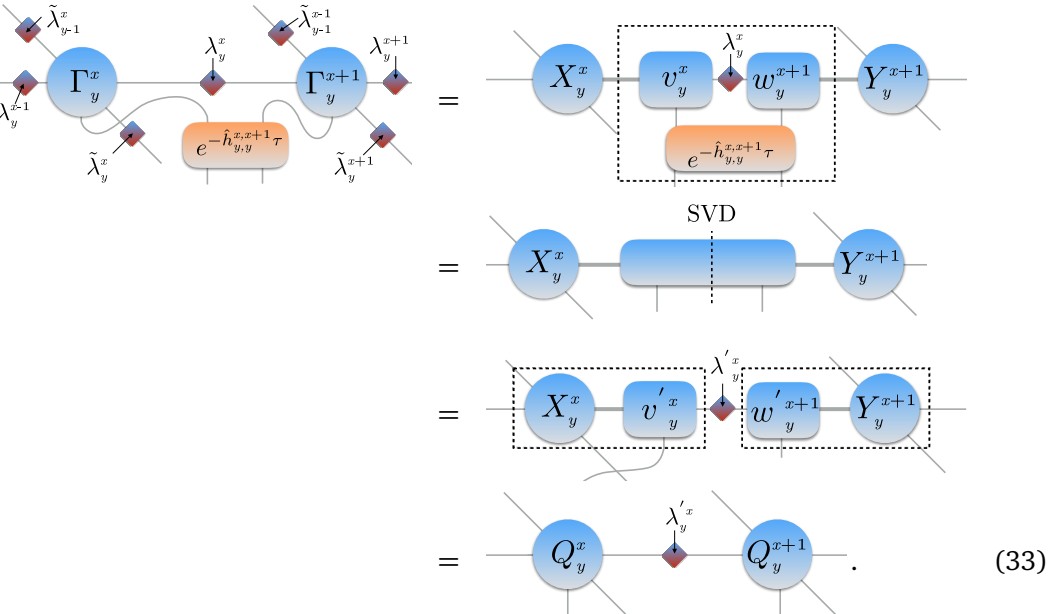

$$(32)$$

Now the Trotter gate is applied to the subtensors on the bond, adding entanglement and potentially increasing the bond dimension to $dD$. To obtain the pair of subtensors $v'^x_y$ and $w'^{x+1}_y$ with the original bond dimension $D$, the simple update relies on a simple SVD,

$$(33)$$

No extra iteration or optimization is required to complete the update (hence, the name "simple" update). The updated diagonal bond matrix $\lambda'^x_y$ contains the $D$ largest singular values, the optimized subtensors are obtained from $v'^x_y = U$ and $w'^{x+1}_y = V^\dagger$.

To restore the form of the iPEPS tensors from $Q_y^x$ and $Q_y^{x+1}$, we apply the inverse of the additional bond tensors, which have not been altered by this optimization step,

$$
\Gamma'^{x}_{y} = \tilde{\lambda}_{y-1}^{x^{-1}} \, \lambda_y^{x-1^{-1}} \, Q_y^x \, \tilde{\lambda}_y^{x^{-1}} \quad , \quad \Gamma'^{x+1}_{y} = \tilde{\lambda}_{y-1}^{x^{-1}} \, Q_y^{x+1} \, \lambda_y^{x^{-1}} \, \tilde{\lambda}_y^{x^{-1}} \, . \tag{34}
$$

The simple update is particular appealing due to its low complexity and high numerical efficiency; the truncation based on a plain SVD in Eq. (33) only scales with $\mathcal{O}(D^3 d^6)$ operations. Yet, the truncation itself cannot be considered optimal in the context of iPEPS. It would have been optimal if we had gauged the surrounding bonds in such a way that they exclusively contain orthonormal basis sets. Unfortunately, this is only possible if the environment is separable, as in the case of MPS or other tensor networks *without loops*. In fact, one can show that a tensor optimization performed in this way presents an optimal update for an infinite tensor network on a Bethe lattice [69].

Any iPEPS representation on a standard 2D lattice, however, does feature loops, which means that we cannot separate the environment into two blocks and find a gauge with orthonormal basis sets on all surrounding bonds. Hence, the simple update introduces a systematic error, as it does not properly account for the full environment of the bond during the optimization. The magnitude of this error turns out to be less severe than one might expect. Especially for systems in gapped phases, the simple update leads to excellent results [45]. Moreover, its numerical efficiency often allows simulations with larger bond dimensions compared to the full update; thus it can give access to complex systems which remain out of reach for full-update calculations.

We conclude this section with a few practical comments concerning the implementation of the simple update:

- For a generic unit cell, the simple update is employed sequentially on all bonds in the system. One can easily work with a second-order Trotter decomposition by reversing the application order of the gates in every second step.

- The normalization of the tensor network can be conveniently achieved on the fly by normalizing the trace of each updated diagonal bond matrix $\lambda'^x_y$ to unity. This procedure leads to a numerically fully stable algorithm.

- To obtain a meaningful representation of the ground state by means of imaginary-time evolution, we start from a random set of tensors and use a fairly large time step $\tau = \mathcal{O}(10^{-1})$. A large initial time step is important since it minimizes the risk of getting stuck in a local energy minimum and, in case of symmetric iPEPS implementation, it enables us to dynamically adapt the symmetry sectors on the bonds (starting from a very small time step, one can get stuck in the initial symmetry configuration and not reach all relevant sectors). To decrease the effect of the Trotter error, we then gradually reduce $\tau$ as soon as we observe convergence with respect to the SVD spectra (typically after a few hundred or thousand time steps). After reaching a time step of the order $\mathcal{O}(10^{-5})$, the ground-state wavefunction is typically converged.

- Measurements of observables are performed with the converged iPEPS representation, obtained from the simple update, as input for the CTM scheme. Relying on CTM, this leads to a total numerical cost scaling of $\mathcal{O}(\chi^3 D^6)$, which is, in principle, equivalent to the cost scaling of the full update. In the latter, however, the full environment has to be calculated in every step and not just at the end to perform measurements.

### 3.5.3 Full update

The full update introduced by Jordan, Orús, Vidal, Verstraete and Cirac [43] represents a clean and accurate protocol for performing the tensor update during imaginary-time evolution. Its name is derived from the fact that the effects of the entire wavefunction on the bond tensors are considered, including the full environmental TN. The only approximation stems from the non-exact contraction of the environmental TN, which we carry out based on the CTM scheme [see Sec. 3.3.1].

After the application of the Trotter gate in Eq. (30), the full update generates the optimized pair of subtensors $v'^x_y$ and $w'^{x+1}_y$ with bond dimension $D$ by minimizing the squared norm between $|\psi(v', w')\rangle$ and the wavefunction $|\psi(\tilde{v}, \tilde{w})\rangle$ containing the exact subtensors $\tilde{v}^x_y$ and $\tilde{w}^{x+1}_y$ with enlarged bond dimension $dD$,

$$d(\tilde{v}, \tilde{w}, v', w') = \left|\left|\left|\psi(v', w')\rangle - |\psi(\tilde{v}, \tilde{w})\rangle\right|\right|\right|^2. \tag{35}$$

To minimize Eq. (35) with respect to $v'^x_y$ and $w'^{x+1}_y$, we first have to obtain an effective representation of the environment with respect to the bond to be updated (marked red):

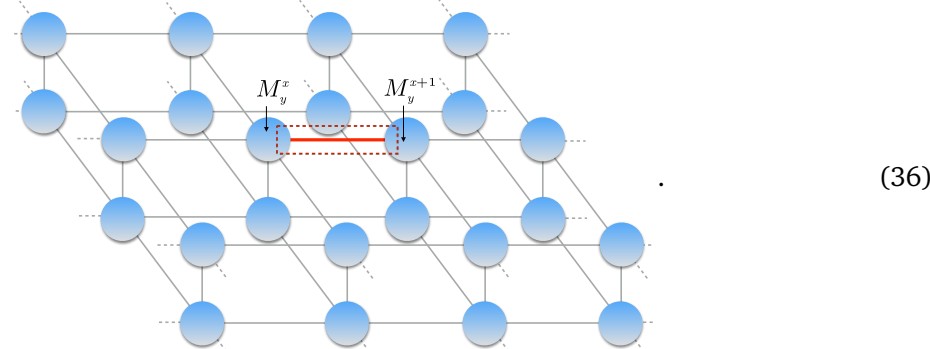

. (36)

This is achieved via the CTM scheme, leading to an approximate representation of the environment in terms of corner matrices and transfer tensors,

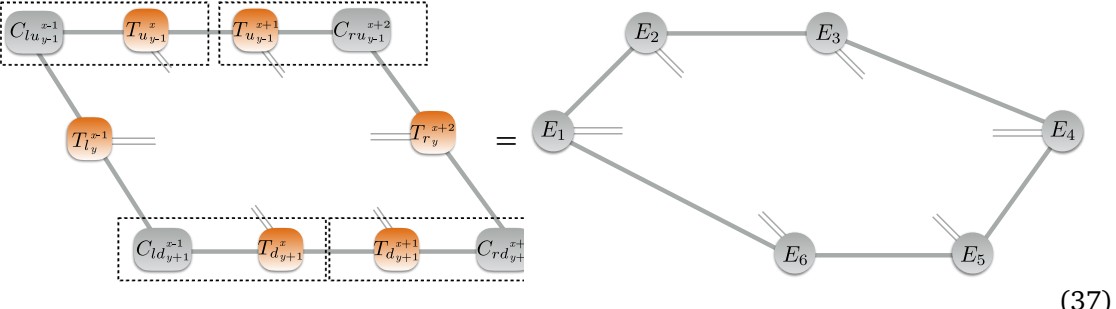

(37)

As in the case of the simple update, we carry out the tensor update for efficiency reasons in the bond projection, as discussed above. In order to generate the full environment in this representation, we have to account for the subtensors $X^x_y$ and $Y^{x+1}_y$ as well as their conjugates, and multiply them to the effective environment shown in Eq. (37), obtaining

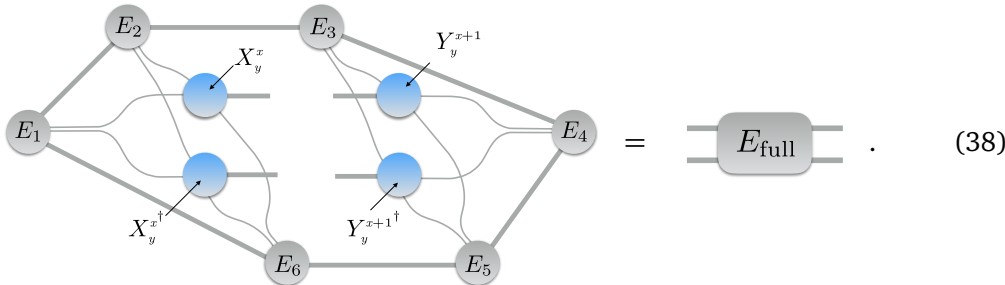

. (38)

In this way, it is possible to represent the cost function (35) diagrammatically,

$$
\begin{aligned}
&d(\tilde{v}, \tilde{w}, v', w') \\
&= \langle \psi(v', w') | \psi(v', w') \rangle + \langle \psi(\tilde{v}, \tilde{w}) | \psi(\tilde{v}, \tilde{w}) \rangle - \langle \psi(v', w') | \psi(\tilde{v}, \tilde{w}) \rangle - \langle \psi(\tilde{v}, \tilde{w}) | \psi(v', w') \rangle
\end{aligned}
$$

$$
=
\begin{array}{c}
\text{(diagram)}
\end{array}
\; + \;
\begin{array}{c}
\text{(diagram)}
\end{array}
$$

$$
-
\begin{array}{c}
\text{(diagram)}
\end{array}
\; - \;
\begin{array}{c}
\text{(diagram)}
\end{array}
. \tag{39}
$$

$d(\tilde{v}, \tilde{w}, v', w')$ is a quadratic function of the tensors $v'^x_y$ and $w'^{x+1}_y$. Thus, the optimized subtensors can be found using an alternating least-square algorithm [43].

To this end, we can first optimize $v'^x_y$ while keeping $w'^{x+1}_y$ fixed. Analogous to the MPS compression, we form the partial derivative of Eq. (39) with respect to $v'^{\dagger,x}_y$,

$$
\frac{\partial}{\partial v'^{\dagger}} d(\tilde{v}, \tilde{w}, v', w') \overset{!}{=} 0 \quad \Rightarrow \quad
\begin{array}{c}
\text{(diagram)}
\end{array}
=
\begin{array}{c}
\text{(diagram)}
\end{array}
. \tag{40}
$$

The solution for $v'^x_y$ in Eq. (40) is found by inverting $R$. Using the bond projection, the inversion can be computed exactly with moderate numerical effort $\mathcal{O}(d^3 D^6)$. The full $M$ tensor representation, on the other hand, leads to an unfeasible costs of $\mathcal{O}(D^{12})$ for the exact inversion, and $\mathcal{O}(D^8)$ employing approximation methods.

After obtaining the optimized subtensor $v'^x_y$, we next update $w'^{x+1}_y$ while keeping $v'^x_y$ fixed by forming the partial derivative of Eq. (39) with respect to $w'^{\dagger,x+1}_y$,

$$
\frac{\partial}{\partial w'^{\dagger}} d(\tilde{v}, \tilde{w}, v', w') \overset{!}{=} 0 \quad \Rightarrow \quad
\begin{array}{c}
\text{(diagram)}
\end{array}
=
\begin{array}{c}
\text{(diagram)}
\end{array}
. \tag{41}
$$

The solution for $w'^{x+1}_y$ is again computed by matrix inversion of $R$.

This alternation process is repeated until the subtensors $v'^x_y$ and $w'^{x+1}_y$ converge. Monitoring the cost function $d(\tilde{v}, \tilde{w}, v', w')$ after every iteration step $i$, the convergence is detected by means of a fidelity measure which, following Phien, Bengua, Tuan, Corboz and Orús [47], can be defined as

$$f_d = |d_{i+1} - d_i| / d_0 \,. \tag{42}$$

The alternating optimization is stopped in case $f_d$ drops below some small threshold $\epsilon_d = \mathcal{O}(10^{-10})$ while showing no sign of large fluctuations.

Equipped with the converged subtensors $v'^x_y$ and $w'^{x+1}_y$, the original iPEPS form is then restored,

$$M'^x_y \;=\; X^x_y \, v'^x_y \,, \qquad M'^{x+1}_y \;=\; w'^{x+1}_y \, Y^{x+1}_y \,, \tag{43}$$

so that we can apply the next Trotter gate and repeat the full update optimization.

### 3.5.4 Alternative approaches

By accounting for the entire many-body wavefunction of the infinite system, the full update provides an optimization scheme that is free from the systematic error plaguing the simple update. Only the CTM representation of the effective environment induces some approximate character to the algorithm. The high accuracy of the method, however, comes at the price of drastically enhanced numerical costs since the full effective environment, in principle, has to be calculated after the application of every single Trotter gate (i.e., typically thousands of times). The fast-full update [47], where one updates the effective environment and site tensors simultaneously, offers an immediate improvement to this problem. Another possibility is the cluster update [71, 72], a hybrid version of the simple and the full update, which takes into account an improved, yet not complete version of the effective environment. Also, we note that it may be possible to achieve improvements in accuracy when computing the environment by properly removing the short-range entanglement residing in loops. To this end, it may be fruitful to combine the CTM method with other entanglement filtering algorithms, such as the Loop-TNR algorithm [73], graph-independent local truncation [74], full environment truncation [75], or entanglement branching [76].

Besides imaginary time evolution based algorithms, gradient-based energy minimization algorithms have also been found to be useful [22, 77, 78]. In particular, an automatic differentiation (AD) approach can be applied to reduce the complexity of the implementation, as the evaluation of gradients involves a huge number of summation of tensor environments [77, 79–83], which always needs to be done iteratively in any case. The prescription is generic, and may therefore also be attractive when combining AD techniques with non-abelian iPEPS in the future.

### 3.5.5 Gauge fixing

A well-known technical fact in the context of MPS is that the gauge degree of freedom on the bond indices can be efficiently exploited to generate a canonical representation [84]. Through the correct gauge, the effective environment of a specific bond, or rather its tensor network representation, can be replaced by identity matrices, ensuring numerical precision and stability of the MPS framework. The success of this scheme is closely linked to the fact that the environmental tensor network of an MPS is separable, such that the left and right block can be gauged independently. In the case of PEPS and iPEPS, the environment no longer factorizes

into different blocks, due to the presence of loops in the tensor network. In other words, cutting the TN at a single bond does not yield a bipartition of the system (as in the case of MPS), and therefore no full canonical PEPS or iPEPS representation exists.

Nevertheless, it is still possible to exploit the gauge degree of freedom on the bonds to improve the stability of the algorithm. Inspired by the 1D gauging protocol, Lubasch, Cirac, and Bañuls [48] recently introduced a gauge-fixing prescription for finite PEPS calculations that was later adapted in the context of iPEPS by Ref. [47]. It yields a significantly better conditioned effective environment and thus strongly improves the stability of the tensor optimization during the full update.

The gauge protocol [48] starts from the effective environment in the bond projection (38) which, after symmetrization, is subject to an eigenvalue decomposition,

$$ \text{(44)} $$

During this process, we remove the contributions from small negative eigenvalues to restore the positivity of $E_{\text{full}}$. Next we independently apply a QR and LQ decomposition to the tensor $Z$,

$$ \text{(45)} $$

and insert two identities $LL^{-1}$ and $R^{-1}R$, into the left and right bond indices of the effective environment, respectively. This yields a renormalized pair of subtensors $\bar{v}_y^x$ and $\bar{w}_y^{x+1}$ and a modified environment $\bar{E}_{\text{full}}$:

$$ \text{(46)} $$

Moreover, one also has to apply the inverse $L^{-1}$, $R^{-1}$ to the subtensors $X_y^x$ and $Y_y^{x+1}$, respectively, so that the full $M$ tensors can be restored properly after the tensor update [c.f. Eq. (43)],

$$ \text{(47)} $$

# 4  Fermionic tensor networks

For the tensor network representations discussed so far, we implicitly restricted our discussion to bosonic quantum many-body models. However, some of the most challenging and

interesting open questions with respect to the physics of strongly correlated systems involve fermions. Especially in two dimensions, the *t-J* model, the Hubbard model, and its multi-band extensions continuously attract much attention, since they are believed to play an important role for understanding of high-Tc superconductivity and quantum criticality. Due to the lack of alternative approaches (QMC is particularly limited by the sign problem in this context), much hope is set on tensor network techniques to treat these complex fermionic models under controlled conditions.

TN representations can incorporate fermionic statistics in any spatial dimension, and several different approaches have been developed for its efficient implementation, being mathematically all equivalent [45, 46, 85–89]. The most useful point of view for practitioners is that taken by Corboz and Vidal [85], adapted to the iPEPS by Corboz, Orús, Bauer and Vidal [45]. It fully implements the fermionic exchange rules in terms of modifications to the tensor network diagrams. In the following, we briefly review the main ingredients for fermionic tensor networks, mostly following [45], although not with the same formal rigor, to keep the presentation compact. We refer to Sec. 4.4 for technical details on the fermionic iPEPS implementation in combination with non-abelian symmetries.

For simplicity, we focus on a lattice of spinless fermions with a local Hilbert space dimension $d = 2$ on every site (though everything can easily be generalized to fermions with $d > 2$ [45]). The fermionic statistic of this model is typically treated at the level of operators, specifically by the anticommutation relations of the fermionic annihilation and creation operators, $\hat{c}_j$ and $\hat{c}_j^\dagger$,

$$\{\hat{c}_j, \hat{c}_{j'}^\dagger\} = \delta_{jj'} \quad \{\hat{c}_j, \hat{c}_{j'}\} = 0. \tag{48}$$

In addition, one always imposes some fermionic ordering of the sites, such that a fully occupied state on the lattice containing $N$ sites can be expressed by means of second quantization using the vacuum state $|0_1\rangle|0_2\rangle \ldots |0_N\rangle$ and an ordered sequence of creation operators,

$$|1_1\rangle|1_2\rangle \ldots |1_N\rangle = \hat{c}_1^\dagger \hat{c}_2^\dagger \hat{c}_3^\dagger \ldots \hat{c}_N^\dagger |0_1\rangle|0_2\rangle \ldots |0_N\rangle. \tag{49}$$

Starting from the techniques discussed in the context of bosonic systems, how can we incorporate the fermionic statistic into the framework of tensor networks? One possibility is to employ a Jordan-Wigner transformation to represent the fermionic operators in terms of Pauli matrices. In this way, the fermionic operator $\hat{c}_j$ is expressed in terms of bosonic operators in a non-local form, which can be described by a so-called *Jordan-Wigner string* acting on all sites $j' < j$ that appear "earlier" in the fermionic order of Eq. (49) [90]. These strings can be treated efficiently in the MPS framework, where it is always possible to choose the fermionic order $j$ equivalent to the position of a site in the MPS chain mapping. However, it leads to severe complications in the context of PEPS, where two nearest-neighbor sites $r = (x, y)$ and $r' = (x + 1, y)$ on the lattice might appear far apart in terms of their fermionic order $j$ and $j'$ [45].

To retain the "locality" of the iPEPS algorithm as well, we here adopt a different approach for the treatment of fermionic statistic in the tensor network language. This formulation builds on two simple "fermionization" rules discussed below, that were pioneered in the context of fermionic MERA by Refs. [85] and [46], and later adapted to the PEPS and iPEPS framework [45].

## 4.1 Parity conservation

A Fermionic Hamiltonian typically preserves the *parity* of the particle number of the state it acts on, defined to be $p = 1$ or $-1$ for an even or an odd number of particles, respectively.

This $\mathbb{Z}_2$ parity symmetry enables us to define wavefunctions and operators in terms of a well-defined parity quantum number $p$, resulting in a block structure in the tensor network. In particular, every index of a tensor can be assigned a well-defined parity.

The first fermionization rule enforces parity conservation in a TN representation. To this end, all tensors have to be chosen to be parity preserving. Taking a generic element of some $M$ tensor as example, it means that

$$M^{[\sigma_y^x]}_{\alpha\beta\gamma\rho} = 0 \quad \text{if} \quad p(\alpha)p(\beta)p(\gamma)p(\rho)p(\sigma_y^x) = -1\,, \tag{50}$$

with $p(\alpha) \in \{-1, 1\}$ describing the parity of the state labeled by the index $\alpha$ [45]. This immediately has the consequence that operators changing the parity number of a state, such as $\hat{c}_j$ have to be encoded with an additional index (see below). Parity conservation does not directly capture the fermionic statistic. However, it is crucial in order to track the fermionic signs, since we are able to distinguish states containing an even or odd number of fermions.

## 4.2 Fermionic swap gates

The second fermionization rule of [85] incorporates the fermionic statistics into the tensor network formalism. It implies that each line crossing in the TN is replaced by a fermionic swap gate,

$$\hat{S}^{\alpha\beta}_{\beta'\alpha'} = \delta_{\alpha\beta'}\delta_{\beta\alpha'} S(\alpha,\beta) = \quad \raisebox{-1.2em}{\begin{array}{c}\alpha \quad \beta \\ \blacklozenge \\ \alpha' \quad \beta'\end{array}} \quad , \tag{51}$$

with $S(\alpha,\beta) = -1$ if $p(\alpha) = p(\beta) = -1$ and $S(\alpha,\beta) = 1$ otherwise.

Why do the swap gates mimic the anticommutation relations of the fermions? Each line of the TN diagrams corresponds to a fermionic degree of freedom representing either physical (site indices) or virtual particles (bond indices). Any line crossing then corresponds to a particle exchange [85]. The implication of such an exchange depends on the nature of the particles. In the case of bosons such a swap is a trivial operation without any consequence. In the context of other particles, such as fermions, the underlying particle statistic does yield non-trivial consequences. For instance, additional factors of $-1$ have to be multiplied to the tensor network when swapping two states with odd fermionic parity number. Thus, the fermionic statistic of any tensor network can be captured by adding swap gates of type (51) to the diagrammatic representation. As a prerequisite, one has to be able to read out the parity of every index in the TN (hence, the first rule).

We emphasize that the fermionization rules can be readily implemented into any standard bosonic TN algorithm *without* altering the leading numerical costs, since the swap gates can typically be absorbed into a single tensor [85]. All steps can be performed completely analogously. In our iPEPS implementation we were able to recycle most parts of our code for bosonic systems by simply adding swap gates at the appropriate lines.

## 4.3 Fermionic operators

Another prerequisite to capture the fermionic statistic in a TN representation relates to the proper definition of local fermionic operators. Consider a generic two-site operator $\hat{O}_{ij}$ acting on sites $i$ and $j$, with $j > i$ not necessarily labeling contiguous sites in terms of the imposed fermionic order. Applied to a generic wavefunction, the resulting TN diagram contains a number of fermionic swap gates (illustrated in detail for MPS and iPEPS below). The impact of these gates on the wavefunction can be interpreted as swapping the physical index of site $i$ such that it becomes contiguous to $j$ with respect to the fermionic order. But this alone does

not fully account for the fermionic statistics. In addition, the fermionic order of the local two-site Hilbert space generated by sites $i$ and $j$ has to be properly incorporated on the level of the operators, which leads to factors of $-1$ for some matrix elements.

While easily generalizable to arbitrary systems [45], we illustrate this briefly for the simple example of spinless fermions, where the operator is expanded in the two-site basis $|\sigma_i \sigma_j\rangle = (c_i^\dagger)^{\sigma_i}(c_j^\dagger)^{\sigma_j}|0_i 0_j\rangle$, with $\sigma_j \in \{0, 1\}$:

$$\hat{O} = \sum_{\substack{\sigma_i' \sigma_j' \\ \sigma_i \sigma_j}} O_{\sigma_i \sigma_j}^{\sigma_i' \sigma_j'}|\sigma_i \sigma_j\rangle\langle\sigma_i' \sigma_j'|. \tag{52}$$

The coefficients $O_{\sigma_i \sigma_j}^{\sigma_i' \sigma_j'}$ are given by

$$O_{\sigma_i \sigma_j}^{\sigma_i' \sigma_j'} = \langle\sigma_i \sigma_j|\hat{O}|\sigma_i' \sigma_j'\rangle = \langle 0_i 0_j|(\hat{c}_i)^{\sigma_i}(\hat{c}_j)^{\sigma_j}\hat{O}(\hat{c}_i^\dagger)^{\sigma_i'}(\hat{c}_j^\dagger)^{\sigma_j'}|0_i 0_j\rangle. \tag{53}$$

If the operator describes a pairing term, $\hat{O} = \hat{c}_i \hat{c}_j$, the only non-vanishing coefficient is

$$O_{0_i 0_j}^{1_i 1_j} = \langle 0_i 0_j|\hat{c}_i \hat{c}_j \hat{c}_i^\dagger \hat{c}_j^\dagger|0_i 0_j\rangle = -1. \tag{54}$$

A standard hopping term $\hat{O} = \hat{c}_i^\dagger \hat{c}_j$ also has only a single nonzero element,

$$O_{1_i 0_j}^{0_i 1_j} = \langle 0_{j'} 0_j|\hat{c}_i \hat{c}_i^\dagger \hat{c}_j \hat{c}_j^\dagger|0_i 0_j\rangle = 1. \tag{55}$$

We conclude this part with an additional comment on operators that change the parity of a state, such as $\hat{O} = \hat{c}_j$. The first fermionization rule restricts our TN description to parity preserving tensors, as defined in Eq. (50). Naively, this would imply that simple annihilation or creation operators could not be properly described by fermionic TNs, since their tensor representation does not conserve fermionic parity. However, any parity changing tensor can be represented by a parity conserving tensor just by adding an additional single-valued index $\delta$ with $p(\delta) = -1$ [45]. For instance, the diagrammatic form $\hat{c}_j$ is then given by

$$(\hat{c})_{\sigma_j,\delta}^{\sigma_j'} = \quad \begin{matrix} \sigma_j' \\ \hat{c}_j \;-\!\!-\!\!^\delta \\ \sigma_j \end{matrix} \quad, \tag{56}$$

where the red line indicates that $\delta$ only takes a single value, i.e., represents a singleton dimension in a rank-3 tensor. This representation ensures that the only nonzero element, $(\hat{c})_{0_j,\delta}^{1_j}$, now satisfies Eq. (50):

$$p(1_j)p(0_j)p(\delta) = (-1)(+1)(-1) = 1. \tag{57}$$

## 4.4 Fermionic PEPS implementation

To enter this discussion, we return to our finite-size PEPS example on a $3 \times 3$ square-lattice cluster used in the beginning of Sec. 3.1. Recall that each site is labeled according to its coordinate in space, $\mathbf{r} = (x, y)$, so that the local basis states are denoted by $|\sigma_y^x\rangle$. In addition, we now have to decide on a specific fermionic order and use an additional label $j$, running from 1 to 9, to enumerate all sites of the system, $|\sigma_{y,j}^x\rangle$ (the red color of the fermionic index acts as guide for the eyes). Thus, a specific state in the Fock space can be expressed as

$$|\sigma_{1,1}^1\rangle|\sigma_{2,2}^1\rangle...|\sigma_{3,9}^3\rangle = (\hat{c}_1^\dagger)^{\sigma_1^1}(\hat{c}_2^\dagger)^{\sigma_2^1}...(\hat{c}_9^\dagger)^{\sigma_3^3}|0_{1,1}^1\rangle|0_{1,2}^3\rangle...|0_{1,9}^3\rangle. \tag{58}$$

Diagrammatically, this ordering *always* corresponds to the order in which the open indices of the wavefunction $|\psi\rangle$ are drawn, and directly affects the specific appearance of the PEPS TN,

(59)

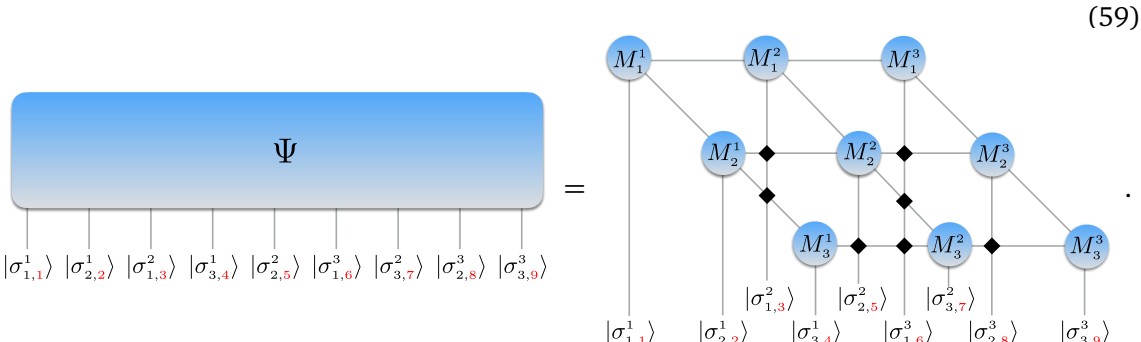

We emphasize that a different fermionic order automatically leads to a different diagrammatic representation, where the swap gates (black diamonds) potentially act on a different set of bonds. In this work, we only consider the fermionic 'zig-zag' order of Eq. (59) which (i) can also easily be applied to an infinite lattice system and (ii) enables us to recycle all bosonic iPEPS diagrams depicted in Sec. 3.1. For an explicit example of imposing another fermionic order, see Ref. [45].

After obtaining the proper diagrammatic form of the PEPS, all subsequent operations follow in complete analogy from the bosonic case. The only additional feature are the swap gates, which are put on every line crossing. For instance, an overlap calculation $\langle\psi|\psi\rangle$, derived in Eq. (14) for the bosonic PEPS by performing a number of jump moves, is carried out similarly for a fermionic system,

(60)

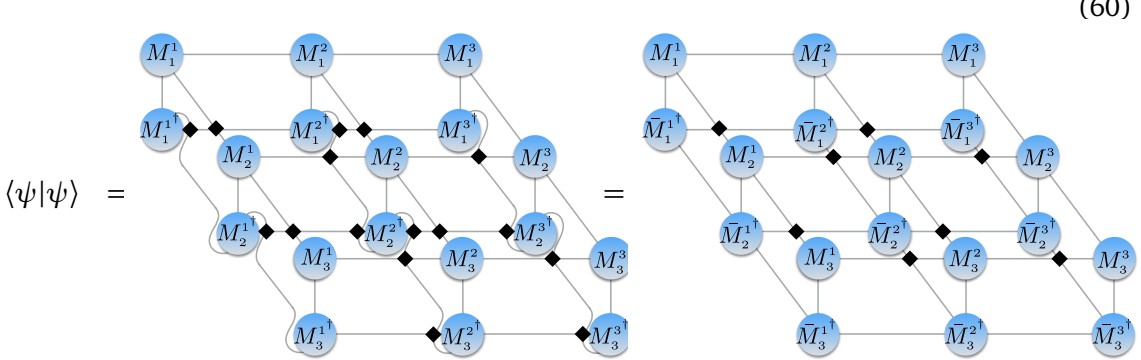

To reduce the complexity of the diagram, we again introduced a modified representation $\bar{M}^{x\dagger}_y$ of the conjugate tensors in the second step of Eq. (60). In contrast to the bosonic case, where $\bar{M}^{x\dagger}_y$ and $M^{x\dagger}_y$ are mathematically equivalent objects [see Eq. (13)], we emphasize that $\bar{M}^{x\dagger}_y$ here includes two fermionic swap gates that are absorbed into the tensor, according to

$$\blacklozenge\, M^{x\dagger}_y\, \blacklozenge \;=\; \bar{M}^{x\dagger}_y \quad. \tag{61}$$

## 4.5 Fermionic iPEPS implementation

Considering fermions in an infinite lattice system, the protocol of imposing a zig-zag fermionic order on the lattice can be adopted in a very straightforward manner [45]. In hindsight, we already implied this kind of ordering when drawing the iPEPS diagrams in Sec. 3.1. The

extensions from the bosonic to the fermionic case is easily achieved by the presence of the fermionic swap gates at line crossings.

In most iPEPS applications, the modified definition of the conjugate tensor $\bar{M}_y^{x\dagger}$, (61), and the fermionic version of the reduced tensor $m_y^x$

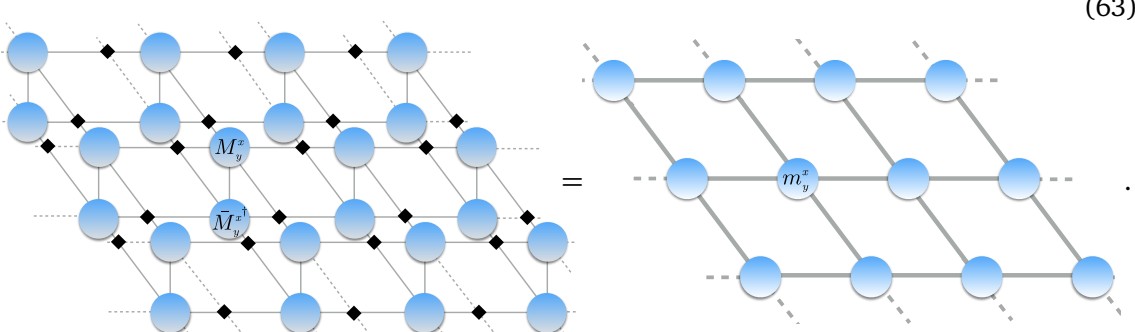

$$(62)$$

simplify the algorithm by a great deal. For instance, the calculation of an overlap $\langle\psi|\psi\rangle$ can even be represented diagrammatically without any swap gates present,

$$(63)$$

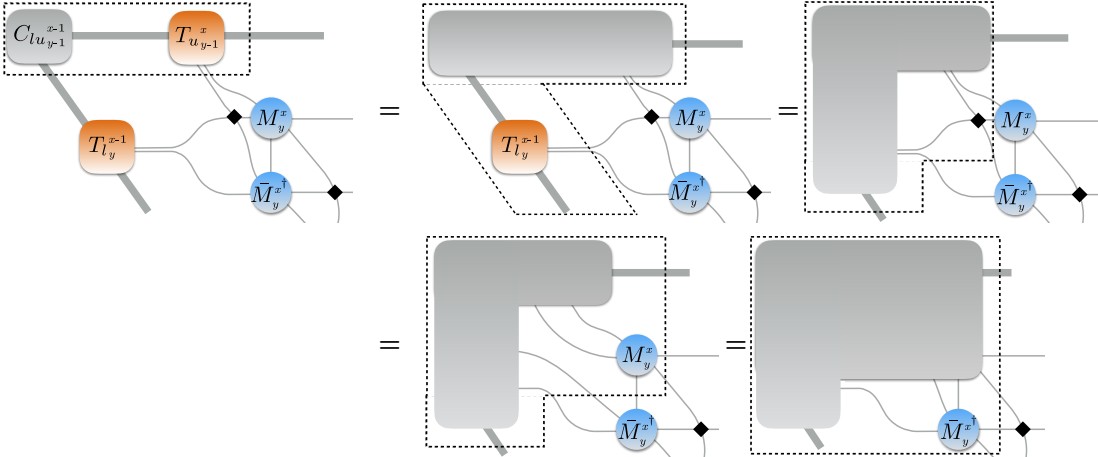

In principle, this would also enable us to carry out the coarse graining steps in the CTM calculation exactly in the same way as in bosonic iPEPS in terms of the reduced $m$ tensors. To perform the algorithm with an efficient cost scaling, however, the $M$ tensors and their conjugates have to be kept separated [see Sec. 3.3]. This typically leads to the presence of four additional swap gates for each site (only two when using $\bar{M}_y^{x\dagger}$).

The strategy of incorporating the swap gates appearing in a TN is to absorb them into one single tensor [85]. Depending on the TN, this is not always possible in the very first contraction step. Nevertheless, every swap gate can typically be absorbed at some intermediate contraction step. We illustrate this procedure for the contraction of parts of the CTM environment,



$$= \qquad . \tag{64}$$

Swap gates also appear in the context of tensor optimization and the evaluation of a two-site operator, such as,

$$\langle \psi | \hat{O} | \psi \rangle = \qquad . \tag{65}$$

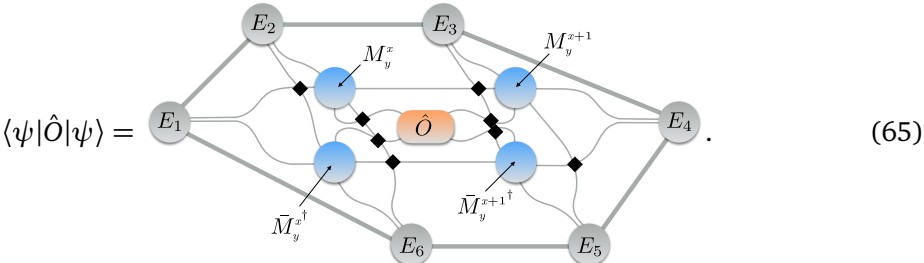

We conclude this section by pointing out the modifications to the full-update protocol in the context of fermions. Again, most of the steps are exactly the same as in the bosonic version of the algorithm. In particular, the actual tensor optimization does not contain any swap gates due to the absence of line crossings in Eq. (39). However, the initialization slightly differs since one has to account for the presence of swap gates when performing the bond projection,

$$\tag{66}$$

Importantly, the swap gate acts differently on the conjugate tensors, so that the conjugate subtensors have to be generated by two independent SVD or QR decompositions,

$$\tag{67}$$

The tensor network representation of the effective environment also contains an additional set

of swap gates,

$$E_{\text{full}} = \qquad\qquad\qquad\qquad .$$

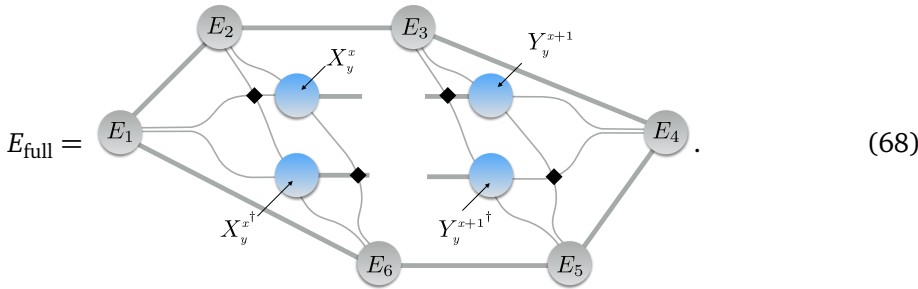

(68)

Whereas the tensor optimization does not differ from the bosonic formulation, the restoration of the actual iPEPS representation after the update works in a slightly modified way,

$$\cdots \boxed{X^x_y}\,\boxed{v'^x_y} = \cdots \boxed{\tilde M'^x_y} = \cdots \boxed{M'^x_y} \cdots .$$

(69)

Compared to the bosonic case in Eq. (43), we have to account for the additional swap gate.

## 5 Implementation of symmetries

The exploitation of symmetries, where available, is very important for writing efficient tensor network codes. In this Section we address various aspects of this issue.

### 5.1 Abelian symmetries

For a lattice model with abelian symmetries, quantum states can be labeled $|ql\rangle$, where $q$ is an abelian "charge" quantum number, and $l$ distinguishes different states with the same charge. Consider the simplest non-trivial example of a rank-3 tensor $A$, which fuses the tensor product of two elementary state spaces with abelian symmetry, $|q'm\rangle$ and $|q''n\rangle$, into the combined tensor product space $|ql\rangle$. This operation can be expressed as

$$|ql\rangle = \sum_{q'l'}\sum_{q''l''} |q'l'\rangle|q''l''\rangle \, (A^q_{q'q''})^l_{l'l''}.$$

(70)

To reflect the system's abelian symmetry, the $A$ tensor carries a $q$-label for the symmetry sector of each of the indices $l$, $l'$ and $l''$. From a numerical perspective this introduces additional bookkeeping effort. At the same time, symmetry-specific selection rules enforce a large number of elements of $A$ to be exactly zero [for the example of U(1) particle conservation, the selection rule takes the form $q = q' + q''$]. Keeping only the nonzero elements leads to sparse tensor structures and, hence, results in significant computational speed-up and reduced memory requirements.

### 5.2 Non-abelian symmetries

Let us now consider the same example in the context of non-abelian symmetries. Then quantum states can be organized into irreducible symmetry multiplets (irreps) that carry an additional label $q_z$ that specifies the internal structure of an individual multiplet, e.g. $|ql\rangle \rightarrow |ql;q_z\rangle$. The decomposition of a direct product of two irreps into a direct sum of irreps is fully defined by the Clebsch-Gordan coefficients (CGCs) of the symmetries present. In this description, the

coefficients of the *A* tensor in Eq. (70) factorize into tensor products of reduced matrix elements and CGCs, so that Eq. (70) generalizes to

$$|ql;q_z\rangle = \sum_{q'l'q'_z}\sum_{q''l''q''_z}|q'l';q'_z\rangle|q''l'';q''_z\rangle \cdot \left\|A^q_{q'q''}\right\|^l_{l'l''} \cdot \left(C^q_{q'q''}\right)^{q_z}_{q'_zq''_z}. \tag{71}$$

Here $\left(C^q_{q'q''}\right)^{q_z}_{q'_zq''_z} \equiv \left\langle qq';q_zq'_z\,\middle|\,q'';q''_z\right\rangle$ represent CGCs, and $\left\|A^q_{q'q''}\right\|^l_{l'l''}$ denote reduced matrix elements of the basis transformation [30]. This allows one to *compress* the nonzero data blocks of the tensors, further reducing the numerical requirements, yet at the price of a significantly increased bookkeeping effort.

The same structure as in Eq. (71) also carries over to the coefficients of arbitrary operators $\hat{O}_{q'q'_z}$ that acts in a given (local) state space $|ql;q_z\rangle$, where the latter itself is already properly organized w.r.t. given symmetries. Clearly, if one wants to exploit symmetries in numerical simulations, these symmetries must be well-defined throughout at every step and, in particular, for each individual tensorial object under consideration. Hence one also needs to know how operators transform under given symmetries. That is, all operators can be reduced to or built from irreducible tensor operators (irrops). These elementary objects consist of a set of operators (like a spinor) that under symmetry operation are transformed into each other completely analogously to the states of a particular irreducible multiplet $q'$, in which case $q'_z$ labels the individual operators in the set. The intimate relation to states becomes apparent when the irrop acts on a scalar state $|0\rangle$, i.e., a singlet in all symmetries having $q=0$ like a vacuum state. Then $\hat{O}_{q'q'_z}|0\rangle \equiv |q'q'_z\rangle$ associates an irrop with an irrep, up to normalization and assuming the state is not destroyed. Both of them transform according to the irrep $q'$. Generally then, a *particular* irrop with multiplet index $l'(=1)$, can be expressed in a factorized form exploiting the Wigner-Eckart theorem,

$$\left\langle ql;q_z\middle|\hat{O}_{q'l';q'_z}\middle|q''l'';q''_z\right\rangle \equiv \left\langle ql;q_z\middle| \cdot \left(\hat{O}_{q'l';q'_z}\middle|q''l'';q''_z\right\rangle\right) = \left\|O^q_{q'q''}\right\|^l_{l'l''} \cdot (C^q_{q'q''})^{q_z}_{q'_zq''_z}, \tag{72}$$

with CGCs $C^q_{q'q''}$ and reduced matrix elements $\|O^q_{q'q''}\|^l_{l'l''}$. The latter describe transitions between multiplets $ql$ and $q''n$ within a given Hilbert space induced by the irrop $\hat{O}_{q'l'}$.

The conceptual structure of the tensor describing a basis transformation or operator matrix elements is thus the same. With focus on the tensor alone, i.e., skipping the ket states contracted with the tensor in Eq. (71), the tensor itself may be written more compactly in the generic form [31],

$$A = \bigoplus_q \|A\|_q \otimes C_q\,, \tag{73}$$

where $q$ now is the full collection of symmetry labels for all indices (legs) in a particular block realization. For example for the cases above, $q \leftarrow (q',q'';q)$ where, by convention, e.g., subscript indices are grouped and listed before superscript indices. This demonstrates that each tensor acquires a block structure (collected via the outer sum), and that for each such block, Clebsch-Gordan tensors are split off in a tensor-product structure. The tensor product involves a reduced matrix element tensor (RMT) and a corresponding generalized Clebsch-Gordon coefficient tensor (CGT) with the same tensor rank. This *reduces* the actual number of freely choosable matrix elements, and thus the effective dimensionality of the tensor, $A \to \|A\|_q$, e.g., going from $D$ states on a given index (leg) to $D^* \leq D$ multiplets. For abelian symmetries there is no reduction, $D^* = D$, whereas for $SU(N)$, one empirically finds an effective average dimensional reduction of $D^* \sim D/3^{N-1} \ll D$.

The conceptual framework described above forms the basis for the QSpace tensor library [30, 31] for building many-body state spaces in the presence of symmetries [Eq. (71)] and for

describing the actions of operators therein [Eq. (72)]. It allows one to construct a tensor network and its constituent tensors step by step in an iterative fashion. For a tutorial illustration of its underlying ideas, see App. A.

## 5.3 Outer multiplicity

When dealing with non-abelian symmetries, one generically also encounters outer multiplicity, i.e. direct sums in which the same irrep or irrop (or the same combination of several of them) occurs more than once. Consider, for example, an SU(2) rank-4 CGT having two incoming and two outgoing legs with symmetry labels $(S_1, S_2)$ and $(S_1', S_2')$, respectively. Then, there are several different possibilities to fuse $(S_1, S_2)$ to an intermediate irrep $\bar{S}$ and to subsequently split the latter into $(S_1', S_2')$:

$$
\begin{array}{c}
S_1 \searrow \qquad \nearrow S_1' \\
\qquad \bigcirc \qquad \\
S_2 \nearrow \qquad \searrow S_2'
\end{array}
=
\sum_{\oplus \bar{S}}
\begin{array}{c}
S_1 \searrow \quad \bar{S} \quad \nearrow S_1' \\
\qquad \bullet \!\!-\!\! \bullet \qquad \\
S_2 \nearrow \qquad \searrow S_2'
\end{array} .
\tag{74}
$$

In this sense, the *outer multiplicity* (OM) of the rank-4 CGT on the left is larger than one. Each of the terms in the direct sum on the right corresponds to an *independent* CGT within a set of *orthognal* CGTs $C_q^\mu$, all carrying the same external symmetry labels $q \equiv (S_1, S_2; S_1', S_2')$, but distinguished by an outer multiplicity label $\mu$ (here given by $\bar{S}$). For example, if $S_1 = S_2 = S_1' = S_2' = S$, then the outer multiplicity label $\mu = \bar{S}$ can take the values $0, 1, \ldots, 2S$. Since the outer multiplicity label is being summed over on the right, it is no longer *visible* at the level of the rank-4 CGT on the left.

SU(2) CGTs generically have OM larger than 1 once their rank is $r \geq 4$. For general non-abelian symmetries such as SU($N \geq 3$), OM larger than 1 already also occurs at the level of rank-3 CGTs, e.g., in the standard state space decomposition as in Eq. (71). There, the same $q$ on the l.h.s. can arise in several different ways, which needs to be distinguished through an outer multiplicity index $\mu$:

$$
|q(l\mu); q_z\rangle = \sum_{q'l'q_z'} \sum_{q''l''q_z''} |q'l'; q_z'\rangle |q''l''; q_z''\rangle \cdot \big\| A_{q'q''}^q \big\|_{l'l''}^{l\mu} \cdot \big( C_{q'q''}^{q\mu} \big)_{q_z'q_z''}^{q_z} ,
\tag{75}
$$

where $\tilde{l} \equiv (l\mu)$ just labels the overall mutiplets on the l.h.s., whereas the multiplicity index $\mu$ on the r.h.s. constitutes an additional dimension of the RMT $\|A\|$ within its particular symmetry sector tied to the CGT $C_{q'q''}^{q\mu}$. In the presence of OM, the tensor representation in Eq. (73) generalizes to

$$
A = \bigoplus_q \Big[ \sum_\mu \|A\|_q^\mu \otimes C_q^\mu \Big] ,
\tag{76}
$$

with a *regular* summation over the multiplicity index $\mu$ here. OM evidently also increases the effective dimension of the reduced matrix element tensors $\|A\|_{q\mu}$. In general, OM needs to be properly accounted for (once and for all) at the level of rank-3 CGTs [30]. Moreover, to ensure overall consistency, OM needs to be tracked meticulously not only when performing direct product decompositions into direct sums, but also when performing (iterative pairwise) contractions of tensors [31].

## 5.4 PEPS with symmetries

Building on the fusion rules for different state spaces in Eq. (71), one can generate symmetric tensor networks consisting of higher-rank tensors. This can be easily understood from the perspective of contracting multiple $A$ tensors to some larger-ranked object. The resulting tensor

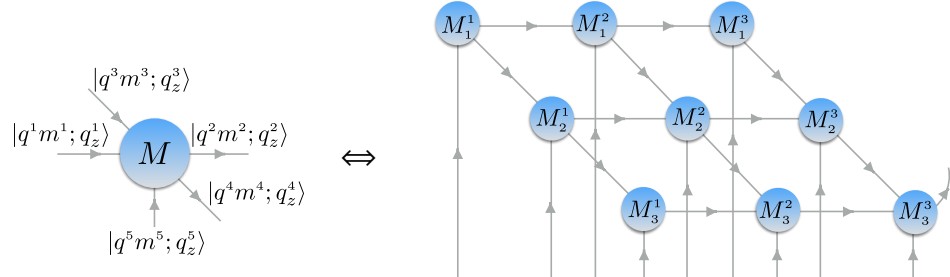

Figure 4: Schematic construction of a PEPS tensor network state. The elementary tensor $M$ associated with each site (left panel) is tiled in a translational invariant fashion into a PEPS (right panel). The index order of its five legs is arbitrary but fixed. Here we use the order $(l, r, t, b, \sigma) \equiv (1, 2, 3, 4, 5)$ for left, right, top, bottom, and local state spaces, respectively. When exploiting symmetries, every individual index (i.e., leg of a tensor or line) represents a state space that must be expressed in terms of symmetry subspaces, throughout. For non-abelian symmetries, a given index describes a state space $s$ that is organized as $|s\rangle \equiv |qn; q_z\rangle$, where $q$ specifies a symmetry sector, $n$ a specific multiplet within the symmetry sector $q$, whereas $q_z$ indexes the internal multiplet structure which can be split off as a tensor product with a generalized CGTs [30].

then represents a tensor product of several state spaces. Setting up a symmetric PEPS tensor network, for example, follows exactly this pattern, leading to the diagrammatic representations in Fig. (4) for a single tensor (left) and a contraction of several such tensors (right): The symmetrized $M$ tensors contain additional arrows on the index lines to indicate which state spaces are incoming and outgoing (i.e., which (group of) state spaces are fused into which, according to Eq. (71)). We have some freedom in fixing the direction of these arrows and some choices might be more convenient to implement than others. Note that the extra index of $M_3^3$ determines the global symmetry state of a specific PEPS representation. Of course, the symmetric PEPS also guarantees that the corresponding quantum state is symmetric, i.e., forms a well-defined symmetry multiplet.

Symmetry-induced selection rules cause a large number of matrix elements to be exactly zero, thus bringing the Hamiltonian into a block-diagonal structure and subdividing tensors into well-defined symmetry sectors. Keeping only the nonzero elements, we can achieve tremendous improvement in speed and accuracy in numerical simulations by the incorporation of symmetries. In the context of non-abelian symmetries, the nonzero data blocks are not independent of each other and can be further compressed using reduced matrix elements together with the Clebsch-Gordan algebra for multiplet spaces.

The special ingredient of our fermionic iPEPS implementation, that sets our work apart from that of other iPEPS practitioners, concerns the explicit incorporation of non-abelian symmetries, such as $SU(2)_{\text{spin}} \otimes SU(N)_{\text{orb}}$ with the fermionic $\mathbb{Z}_2$ parity symmetry in the particle sector. The non-abelian symmetries are fully encoded in the QSpace [30] tensor library, which automatically handles the symmetry-induced fusion rules of both the reduced matrix elements and the Clebsch-Gordan space.

Non-abelian iPEPS was pioneered by Liu, Li, Weichselbaum, von Delft and Su [37] for the case of the spin-1 Kagome Heisenberg antiferromagnet, which illustrated an $SU(2)_{\text{spin}}$ symmetric iPEPS representation in terms of a "projection" picture. Following ideas of $SU(2)$ invariant iPEPS representations for the spin-$\frac{1}{2}$ resonating valence-bond state [91, 92] and the spin-1 resonating AKLT state [93], the symmetric iPEPS tensors can be understood as emerging from

sets of "virtual particles" associated with each site that are pairwise maximally entangled along each virtual bond with their nearest neighbor sites, and then projected into the local degrees of freedom of the corresponding site. Starting from such an SU(2) invariant iPEPS, eventually one only specifies the effective bond dimension $D^*$, and lets the tensor optimization dynamically determine the relevant symmetry sectors on each bond. The number of multiplets $D^*$ translates into a significantly larger actual number of states, $D$, associated with each bond (note that $D$ may vary for the same $D^*$ depending on the actual multiplets being used). In practice, the maximal feasible values for $D^*$ correspond to retaining an actual number of states $D$ which typically lies out of reach of standard iPEPS calculations incorporating abelian symmetries only.

## 5.5 Technicalities

In the remainder of this section we briefly point out some important technicalities when implementing non-abelian iPEPS.

### 5.5.1 Global symmetry sector

Ref. [37] states that the projection picture is dense, in the sense that it can cover the full Hilbert space and generate any symmetry eigenstate. Whereas this is true for finite-size PEPS, we emphasize that for translational invariant systems where the iPEPS is tiled with the *same* $M$ tensor, by construction, there cannot be a "drift" in average value of a quantum number along any line of $M$ tensors. In the case of non-abelian symmetries this implies that the global symmetry label of the iPEPS is *always* constrained to the singlet sector. This is conceptually similar to the case of U(1) symmetries in iPEPS, where states are restricted to a global symmetry sector corresponding to the quantum number zero, i.e., $q = 0$ (see Ref. [94], referred as 'identity charge' therein).

We note, however, that for abelian U(1) symmetries such as charge, any local filling can be realized based on the simple observation that U(1) symmetry labels are additive. Hence one is free to shift them locally and scale them globally at will. Specifically, one may shift the charge labels associated with the local state space of each site relative to the targeted mean local occupation $\bar{q}$, i.e., $q \to q - \bar{q}$. By this simple relabeling trick, average charges associated with the virtual bonds can fluctuate around $q = 0$. For non-abelian symmetries, however, such a relabeling scheme appears ill-suited, so that, by construction, our iPEPS implementation represents a global singlet. For our results below at finite doping, we still also only use $\mathbb{Z}_2$ charge parity even though charge itself is conserved.

### 5.5.2 Arrow convention

When exploiting symmetries, every index represents a state space with a particular symmetry multiplet. Now when fusing state spaces across tensors, this naturally introduces the concept of state spaces that 'enter' a given tensor, and state spaces that 'emerge' from it. For tensors this implies in a graphical depiction that one has to distinguish *ingoing* and *outgoing legs*, i.e., every leg acquires a direction, specified by an arrow [e.g., see Fig. 4]. Mathematically, this is equivalent to distinguishing between co- and contravariant indices (a notational convention not used here) [31]. The action of raising or lowering indices then corresponds to reverting arrows, as schematically depicted in Fig. 5. This is an operation that represents gauge-transformations of tensor network states, leaving the physical properties of the individual states unaffected [95]. Importantly, within a tensor network state, a summed over, i.e., *contracted* index connecting a pair of tensors, where it is outgoing from one tensor, and incoming to the other.

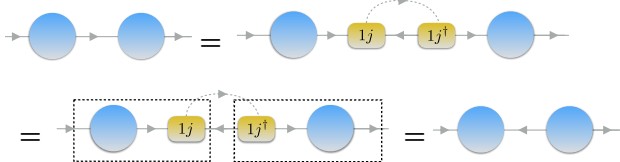

Figure 5: Arrow inversion. An identity, $\mathbb{I} = U^{\dagger}U$, is inserted on a bond (here the center bond) where up to normalize the unitary $U$ represents a $1j$ symbol [31], i.e., a (degenerate) rank-3 tensor which combines two state spaces, $q$ and its dual $\bar{q}$ into a scalar singlet state. Upon absorbing $U$ and $U^{\dagger}$ on opposite sides with the neighboring tensors, effectively, the arrow on the center bond has been reverted. The singlet index (dashed line) can be omitted in the end.

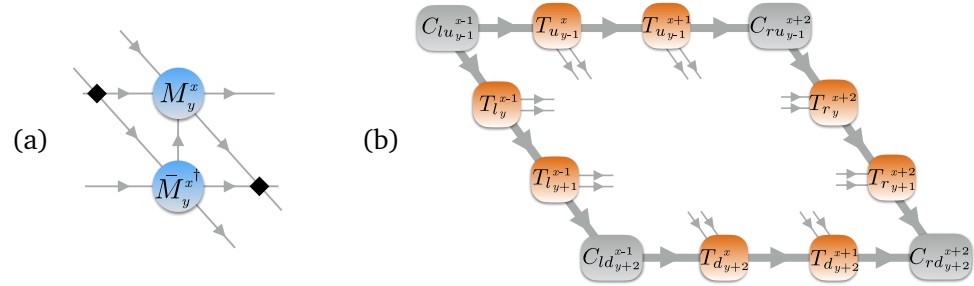

Figure 6: Arrow convention for $M$ tensor (panel a) as they enter inside the corner transfer matrix (CTM) setup in (panel b). The latter combines bra and ket state as required for the minimization of the total energy $E = \langle \psi | H | \psi \rangle$ when truncating [58]. From the perspective of an individual site, this "double layer" tensor network translates into $\langle M | \ldots | M \rangle$. For this, note that we have reverted the bond indices of the 'bra-tensor' $M \rightarrow \bar{M}$ such that they point in the *same* direction as the corresponding indices of $M$. Only then one can fuse the 'double bond index' into a single *fat* index. This greatly simplifies many fusion steps during the CTM procedure. The black diamonds in (a) indicate fermionic swap gates [45, 85].

When setting up a symmetric iPEPS representation, we therefore have to choose an "arrow convention" for all iPEPS tensors. On a square lattice, when a single $M$ tensor with four virtual bond indices tiles an entire 2D iPEPS, this necessarily implies that two virtual bond indices must be ingoing and two outgoing [cf. Fig. 4(b)].

For compactness and readability of the code, we want to minimize the number of steps in the algorithm that involve reverting arrows as in Fig. 5. To this end, we establish the arrow convention for $M$ tensors as well as the corner transfer matrices as shown in Fig. 6. Thus the quantum labels on all virtual bonds always "flow" from the upper left to the lower right corner of the tensor network.

### 5.5.3 Efficient contractions

The standard procedure when contracting tensors *in the absence of any symmetries* is to reshape a contraction into an effective matrix product [96] where efficient libraries can be utilized. That is, for any tensor in a contraction, the indices that are contracted as well as the ones that are kept, are grouped, i.e., permuted into order, and then fused into hyperindices.

This strategy also carries over when implementing symmetries, abelian and non-abelian alike. In principle, one has the option of matching symmetry sectors first, and then do the

contractions for every match in the above spirit. However, for abelian and non-abelian symmetries alike, this would cause a significant proliferation of symmetry sectors with increasing rank already for an individual tensor, yet also for matching symmetry combinations when contracting a pair of tensors. Roughly, if there are on average $m$ symmetry sectors associated with each of the $r$ legs of a given rank-$r$ tensor, one may expect up to $m^r$ possible symmetry combinations. The situation is worse still for non-abelian symmetries, where the tensor products of two multiplets can give rise to many different multiplets. Therefore a computation of a contraction is slowed down by (exponentially) many combinations with increasing rank of the involved tensors. Yet the individual contractions of matching symmetry sectors often involve only small effective block matrices. As a consequence, the above strategy becomes prohibitively inefficient strongly with increasing rank of the tensors. For an efficient way to proceed, one therefore *first* needs to merge indices into hyperindices (respecting fusion rules in the presence of non-abelian symmetries), and then do the contraction.

An efficient non-abelian iPEPS implementation therefore must fuse indices in contractions prior to the actual contraction, while being aware that only legs that point in the same direction can be fused [e.g. see Fig. 6]. After the contraction, the remaining open indices must be given back their original structure. In the presence of non-abelian symmetries, the fusion is effectively taken care of by an additional contraction with unitary tensors, which need to be reapplied on the open indices. This is an extra layer of complication that concerns each and every contraction that involve tensors with rank $r \gtrsim 4$.

To be specific, we consider the the two-band Hubbard model (discussed in Sec. 6.1 below) with $\mathbb{Z}_2 \otimes SU(2)_{\text{spin}} \otimes SU(2)_{\text{orb}}$, retaining $D^* = 6$ multiplets on each bond. Already the $M$ tensors of rank 5 are complicated objects. However, the numerically most demanding tensors appear during the CTM coarse graining. Here, we typically have to deal with rank-6 and rank-7 tensors, and it depends strongly on the implementation details whether the CTM procedure is still feasible. Let us focus on a typical rank-6 tensor appearing several times in a CTM step, obtained by contracting the following TN diagram,

$$
\begin{array}{c} \text{(diagram)} \end{array} = \begin{array}{c} \text{(diagram)} \end{array} . \tag{77}
$$

Each thin line corresponds to a single-layer bond index of dimension $D^*$, while the thick lines are environmental bond indices of dimension $\chi^* = 80$. The resulting tensor on the r.h.s of Eq. (77) requires only 390 MB of memory for the reduced matrix elements, as compared to an estimated 883 GB without symmetries. This highlights the efficiency of the non-abelian symmetries, where here we gain more than factor of 2,000 only in terms of storage requirement! At the same time, its QSpace consists of about $430,000(!)$ individual symmetry blocks. Numerically, this number corresponds to $(0.61\,\chi^*)^2(0.61\,D^*)^4$, in agreement with an expected exponential proliferation of symmetry blocks with increasing rank. The sizes of the symmetry blocks, of course, are comparatively small, on average containing only $100 = 10^2$ individual coefficients.

To reduce the rank of this tensor, it is possible to fuse the three indices pointing to the left and to the bottom, respectively. This yields the rank-2 matrix representation,

$$
\begin{array}{c} \text{(diagram)} \end{array} = \begin{array}{c} \text{(diagram)} \end{array} , \tag{78}
$$

with size $28,000 \times 28,000$ on the multiplet level. Being a rank-2 object, it must be block-diagonal. The matrix only contains 37 symmetry blocks of larger size (on average, each block consists of $750^2$ coefficients). Remarkably, the reduced matrix elements of the latter matrix require *slightly less* memory (350 MB) than those of the original rank-6 tensor. To a very minor extent, this may be attributed to overhead for organizing the long lists of symmetry blocks in the tensor. More importantly, the rank-6 tensor has significant outer multiplicity [30, 31], which is absent in the rank-2 tensor. Most importantly, however, this simple comparison strongly suggests that the symmetry blocks in the rank-2 matrix representation are densely populated by the entries in the rank-6 tensor.

Now how do the two different representations perform in terms of contraction speed? To compare them, we consider the next step of the CTM scheme, which requires forming the upper part of the environment, by contraction the following tensor network, both in the rank-6 and rank-2 representation

$$\Longleftrightarrow \qquad . \tag{79}$$

The speed of the contraction vastly differs. Contracting two rank-2 objects results in 37 contractions of the block-diagonal rank-2 objects, which is performed with QSpace [30] in about one second of CPU time. In contrast, we had to terminate the contraction of the rank-6 tensors after four hours (!) of calculation time. In the latter case, $10^9$ individual contractions are allowed by symmetry. Although the effort for each of these contractions is minimal, having to process their vast number step by step leads to a significant overhead, and thus to a drastic decrease in numerical efficiency.

# 6 Examples

Our main goal here is to illustrate the potential of non-abelian iPEPS, discussing both the benefits and limitations of exploiting non-abelian symmetries, by showing exemplary results for symmetric two and three band Hubbard models. A full analysis of the intricate physics of each of these systems goes beyond the scope of this work and is left for future studies.

Whereas the one-band Hubbard model already features important aspects of strongly correlated materials, such as the Mott insulator transition or the emergence of $d$-wave super-conducting pairing, for a multi-band Hubbard model a number of fascinating phenomena emerge from the interplay of different electron orbitals which cannot be captured by an effective model with a single band. Both intra-atomic Coulomb exchange or the presence of crystal field splitting can give rise to a number of intriguing effects, such as the existence of an orbital-selective Mott insulating phase, where only one orbital becomes insulating while the other retains its metallic properties [97–101]. In order to understand this physics from a theoretical perspective, it is clearly necessary to go beyond a single-band system and study multi-band generalizations of the Hubbard model.

In addition to perspectives in strongly correlated materials, multi-band high-symmetry models, such as SU($N$) Hubbard models or related Heisenberg models give rise to fascinating new types of quantum states including exotic magnetically ordered phases. These are not only of general academic interest but recently have also become experimentally accessible in the context of cold atoms [102, 103].

The exponentially large quantum many-body Hilbert space and the ensuing strong electronic correlations pose an extreme challenge to numerical approaches. Besides, one also

has to deal with an enlarged parameter space that substantially adds to the complexity of these systems. For instance, the spinful symmetric two-band Hubbard model with only on-site interactions already contains additional parameters such as Hund's interaction energy in comparison to its single-band version. Therefore, wide regions of the phase diagram of these models remain blank and there is a compelling need for developing numerical methods that can reliably deal with such systems in an unbiased way.

## 6.1 Spinful two-band Hubbard model

In this section, we demonstrate that fermionic iPEPS enhanced with non-abelian symmetries is a valuable ansatz to deal with symmetric complex multi-band systems in 2D. As a first example, we consider the repulsive Hubbard model with $M = 2$ bands and spin and orbital degeneracy on the square lattice. Specifically, we consider the Hamiltonian [104],

$$\hat{H} = \sum_{\langle ij \rangle} \sum_{m\sigma} \left( -t \hat{c}^{\dagger}_{im\sigma} \hat{c}_{jm\sigma} + \text{H.c.} \right) + \frac{U}{2} \sum_i \hat{n}_i (\hat{n}_i - 1) \tag{80a}$$

$$\hat{H}_{\mu} = \hat{H} - \underbrace{(\mu + \frac{3U}{2})}_{\equiv \mu_0} \sum_i \hat{n}_i, \tag{80b}$$

with hopping amplitude $t$ between nearest-neighbor sites $\langle ij \rangle$, spin index $\sigma \in \{\uparrow, \downarrow\}$, orbital index $m = 1, \ldots, M$, and site occupation $\hat{n}_i \equiv \sum_{m\sigma} \hat{n}_{im\sigma}$. We take $t := 1$ as unit of energy, throughout. We tune the average occupation via the chemical potential $\mu$ in Eq. (80b). But when computing the ground state energies, we compute the expectation values of the Hamiltonian in Eq. (80a), otherwise. The chemical potential in Eq. (80b) was offset by $\mu_0$ such that $\mu = 0$ corresponds to half-filling in the presence of a finite onsite Coulomb energy $U$. Overall, the Hamiltonian in (80) features both an $\text{SU}(2)_{\text{spin}}$ and $\text{SU}(2)_{\text{orbital}}$ symmetry, which we exploit in our iPEPS implementation. We ignore local Hund's coupling. Therefore spin and orbital index become interchangeable, resulting in 4 equivalent flavors. Overall, this actually leads to an enlarged $\text{SU}(4)$ symmetry of 4 spinless flavors (not exploited here). Also, we exploit only charge *parity* conservation rather than U(1) charge, and tune the filling via a chemical potential. The reason for this is partly technical, in that by being interested in finite doping we do not necessarily have integer filling in our unit cell. As a benefit, by just tracking charge parity, this immediately also permits the study of superconducting correlations.

For the ground state of a given average filling $n = n(\mu)$, set via Eq. (80b), we define the ground state energy per site $e_0$, the bond energy $e_0^{ij}$, and the generalized spin-singlet pairing amplitude $\Delta^{ij}$ as the expectation values

$$e_0 \equiv \frac{1}{N} \langle \hat{H} \rangle \tag{81a}$$

$$e_0^{ij} \equiv \left\langle -t \sum_{m\sigma} \left( \hat{c}^{\dagger}_{im\sigma} \hat{c}_{jm\sigma} + \text{H.c.} \right) + \frac{U}{8} \left[ \hat{n}_i (\hat{n}_i - 1) + \hat{n}_j (\hat{n}_j - 1) \right] \right\rangle \tag{81b}$$

$$\Delta^{ij} \equiv \frac{1}{\sqrt{2}} \sum_m \left\langle \hat{c}_{im\uparrow} \hat{c}_{jm\downarrow} - \hat{c}_{im\downarrow} \hat{c}_{jm\uparrow} \right\rangle, \tag{81c}$$

with $N$ the (fictitious total) number of sites. Here the 'bond energy' includes the Coulomb interaction energy $U/2$ of each of its associated pair of sites, weighted by $1/z$ with $z = 4$ the coordination number on the square lattice. Therefore, the average bond energy of all nearest beighbor bonds, $\overline{e_0^{ij}} = \frac{1}{4} \sum_{j \in [\text{n.n. of } i]} e^{ij}$, is related to the average energy per site, $e_0$, by $\overline{e_0^{ij}} = \frac{e_0}{2}$, since on average there are two bonds associated with each site.

Table 1: Typical multiplet configurations on the auxiliary bonds obtained from symmetric iPEPS simulations on the square lattice with two symmetric spinful orbitals. The rows show the results for varying multiplet bond dimension $D^*$ (left column) at half filling. The corresponding state space dimension $D$ is listed in the right column. In the multiplet listing on the left, the notation $(\cdot)^m$ indicates $m$ multiplets in the symmetry sector $(\cdot)$, with $m = 1$ if not specified. For better readability, we also adopt the QSpace [30] convention of specifying SU(2) multiplets through the integer number $2S$ (i.e., the number of boxes in the corresponding Young tableaux).

| $D^*$ | multiplets in symmetry sectors $(\mathbb{Z}_2, \mathrm{SU}(2)_{\mathrm{spin}}, \mathrm{SU}(2)_{\mathrm{orb}})$ | $D$ |
|---|---|---|
| 3 | $(1,0,0) \oplus (-1,1,1) \ \oplus (1,2,0)$ | $1+4+3=8$ |
| 4 | $(1,0,0) \oplus (-1,1,1)^2 \oplus (1,2,0)$ | $1+8+3=12$ |
| 5 | $(1,0,0) \oplus (-1,1,1)^2 \oplus (1,2,0) \oplus (1,2,2)$ | $1+8+3+9=21$ |
| 6 | $(1,0,0) \oplus (-1,1,1)^2 \oplus (1,2,0) \oplus (1,0,2) \oplus (1,2,2)$ | $1+8+3+3+9=24$ |

We study the Hamiltonian (80) for finite hole hoping by tuning $\mu \leq 0$ (which is equivalent by particle-hole transformation to particle doping $\mu \geq 0$). To our knowledge, the phase diagram of this system is largely unknown away from integer filling. However, some interesting results are available for certain points in parameter space.

At half-filling $\langle n \rangle = 2$, several studies based on a sign-problem-free determinant quantum Monte-Carlo method addressed the magnetic properties of the model [105–107]. Their findings support the existence of long-ranged antiferromagnetic (AF) order for larger interactions $U \geq 2$ [106]. Interestingly, the AF order does not show a monotonic behavior with respect to $U$; instead, it exhibits a maximum around $U \approx 8$ and then decreases again towards larger interactions strengths. Whether or not the long-ranged AF order persists in the limit $U \to \infty$ remains an open question. A previous QMC study of the corresponding Heisenberg model found no AF order but potentially a gapless spin-liquid phase in this regime [108]. Another recent work based on variational QMC [109] addressed the Mott transition of the half-filled Hubbard model, finding a critical coupling $U_c \approx 11$ for the case of degenerate bands (their ansatz is rather biased, however, as it only accounts for a non-magnetic solutions).

In the quarter-filled case $\langle n \rangle = 1$ at infinite $U$, the Hamiltonian (80) can be mapped on an SU(4)-symmetric Heisenberg model, which was studied in Ref. [110]. Their combined iPEPS and ED study finds a rather exotic Neel-like order with dimers alternating between pairs of flavors, pointing towards a spontaneously broken SU(4) symmetry with an enlarged unit cell.

In this section, we present a first step towards a systematic iPEPS study of the symmetric two-band Hubbard model (80) that, in addition to half- and quarter filling, also investigates arbitrary doping regimes. The main challenge for iPEPS in the context of such a two-band model is the strongly enlarged local Hilbert space. In total, we need to deal with four different flavors of fermions (2 spins × 2 orbitals) resulting in a local state space dimension $d = 16$ per site, larger by a factor of four relative to the $d = 4$ in the one-band version.

To treat systems with a large local state space within iPEPS (or other TN approaches) one can follow two different strategies, as illustrated in Fig. 7: (a) either one keeps a lattice as a single unit with a large local state space (and hence preserves its symmetry), or (b) artificially splits it, for the sake of the iPEPS simulation, into smaller sublattices. Strategy (a) is hardly feasible for standard iPEPS techniques, even when incorporating all abelian symmetries of the system. For (b), a natural choice is to split the lattice into two interleaved sublattices, one for each orbital. The drawback, besides an artificially broken lattice symmetry, is that iPEPS then has to handle longer-ranged interactions and correlations in its ansatz. This necessitates

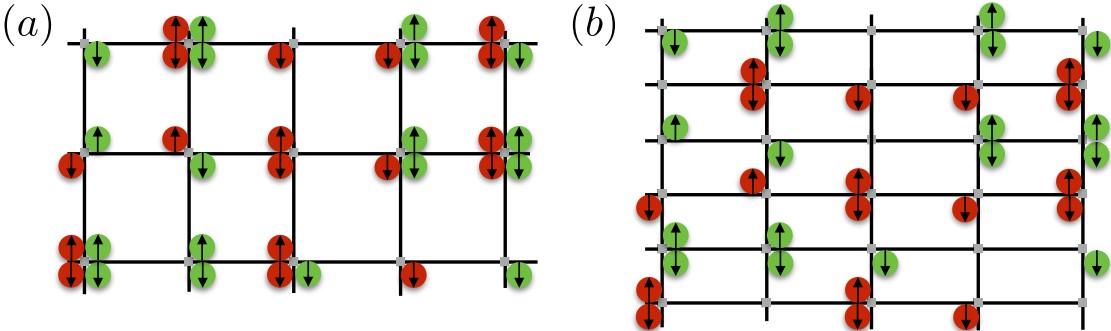

Figure 7: Schematic depiction of two-band setups for a spinful Hubbard model, with the two bands depicted by the different colors red and green. In setup (a) all four fermionic flavors still reside on a given lattice site, leading to an enlarged Hilbert space of $d = 4^2$. This setup respects flavor symmetry, which thus may be exploited. Setup (b) avoids the enlarged local Hilbert space by splitting the lattice into two sublattice, one for each band. This comes at the cost of introducing an additional set of sites, causing interaction terms to become longer-ranged.

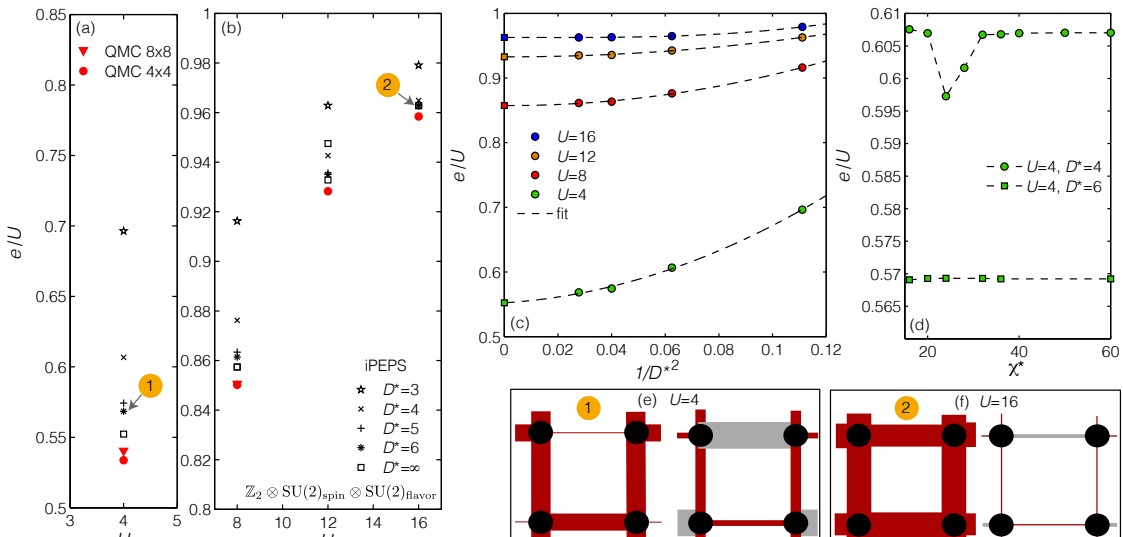

Figure 8: Non-abelian iPEPS results for the two-band Hubbard model with a 2×2 unit cell using simple update at half-filling $\langle n \rangle = 2$. Panels (a) and (b) display the normalized ground-state energy per site $e_0/U$ as a function of $U$ from iPEPS for various multiplet bond dimensions $D^*$ (black symbols) in comparison to QMC data (red symbols). The iPEPS energies were obtained by extrapolation vs. $1/D^{*2} \to 0$ (squares), with the extrapolations shown in (c). The convergence of the energy with the environmental bond dimension $\chi^*$ is shown in (d), where the maximum $\chi^* = 60$ roughly corresponds to $\chi = 200$. Labels (1) and (2) in panels (a) and (b) point to individual iPEPS wavefunctions characterized in panels (e) and (f). There the diameter of the black dots is proportional to the average local occupation, and the bond width to the bond energy $e_0^{ij}$ [Eq. (81b)]. To better illustrate the breaking of translational invariance in the unit cell, the right subpanels in (e) and (f) depict the same wavefunctions, but with bond energies shifted relative to their mean, $e_0^{ij} \to e_0^{ij} - (e_0/2)$. Here red (gray) bond correspond to positive (negative) values, respectively.

swap gates in the implementation of imaginary time evolution, which generates an additional source of error.

Here we follow strategy (a) because this preserves the orbital SU(2) symmetry, where we can fully exploit all available non-abelian symmetry. Specifically, with finite doping in mind, we incorporate $\mathbb{Z}_2 \otimes \text{SU}(2)_{\text{spin}} \otimes \text{SU}(2)_{\text{orb}}$ symmetry. This way, the local state space with $d = 16$ is reduced to an effective multiplet dimension $d^* = 6$. At the same time this enables us to retain up to $D^* = 6$ multiplets on each virtual bond, which corresponds to an effective bond dimension of $D = 24$ states [cf. Table. 1]. This enables us to run simple-update simulations for a wide regime of parameters, the results of which are presented in the following.

We start with the half-filled case $\langle n \rangle = 2$, i.e., $\mu = 0$ in (80b), to benchmark against existing determinant projector QMC data [111]. The results of this analysis are summarized in Fig. 8. Panels (a,b) show the normalized ground-state energies per site versus the interaction strength $U$ obtained from a simple-update iPEPS simulation on a $2 \times 2$ unit cell. The various bond dimensions $D^* = 3, 4, 5, 6$ in Fig. 8(a,b) are made up of dominant multiplets which emerge dynamically from the iPEPS simulations for each $D^*$. They are listed in Table 1, for completeness. The extrapolated energies for $1/(D^*)^2 \rightarrow 0$ are empirically determined by polynomial fits as depicted in Fig. 8(c). The convergence of our data with respect to the environmental bond dimension $\chi^*$ is shown in Fig. 8(d). We attach no significance to the bump seen at small $\chi^*$, since our focus is on the large-$\chi^*$ convergence. Note that QMC simulates finite-size systems with periodic BC, hence its ground state energy, specifically so in Fig. 8(a), is expected to still increase with increasing system size, as it converges from below. Nevertheless, we find good agreement, to within 1%, of our extrapolated energies with the QMC results, confirming the reliability of our approach.

At half filling, following the work of Ref. [106], we expect the presence of long-ranged AF order for all values of $U$ considered in Fig. 8. This is also supported by the Mott plateau seen in Fig. 9(b,d) at half-filling. Since by construction our iPEPS is $\text{SU}(2)_{\text{spin}}$ invariant, however, a direct measurement of the local magnetization is not possible. Nevertheless, we expect that the symmetry-breaking AF order still to be present, yet *symmetrized* and hence only accessible via static spin-spin correlations over longer distances.

In the context of symmetric iPEPS simulations for a spin-$\frac{1}{2}$ Heisenberg model, we have observed (not shown) that the two-fold degeneracy in the AF ground state manifests itself as a spontaneous formation of row or column stripes which, in agreement with the AF state itself, breaks translational symmetry within the unit cell. Interestingly, we here also observe such an effect in the iPEPS wavefunctions in the 2-band Hubbard model as shown in Figs. 8(e,f). For $U = 4$ [Fig. 8(e)], we clearly observe that two out of eight independent bonds in the unit cell carry a substantially reduced energy. This suggests (at least) a 4-fold degenerate ground state.

Based on this loose connection, we will refer to the symmetry-broken regime as the AF regime where the strength of the spatial symmetry breaking in our simulations may roughly correlate with the AF magnetic moment. For $U = 10$ [Fig. 8(f)] the "AF order" is weaker than at $U = 4$, consistent with the finding of Ref. [106] that the strength of AF order decrease for $U \rightarrow \infty$. Ultimately, of course, the precise AF nature needs to be studied via long-ranged spin-spin correlations. This is left for future work.

Next we turn to the case of arbitrary filling away from half-filling, which is equally accessible to iPEPS, but not to QMC. We focus on small to intermediate interactions, $U = 4$ and $U = 8$. By symmetry, it is sufficient to consider only the case of finite hole doping, $\delta \equiv 2 - \langle n \rangle > 0$, i.e., $\langle n \rangle < 2$. For the 2-band case, this regime has not been explored in detail by other methods so far. Figure 9 summarizes our iPEPS results as a function of filling, tuned by means of a chemical potential [cf. Eq. (80b)]. Figures 9(a,c) show the ground-state energy per site, $e_0/U$, as a function of $\delta$ for $D^* = 5$ and 6.

The dependence of the filling $\langle n \rangle = 2 - \delta$ on the chemical potential is shown in Figs. 9(b,d).

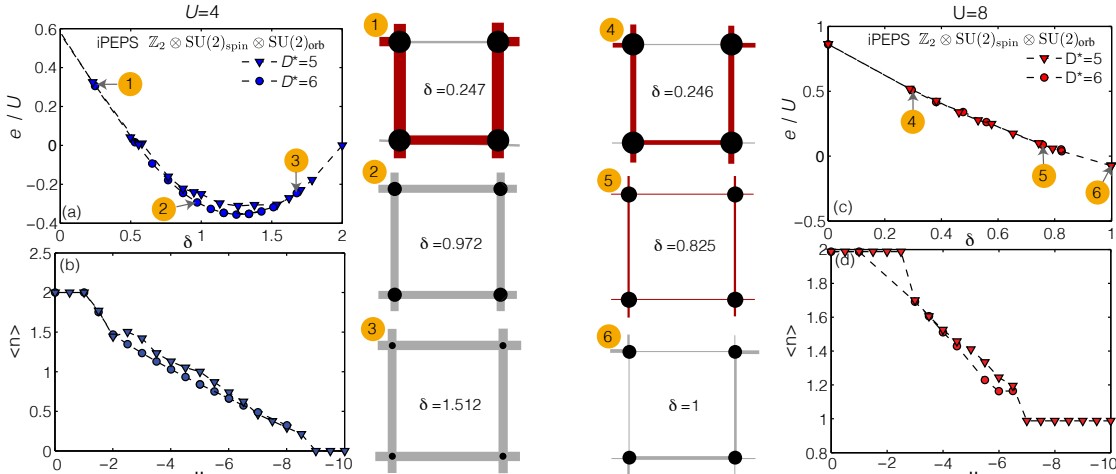

Figure 9: Non-abelian iPEPS results for the two-band Hubbard model away from half filling for $U = 4$ (left panels) and $U = 8$ (right panels). Panels (a) and (c) display the normalized ground-state energy $e_0/U$ as a function of doping $\delta$ for multiplet bond dimensions $D^* = 5, 6$. (b) and (d) show the filling $\langle n \rangle$ as a function of the chemical potential $\mu$. (In contrast to Figs. 8(a,b), no QMC data are available for comparison here, hence no $1/D^{*2}$ extrapolations were performed.) The parameter points (1) to (6) are analyzed in detail in the corresponding panels (1-6) in the center by characterizing the underlying iPEPS wave function. Again, the filling per site and the bond energy are proportional to the diameter of the black dots and the width of the bonds, where red (gray) bond correspond to positive (negative) values, respectively. The bond energies change signs at small doping, which is due to the definition of $e_0^{ij}$ in Eq. (81b), where the Coulomb interaction energy (positive) competes with the kinetic energy (negative).

For either $U$, the systems are in the AF regime for zero or small doping $\delta$, as inferred from the symmetry-broken states depicted in Figs. 9(1,4). For $U = 4$ we find an energy minimum around $\delta \simeq 1.2$. In this regime, we still observe a significant dependence of the energy on bond dimension $D^*$, hinting at a strongly entangled ground state. For $U = 8$, for the same range in chemical potential [Fig. 9(b,d)], we reach a smaller range in doping [Fig. 9(c)]. Since here the interaction strength is comparable to the non-interacting bandwidth is $W = 8$, we also see a Mott plateau at $\langle \hat{n} \rangle = 1$ [Fig. 9(d)] that is absent for $U = 4$ [Fig. 9(b)] [112].

At zero filling, i.e, $\delta = 2$, the ground state energy is zero, i.e. with Eq. (81a), $e_0(n = 0) = 0$ [similar as in Fig. 9(a)] irrespective of the strength of $U$. Therefore the data in Fig. 9(c), already turning negative, will necessarily also reach a minimum somewhere in the regime for $1 < \delta < 2$.

In addition to antiferromagnetism, we also expect superconducting order to play an important role in the two-band Hubbard model at finite hole doping. To check for the presence of $d$-wave superconductivity, we measure a generalized singlet-pairing amplitude $\Delta^{ij}$ [Eq. (81c)]. The results for different values of $U$ and $\delta$ are displayed in Fig. 10. We find that, indeed, superconducting order is present for the entire doping range $0 < \delta < 1$ for all considered interaction strengths. Two effects that will require further attention in the future, are the suppression of superconductivity at $\delta = 1$, and the fact that $\Delta$ decreases with increasing interaction strength. Both appear justified on intuitive grounds, however: Charge fluctuations are suppressed with increasing interaction strength, specifically so at integer filling. Moreover, for filling $n \lesssim 1$, local double occupancy is strongly suppressed for sizable $U$, yet double occupancy is required for finite $\Delta$ to start with. We also observe strong inhomogeneity of $\Delta^{ij}$ across different bonds.

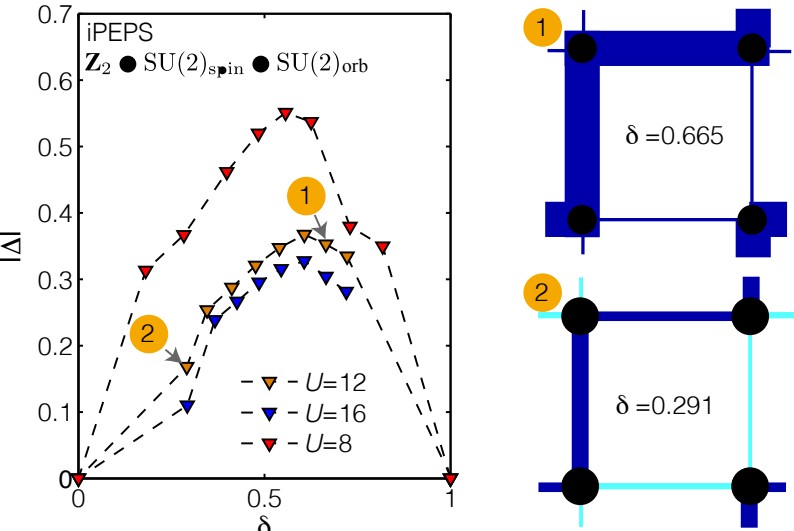

Figure 10: Generalized singlet-pairing amplitude $|\Delta|$ per site, extracted from iPEPS wavefunctions with $D^* = 5$ as a function of the hole doping $\delta$. $|\Delta| \equiv \langle |\Delta^{ij}| \rangle$ is obtained by averaging over the absolute value of $\Delta^{ij}$ for each bond in the unit cell. Labels 1 and 2 point to individual iPEPS wavefunctions characterized on the right, where the filling per site and the singlet-pairing amplitude are proportional to the diameter of the black dots and the width of the bonds, respectively [blue (cyan) bond correspond to positive (negative) $\Delta^{ij}$].

This may indicate a tendency toward spontaneous symmetry breaking of the orbital symmetry that is conserved by construction in our iPEPS implementation, or to the fact that the actual ground state breaks translational symmetry in a different way. Simulations on different unit-cell geometries are needed to shed light on this issue.

In conclusion, we have presented first fermionic iPEPS simulations of the two-band Hubbard model, which incorporates spin- and orbital SU(2) symmetry explicitly in the TN ansatz. The excellent agreement of our results found at half-filling with QMC data encouraged us to explore also the hole-doped regime, where our initial results uncover a number of intriguing features. Going forward, much work remains to be done to fully understand the guiding mechanisms and phases in this regime. This includes the study of longer-ranged spin-spin correlators, the comparison to simulations on different unit cells and unveiling the dependencies of various quantities such as energy and $d$-wave pairing as a function of interaction strength and doping more carefully. Since in the present model spin and orbital flavors are equivalent (e.g., there is no onsite Hund's coupling $J$), the efficiency of iPEPS could be further enhanced by exploiting the full SU(4)$_{\text{flavor}}$ symmetry present in the Hamiltonian within QSpace[30]. After fully understanding the phase diagram in this parameter regime, it will be highly interesting to study the effects of finite Hund's coupling $J$ on the emergence of superconductivity and other competing orders. Moreover, it would also be worthwhile to analyze whether abelian iPEPS simulations are numerically feasible in a modified setup involving separate sublattice for the two bands (c.f. Fig. 7). This would yield a different perspective on the ground-state properties of the model, especially in the context of spontaneous symmetry breaking.

## 6.2 Three-flavor Hubbard model

In addition to basic SU(2) symmetries, QSpace[30] also provides a convenient framework for the incorporation of more complex non-abelian symmetries such as SU($N > 2$). To explore the

potential of this feature within fermionic iPEPS, we consider a symmetric spinless three-flavor Hubbard model where we fully exploit the SU(3) flavor symmetry. Its Hamiltonian has the same form as in (80), except that the composite index $(m, \sigma)$ is replaced by a single flavor index, $m = 1, 2, 3$. Choosing $\mu_0 = U$ here, this again also ensures that $\mu = 0$ corresponds to half-filling. In contrast to the spinful case, however, the fact that $N = 3$ is odd implies that half-filling is metallic, unless symmetry broken (see below). Only integer filling results in Mott or Heisenberg physics for larger $U$ [112].

Although systems with a total of three symmetric flavors are not naturally realized by the atomic configuration of any real electronic material, SU($N > 2$) realizations of the fermionic Hubbard model currently attract a lot of attention in the context of cold-atom experiments based on alkaline earth-like atoms such as ytterbium [102, 103], where such systems have become directly accessible in highly controlled setups. SU($N$) symmetric systems feature a number of exotic phases and magnetic properties, which are of interest from a condensed matter perspective. In addition, they are also relevant for other fields, for example in the context of studying lattice gauge theories for quantum chromodynamics [113].

So far, little is known for the spinless SU(3) Hubbard model on the 2D square lattice. Some work has been done for the weak to intermediate coupling limit, where one expects the emergence of a flavor density wave breaking the translational symmetry of the lattice [114]. At half filling in particular, it is expected that two flavors occupy the same lattice site whereas neighboring sites exclusively host the third flavor, such that a bipartite two-sublattice structure emerges. This is motivated by the following consideration: a site with single occupancy transforms in the defining three-dimensional representation **3** of SU(3), whereas a doubly filled site is a fully filled site with one hole, which transforms in the conjugate representation **3̄**. Within the symmetry broken setting above then, neighboring sites could, in principle, bind into a singlet configuration.

At integer filling $n = 1$ and in the strong coupling limit, the model can be mapped onto an SU(3) Heisenberg model in the defining **3** representation (physically equivalent, for $n = 2$, this becomes the dual **3̄**). This is believed to favor a three-sublattice order with finite magnetic moments [115]. On intuitive grounds, note that for an SU(3) Heisenberg model in the **3** representation, a multiple of three sites is required to form a singlet. This is not naturally suited to the square lattice, and hence results in frustration, eventually giving rise to a three-sublattice order.

We have again reduced the numerical complexity of our model system by fully incorporating the non-abelian SU(3) symmetry in the fermionic iPEPS ansatz. To this end, the full local fermionic state space, $d = 8$, can be reduced to $d^* = 4$ multiplets. We then performed simple-update calculations with a multiplet bond dimensions up to $D^* = 6$. Again, the symmetry sectors are dynamically adapted during the optimization. We illustrate examples of the relevant multiplet contributions encountered in iPEPS simulations with varying $D^*$ at half filling in Table 2.

We performed iPEPS simulations on both $2 \times 2$ and $3 \times 3$ unit cells with two and three different tensors, respectively, to slightly bias the emergence of the two- and three-sublattice order expected from the considerations discussed above. Any tendency towards spontaneous symmetry breaking of SU(3), are, however, symmetrized by our setup. Figures 11(a,b) summarize our results for the ground-state energy per site, $e_0/U$, as a function of filling, $\langle n \rangle$, both at weak coupling $U = 1$ and stronger coupling $U = 6$. In either case, the simulations on both unit-cell geometries surprisingly yield very compatible ground-state energies.

For $U = 1$ at half-filling, which in the present case of $N = 3$ corresponds to half-integer filling on average, we observe a tendency toward translational symmetry breaking in the form of modulation of the occupancy on different sites for both $2 \times 2$ and $3 \times 3$ clusters (wavefunction 1 and 2). This is in qualitative agreement with Ref. [114], which predicts a phase with

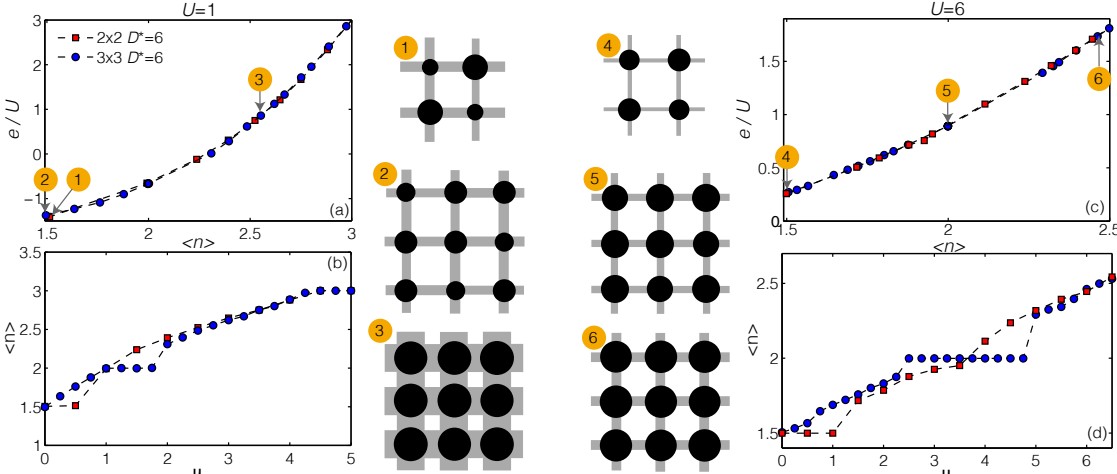

Figure 11: Non-abelian iPEPS results for the three-flavor Hubbard model for $U = 1$ (left panels) and $U = 6$ (right panels). Panels (a) and (c) display the ground-state energy $e_0/U$ as a function of filling $\langle n \rangle$ for iPEPS simulations on a $2 \times 2$ and $3 \times 3$ unit cell, whereas (b) and (d) show the filling $\langle n \rangle$ as a function of the chemical potential $\mu$. Panels (1) to (6) depict individual iPEPS wavefunctions at the points marked in panels (a,c). The filling per site and the bond energy are proportional to the diameter of the black dots and the width of the bonds, respectively.

Table 2: Typical multiplet configurations on the auxiliary bonds obtained from SU(3) symmetric iPEPS simulations on the square lattice Hubbard model with three equivalent flavors. The different rows show the results for increasing multiplet bond dimension $D^*$ (left column) at half filling. The SU(3) multiplet labels are in Dynkin form, where we adopt the compact QSpace[30] notation. For the center column we use the same notation as in Table 1.

| $D^*$ | multiplets in symmetry sectors ($\mathbb{Z}_2$, SU(3)$_{\text{flavor}}$) | $D$ |
|---|---|---|
| 4 | $(-1,00) \oplus (-1,01) \oplus (1,01) \oplus (1,10)$ | $1 + 3 + 3 + 3 = 10$ |
| 5 | $(-1,00) \oplus (-1,11) \oplus (1,01) \oplus (1,10)$ | $1 + 8 + 3 + 3 = 16$ |
| 6 | $(-1,00) \oplus (-1,11) \oplus (1,01)^2 \oplus (1,10)$ | $1 + 8 + 6 + 3 = 19$ |

two-sublattice order with single and double occupancy on neighboring sites. This is almost realized by wavefunction 1 shown in Fig. 11, with occupancies $N \approx 1.19$ and $N \approx 1.81$ on neighboring sites. For the $2 \times 2$ cluster, this also goes hand in hand with a pinning of the occupation at average $\langle n \rangle = 1.5$ [Fig. 11(b,d)], suggesting that the system energetically prefers a translationlly symmetry broken state. The density modulation are substantially suppressed on the $3 \times 3$ unit cell, where we find two sites having the same occupancy $N \approx 1.58$ while slightly fewer particles occupy the third site $N \approx 1.32$ at essentially no pinning of the average occupation when changing the chemical potential. The density-wave modulation disappears both in the case when the occupation significantly deviates from the half-filled case, and also for stronger interaction, as illustrated by the wave functions 3, 4, 5, and 6 in Fig. 11.

As already pointed out with Figs. 11(b,d), the occupancy is clearly not a smooth increasing function of the chemical potential, which drives the filling. While the $2 \times 2$ unit cell shows plateaus – and hence favors – half-filling, this is not the case for the $3 \times 3$ unit cells. The situation is completely reverse, however, at integer filling $\langle n \rangle = 2$ as seen in Figs. 11(b,d).

At this filling, a $2 \times 2$ unit cell cannot be in a singlet configuration, but has residual spin. Hence there is a certain degree of frustration in this setup. By contrast, the $3 \times 3$ unit cell can host a singlet configuration at $\langle n \rangle = 2$. Interestingly, the $3 \times 3$ unit cell already shows charge locking for the case of rather smaller $U = 1$, which may be due to frustration in the present case. Eventually, however, this will require a more thorough analysis based on an extrapolation of $D^* \to \infty$.

Locking of charge at integer filling is typically a signature of Mott physics, which is to some extent also expected in the three-flavor model at $\langle n \rangle = 2$ [112, 116]. However, locking may also occur if the occupation inside an enlarged unit cell changes by integers. This effect may be physical, e.g., as suggested above, in that $\mathbf{3}$ and $\bar{\mathbf{3}}$ bind into singlets, which occurs at half-filling. The effect may also be artificial, in which case it depends on numerical details and should become less pronounced with increasing $D^*$. This can be observed for the plateau at filling $\langle n \rangle = 1.5$ (data not shown).

In summary, nevertheless, based on the earlier arguments we do expect that in the present case the $2 \times 2$ unit cell is more suitable for the half-filled case, whereas the $3 \times 3$ unit cell is a better fit for integer filling. Furthermore, it should be possible to reveal additional information about the flavor order by studying (i) longer-ranged correlators and (ii) switching off the SU(3) in favor of two abelian U(1) symmetries and explicitly allowing spontaneous breaking of the flavor symmetry.

# 7 Conclusion

In this review, we attempt to give an overview of the rapid developments of iPEPS, which has reached a remarkable sophistication over the last few years. A large part of the review, addressed to newcomers to the field, is dedicated to to two widely used ground state search methods: simple-update and full-update. Simple-update is very competitive in run-time, while full-update yields highly accurate results that are important to characterize ground states of correlated electrons. Besides that, we present a comprehensive technical detail about using non-abelian symmetry in iPEPS, where a seemly formidable computational overhead can be avoided by careful implementation. Two non-trivial examples, the two-band Hubbard model and the three-flavor Hubbard model, are included to show how exploiting symmetry can be useful. All in all, we hope that this review will motivate more efforts to the development of 2D tensor network algorithms, which have the potential for achieving crucial for advances in computational studies of correlated electrons.

**Funding information** The Deutsche Forschungsgemeinschaft supported BB, JWL and JvD through the Excellence Cluster "Nanosystems Initiative Munich" and the Excellence Strategy-EXC-2111-390814868. JWL was also supported by DFG WE4819/3-1. AW was supported by the U.S. Department of Energy, Office of Basic Energy Sciences, under Contract No. DE-SC0012704.

# A Constructing tensors with symmetry

In this Appendix, we provide a sketch of how to deal with non-abelian symmetry in tensor networks. For simplicity, we use SU(2) as a concrete example. The strategy can be generalized to SU($N$) (for more detail, we refer to Ref. 30, 31). The example illustrates the conceptual bottom-up approach underlying the QSpace tensor library [30, 31] for implementing symmetries in tensor networks: construct all ingredients step by step, systematically combining

elementary building blocks into more complex structures.

## A.1 SU(2) spin algebra

A group element of SU(2) can be represented by an unitary transformation, $\hat{G} = e^{i\hat{S}}$, in a complex vector space, with $\hat{S}$ an arbitrary Hermitian matrix in that space. This matrix can be parametrized by three independent real numbers $\varphi_a$, with $a \in \{x, y, z\}$, such that $\hat{S} = \sum_a \varphi_a \hat{S}^a \equiv \varphi \cdot \hat{\mathbf{S}}$, with $\hat{S}^a$ the generators of the symmetry, satisfying $[\hat{S}^a, \hat{S}^b] = i\epsilon^{abc}\hat{S}^c$. In the defining, two-dimensional representation with spin multiplet label $S = \frac{1}{2}$, the generators can be chosen as $\hat{S}^a \equiv \frac{1}{2}\sigma^a$, with $\sigma^a$ the Pauli matrices.

This is the smallest non-trivial SU(2) matrix representation. More generally, an SU(2) irreducible representation (irrep) with spin $S$ has dimension $2S + 1$, i.e. the generators $\hat{S}$ are represented by matrices of size $(2S + 1) \times (2S + 1)$.

In general, an irreducible multiplet consists of a set of states that can be labeled by their eigenvalues of the generators that were chosen diagonal, i.e., in the SU(2) context, $S_z$. For a general non-abelian symmetry, this can be a set of generators, say $Q_z$, resulting in a tuple of labels $q_z$. These can be lexicographically sorted, with the *largest*, i.e., the maximum weight state being unique. Its *weights* $q_z$ therefore can be used to label the entire multiplet. In the SU(2) context, $\max(S_z) = S$.

Alternatively, the complete set of Casimir operators also labels a multiplet uniquely. Hence there exists a well-defined polynomial mapping from the maximum weight labels to the Casimir labels. For example, in the case of SU(2), $S \to S(S + 1)$, with the latter being the eigenvalue of the quadratic Casimir, $\hat{S}^2 \equiv \hat{S}^a \hat{S}_a$, using Einstein summation convention. Other than that, the Casimir operators are not required, and so we do not use them. Instead, we use convenient internal conventions on the normalization of generators, with a subsequent *linear* mapping of the maximum weight labels to standard Dynkin labels. In particular, this implies for SU(2) the symmetry labels $q = 2S$. For one, this is consistent with SU($N > 2$), e.g., in that $q$ is equivalent to the number of boxes in its corresponding Young tableau. Moreover, this also has the advantage that all symmetry labels are integers, which we find more readable and convenient on practical grounds. We use the notation $q$ as label for irreducible multiplets, in order to emphasize that this can be a tuple of labels for an irreducible multiplet for any symmetry.

## A.2 Tensor product decomposition

A tensor product of two irreps $q_1$ and $q_2$ can be decomposed into a direct sum of irreps,

$$V^{q_1} \otimes V^{q_2} = \bigoplus_q M^q_{q_1 q_2} V^q , \tag{82}$$

where in this symbolic notation, the multiplicity coefficients $M^q_{q_1 q_2}$ are integers encoding the fusion rules. That is, irreps that do occur in the product decomposition have $M^q_{q_1 q_2} > 0$, whereas multiplets $q$ with $M^q_{q_1 q_2} = 0$ do not occur in the decomposition [for SU(2) $M^q_{q_1 q_2} = 1$ for $q = 2S \in |q_1 - q_2|, |q_1 - q_2| + 2, q_1 + q_2$, and $M^q_{q_1 q_2} = 0$ otherwise]. For general non-abelian symmetries as for SU($N \geq 3$), the *same* irrep $q$ can routinely occur multiple times, i.e., having outer multiplicity $M^q_{q_1 q_2} > 1$.

The coupled basis vector $|q, q_z\rangle$ and the direct product basis $|q_1, q_{1z}\rangle \otimes |q_2, q_{2z}\rangle$ are related by a unitary basis transformation, namely,

$$|q_1, q_{1z}\rangle \otimes |q_2, q_{2z}\rangle = \Big( \underbrace{\sum_{q, q_z} |q, q_z\rangle \langle q, q_z|}_{\equiv \left( C^q_{q_1 q_2} \right)^{q_z}_{q_{1z} q_{2z}}} \Big) |q_1, q_{1z}\rangle \otimes |q_2, q_{2z}\rangle , \tag{83}$$

where $(C^q_{q_1 q_2})^{q_z}_{q_{1z} q_{2z}}$ are the standard Clebsch-Gordan coefficient (CGC) spaces. The notation here emphasizes the tensorial structure, in that the Clebsch-Gordan *tensor* (CGT) $C^q_{q_1 q_2}$ is indexed by the $q_z$-labels (in the presence of inner multiplicity where degeneracies in the the $q_z$-labels occur, caveats apply [30]). The rank-3 CGTs above are fundamental building blocks since any higher-rank CGC can be generated from them.

**Example: Direct product of two spin-half multiplets**   The tensor product of the two vector spaces of two spin-half multiplets, having $q_1 = q_2 = 2S = 1$, can be decomposed into a spin-singlet, $q = 0$, and a spin-triplet, $q = 2S = 2$, i.e, $1 \otimes 1 = 0 \oplus 2$.

The unitary basis transformation matrix from the direct product basis to the coupled basis can be read as

(using the familiar labels $\langle S, S_z |$ for the rows and $| S_{1z}, S_{2z} \rangle$ for the columns on the r.h.s.),

$$
\begin{pmatrix} \dfrac{C^0_{q_1 q_2}}{\rule{0pt}{0pt}} \\ C^2_{q_1 q_2} \end{pmatrix} \;=\; \begin{array}{c} \phantom{x} \\ \langle 0,0 | \\ \hline \langle 1,1 | \\ \langle 1,0 | \\ \langle 1,-1 | \end{array} \;\; \overset{\displaystyle |\uparrow\uparrow\rangle \;\; |\uparrow\downarrow\rangle \;\; |\downarrow\uparrow\rangle \;\; |\downarrow\downarrow\rangle}{\begin{pmatrix} 0 & 1/\sqrt{2} & -1/\sqrt{2} & 0 \\ \hline 1 & 0 & 0 & 0 \\ 0 & 1/\sqrt{2} & 1/\sqrt{2} & 0 \\ 0 & 0 & 0 & 1 \end{pmatrix}} \;\; . \tag{84}
$$

This includes two sets of CGCs concatenated vertically, namely for $q = 0$ and $q = 2$, as indicated by the horizontal lines separating them. These CGCs are fully defined by symmetries. They can be explicitly computed via (generalized) tensor-product decomposition [30], and stored as separate tensors in sparse format in a database.

## A.3   Irreducible tensor operator

State spaces and operators are tightly related. For example, if one creates a particle with spin-half on top of a singlet (or vacuum state), $\hat{c}^\dagger_\sigma |0\rangle \equiv |\sigma\rangle$, the operator on the l.h.s. necessarily needs to transform under symmetry like the resulting state on the r.h.s. Symmetry operations on the state to the right translate into commutation relations for the operators on the left, and Clebsch-Gordan coefficients come into play, as also evidenced by the Wigner-Eckart theorem.

For example, in the case of SU(2), the operation of a raising or lowering operator for an irreducible operator (irrop) $\hat{T}^{SS_z}$, which by notation transforms like a spin-$S$ multiplet, translates into the following relations:

$$
[(\hat{S}_x \pm i\hat{S}_y), \hat{T}^{SS_z}] \;=\; \sqrt{S(S+1) - S_z(S_z \pm 1)} \; \hat{T}^{SS_z \pm 1} \tag{85}
$$

$$
[\hat{S}_z, \hat{T}^{SS_z}] \;=\; S_z \, \hat{T}^{SS_z} \, . \tag{86}
$$

This demonstrates that just as a multiplet $|SS_z\rangle$ is irreducible under a given symmetry, so is the irrop. In particular, an irrop represents a set of operators, here labeled by $S_z$, which carries a representation of the symmetry group.

Now if an irrop acts on a non-trivial state space that itself transforms like a non-scalar symmetry multiplet, the resulting states correspond to a tensor product of symmetry multiplets, and the rules of tensor product decomposition of multiplets apply. This is manifested in the Wigner Eckart theorem (returning to generic 'q-labels'),

$$
\langle q_1 q_{1z} | \hat{T}^{q q_z} | q_2 q_{2z} \rangle \equiv \langle q_1 q_{1z} | \cdot \left( \hat{T}^{q q_z} \times | q_2 q_{2z} \rangle \right) = \langle q_1 \| \hat{T}^q \| q_2 \rangle \left( C^{q_1}_{q q_2} \right)^{q_{1z}}_{q_z q_{2z}}, \tag{87}
$$

where the *reduced matrix element* $\langle q_1 \| \hat{T}^q \| q_2 \rangle$ is the only remaining effective matrix element

not determined by symmetry, but depending on the physical action of the operator. Other than that, the Wigner-Eckart theorem demonstrates that the matrix elements of an irrop are not independent of each other, but are highly constrained by symmetry operations, i.e., related by CGCs. In other words, an SU(2) symmetric tensor can be factorized into two parts, *reduced tensor elements* and CGCs. With both state space decomposition and operator representation thus linked to CGTs, this forms a natural framework to build tensors of arbitrary complexity precisely on the basis of tensor network states.

## A.4   PEPS tensor construction

Here we demonstrate, based on an instructive example relevant for the present context, how complex tensors can be built from elementary building blocks, making explicit use of rank-3 CGTs. With the focus on iPEPS in this work, the building block of the iPEPS itself is a local rank-5 tensor $M$, with four virtual bond indices, say L(eft), R(ight), U(p), and D(own), together with a local state space, P(hysical). So how would one build, or even initialize such a tensor while respecting non-abelian symmetries in a generic fashion?

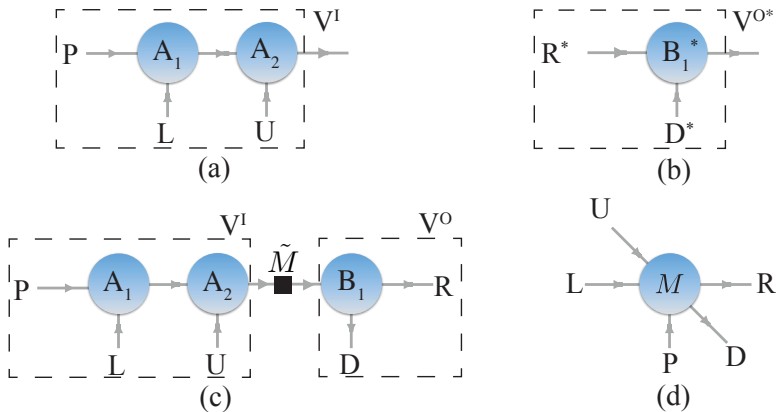

Figure 12:   Steps (a)-(c) for building rank-5 PEPS tensor in (d) from elementary rank-3 tensors.   (a) Iterative fusion of the "incoming" state spaces, $V^P \otimes V^L \otimes V^U \equiv \left( V^P \otimes V^L \right) \otimes V^U \equiv V^I$, first combining the physical states space $P$ with the left bond $L$ via $A_1$, then fusing the result with the upper bond $U$ via $A_2$. (b) Fusion of the 'outgoing' states spaces, $V^{R*} \otimes V^{D*} \equiv V^{O*}$. (c) Fusing together the results of (a) and the *conjugate* [31] of (b) into a global singlet via the bond tensor $\tilde{M}$. The latter is block-diagonal with trivial CGCs; its reduced matrix elements can be chosen arbitrarily, e.g., such that the overall tensor may satisfy certain lattice symmetries. The indices $R$ and $D$ became outgoing indices in the last step. (d) The final rank-5 PEPS tensor after contracting all tensors in (c). Each leg $U$, $D$, $L$, $R$ and $P$ represents a state space which in the presence of non-abelian symmetries is organized via the generic composite labels $|ql; q_z\rangle$ as introduced with Eq. (71).

The prescription to build such a rank-5 PEPS tensor $M$ is summarized in Fig. 12. We assume that the state spaces of each of the constituent bonds are already specified (or have been obtained in some fashion). Importantly, each bond also has a direction, indicating whether an index *enters* or *leaves* the final desired object. As indicated in Fig. 12(d), the tensor $M$ has three incoming indices, $(P, L, U)$, and two outgoing indices, $(R, D)$. So one can fuse the state spaces in either case. A set of state spaces, such as $(L, U, P)$, can be build iteratively by adding one state space at a time, $V^P \otimes V^L \otimes V^U \equiv \left( V^P \otimes V^L \right) \otimes V^U \equiv V^I$, where $I$ stands for the combined incoming state space, as depicted in Fig. 12(a). The same can be done for the outgoing state spaces, except that by the very concept of fusing input spaces into output, the tensor product

deals with the opposite direction of $R$ and $D$, namely also ingoing. Hence the state space in the dual (or conjugate) representation needs to be considered in the fusion itself, as indicated by the asterisk in $V^{R*} \otimes V^{D*} \equiv V^{O*}$ in Fig. 12(b). Here $O$ stands for the combined state spaces of legs eventually leaving the tensor $M$.

Having fused in- and outgoing state spaces separately and without truncation, the final step is to tie together the two fused state spaces into a global singlet, symbolically written as $V^I \otimes V^{O*} \to 0$, with $q = 0$ the scalar representation, i.e., a singlet. This sixth index, namely a global singlet state, then corresponds to a singleton dimension that can be skipped. Since for any irrep $q$ only the combination with its dual $q^*$ can result in a singlet, the CGCs for this step are simple ($1j$ symbols in the language of [31]), as one can only link in a 1-to-1 correspondence between $V^I$ and $V^{O*}$. However, the situation is much simpler still, since one needs to contract the *conjugate* of the $V^{O*}$ above, in order to obtain the final desired index directions. The conjugate of the entire object in Fig. 12(b) can be drawn pictorially as a mirror image [here left-to-right, as in Fig. 12(c)], with all arrows reversed [see [31] for details]. This now can be simply contracted with the tensors in Fig. 12(a) as shown in Fig. 12(c). Since arrow directions are preserved, this implies that the only free choice of tensor coefficients left are in the tensor $\tilde{M}$ in Fig. 12(c) that ties together in- and outgoing state spaces. Via Wigner Eckart theorem, the tensor $\tilde{M}$ is a scalar operator, where with the corresponding singleton dimension of the irrop set skipped, this can be written as a plain block-diagonal tensor, with the corresponding CGTs $C_{0q}^q = 1^q$ being trivial identity matrizes in the multiplet space of the respective multiplet $q$.

The above example reflects the generic transparent guiding principle when working with symmetric tensors, namely: to construct arbitrarily complex tensors from known, manageable, elementary building blocks. In the present case, this included (i) the fusion of pairs of state spaces [via $A_i$ and $B_i$, as well as the final fusion into a trivial scalar multiplet via $\tilde{M}$ in Figs. 12(a-c)]. This was followed by (ii) the pairwise contraction of symmetric tensors [31] to obtain $M$ in Fig. 12(d). Here, for example, one may have used the nested pairwise grouping $A_1 A_2 \tilde{M} B_1 = ((A_1 * A_2) * (\tilde{M} * B_1))$, where '$*$' refers to the contraction of a pair tensors on fully connected indices, which simply generalizes matrix multiplication to tensors.

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
