# Peer review of "A beginner's guide to non-abelian iPEPS for correlated fermions"

_SciPost Physics, doi:SciPost Phys. Lect. Notes 25 (2021)_

## Round 2 · Referee Report · Anonymous (Referee 2) · 2020-8-14

Report

This review provides a detailed introduction to infinite projected entangled-pair states (iPEPS) for fermionic systems with a non-abelian symmetry. It is explained in detail how the symmetry and fermions can be implemented on the level of the PEPS tensors. The involved technical frameworks are not new and have been previously discussed in other papers, however, here they are presented in a unified way. The most important ingredients of the iPEPS algorithms based on imaginary time evolution are also discussed. As example applications, the authors present initial results for two different multi-band Hubbard models.

The paper is clearly structured and written in a pedagogical way, making it a valuable reference especially for non-experts. The example applications are interesting (although a full study will require more calculations as the authors point out), providing evidence of the validity of the approach and showing the impressive bond dimensions that can be reached by exploiting the non-abelian symmetry.

Thus, I can recommend this paper to be published in SciPost Physics. I only have minor comments, listed below, which the authors may want to address when revising their paper.

List of comments

1) Eq.6: is there a particular motivation to use a ket notation (|\sigma^y_x>) on the physical leg, rather than just putting a physical index? After Eq (4) the physical index \sigma^x_y is introduced, thus why not using the physical index instead, which is more common, rather than a ket? That notation may be confusing (also calling the tensor in Eq.6 a rank-4 tensor, rather than a rank-5 tensor may be confusing). The same comment holds also for other figures with labels on the physical leg (Eq(7), Eq(11), …)

2) In Sec 3.5 it would be good to also mention the latest developments in using automatic differentiation to perform a gradient-based energy optimization, see H.-J. Liao, J.-G. Liu, L. Wang, and T. Xiang, Phys. Rev. X 9, 031041 (2019).

3) In the study of the spinful two-band Hubbard model, it would indeed be interesting to study larger unit cell sizes (as the authors point out). But I agree that this is probably beyond the scope of the present work.

4) Figure 11(a) and (c): on this scale it is difficult to see which energy is lower. Maybe it would be better to show the difference between the two (or subtracting the mean value), in order to better see which of the two states is lower in energy and by how much.

5) Minor typos:
- on page 40, 2nd line: a verb is missing after "10^9 individual contractions"
- page 41, line 4: "But whem" -> "But when"
- page 42, in the middle: "(a) Either" -> "(a) either"
- page 46, second paragraph, after Qspace [30]: "fter" -> "After"

  • validity: -
  • significance: -
  • originality: -
  • clarity: -
  • formatting: -
  • grammar: -

Author:  Jheng-Wei Li  on 2021-01-11  [id 1138]

(in reply to Report 2 on 2020-08-14)
Category:
answer to question

Attached (reply.pdf) is our reply to the referee's comment

Attachment:

reply_Z15GSSl.pdf

Anonymous on 2021-01-18  [id 1158]

(in reply to Jheng-Wei Li on 2021-01-11 [id 1138])
Category:
answer to question

Please refer to the attached as our reply to the referees' comment.

Attachment:

reply.pdf

---

## Round 2 · Referee Report · Matteo Rizzi (Referee 1) · 2020-8-14

Strengths

1- The manuscript has a pretty clear scope and structure: it provides an (almost) self-contained solid guide for PEPS-beginners, while providing some concrete examples concerning relatively uncharted systems. The latter part makes it visible, why the whole effort is worth.

2- The technical part on non-Abelian symmetries complements other existing sources in terms of a different approach to the same problem, though it would have been great to see a deeper discussion in this respect (see “Weaknesses”).

3- This review collects a lot of useful tricks, which one should otherwise search for between the lines of an increasingly large body of literature — thus offering a precious service to the beginners.

Weaknesses

1- According to the SciPost criteria, I would rate the manuscript not to contain enough new results to be published in the standard section, but rather to perfectly fit (up to minor changes) in the Lecture Notes one. The final Sec. 7 about the Hubbard model(s) offers an interesting collection of concrete and physically relevant examples, but does not convey a clear (possibly new) take-home message about the physics of those model(s). This is perfectly fine in a Lecture Note, but sub-optimal in a standard research paper.

2- The relative weight between the more standard PEPS formalism and the more peculiar one about non-Abelian symmetries is not well balanced to my eyes. Clearly, the latter is explained in full detail in Ref.[30], but the same applies even more for the whole of (i)PEPS general formalism in Sections 3 and 4/5. Given the title and declared aim of this manuscript, I would have expected a pedestrian summary of the QSpace approach here, in order to keep the whole (i) truly self-contained and (ii) at the beginner’s level as the rest.

Report

The Manuscript offers an ordered review of (i)PEPS techniques, highlighting the most relevant ones for the implementation of an algorithm from scratch, thus sparing considerable efforts to the beginners and newcomers to the field.

Particular emphasis is given, in the intentions at least, to the incorporation of fermionic statistics and non-Abelian symmetries, both crucial ingredients when aiming to tackle condensed matter paradigms like the multi-band Hubbard model, where a lot remains to be explored. In this respect, some concrete examples of calculations are provided in the final section, thus motivating the interested reader to struggle with the algorithm implementation to explore new physics.

Certainly, after the first thirty self-contained didactical pages on general aspects of (i)PEPS, the Reader would expect to find the same level of user-friendly description for the ingredients that set the Authors’ approach apart from other ones in the literature, namely the QSpace approach to non-Abelian symmetries. Unfortunately, however, apart a short comment about being a fully alternative approach to Ref. [48], not much more is provided to the reader, except for addressing to the extensive Ref.~[30]. At least a sketchy recapitulation of the QSpace idea and its implementation details would be highly desirable to keep the review self-contained and homogeneous in style, especially given the promise in the title.

Given the overall spirit of the Manuscript, I would rate it definitely as suitable for the Lecture Notes section of SciPost more than as Regular Article. A few more punctual remarks — anyway not as crucial as the criticism / suggestion above about QSpace — are listed below, with the aim of improving the clarity of the presentation.

Requested changes

1) At page 2, the claim “Computational costs scaling as $D^\alpha$ can thus potentially be reduced by a factor of (D/D^*)^\alpha” should actually be made a bit milder by taking into account the growing number of smaller objects to be contracted, as correctly discussed later in Sec. 6.1.3. E.g., for a matrix-matrix multiplication it should read $(D/D^*)^{\alpha-1}$ since there are order of $(D^*/D)$ small blocks to be multiplied with each other. Anyway, nothing to say about the fact that the gain is going to be quite spectacular :-)

2) At page 4, the statement “In other words, the computational cost to simulate a low-entangled state using a PEPS scales only polynomially with system size.” is misleading at this stage, since it somehow suggests a perfect analogy with the efficiency of MPS algorithms. However, as correctly discussed in Sec. 3.3, the contraction of a PEPS “represents an exponentially hard problem” and one should resort to “a variety of approximate schemes to deal with this issue.”

3) At the bottom of page 11, more quantitative indications and/or criteria to judge the convergence would be helpful for beginners. In particular, more quantitative substance to expressions like “typically multiple times”, “a few full steps”, “can significantly increase” would be of great help for the beginner trying to assess the performance of its own simulations. Same applies later on in Sec. 3.5.2 when discussing the convergence in terms of the imaginary time steps.

4) At the end of Sec. 3.4, when discussing the convergence in the environment bond dimension $\chi$, I would have expected a small discussion about the extrapolation criteria based on the discarded probability introduced in this specific context by P. Corboz.

5) At the end of Sec. 3.5.3, a sketchy description of the other approaches (e.g., fast-full update, but not only) might be useful, maybe under a further subsection “Alternative Approaches”. One could spare the necessary space, if needed, by dropping some little redundancies in the previous pages (e.g., Fig. 2).

6) I am not really sure why Sec. 4 and 5 about fermionic statistics should be two separate ones, and not merged into a single one :-)

7) The notation of Eq.(71) is sub-optimal: dropping the dependence of the Clebsch-Gordan coefficients on the symmetry charges (q,q’,q”) might implicitly suggest a full decoupling of the factors A and C, which is instead not there.

8) I was quite surprised not to find any discussion on how the Authors’ approach deals with non-trivial outer multiplicities, being them coming from SU(2) tensors with rank larger than three or form the symmetry group(s) themselves (like for SU(3)). In the end, they understandably claim that this is what “sets our work apart from that of other iPEPS practitioners”, but they do not provide the reader with any detail, even sketchy, about it. Especially for the beginners, who are declared target of the manuscript, that would be of vital importance, without requiring them to dig into the 75 pages of Ref.[30] at first round of reading. Same applies to the compact notation for the SU(N) labels :-)

9) About the actual data on Hubbard model(s), l wonder whether the extrapolation in $(D^*)^2$ has some deep meaning behind, or is that simply a numerical observation at this stage (I would be perfectly fine with it, if stated). Same applies to the weak (absent?) scaling with $\chi^*$ in panel 8(d): by the way, what is the strange bump in the $D^*=4$ series there? and why is $D^*=5$ not displayed?

9b) Are site occupation and bond energy extrapolated in a similar way before plotting them in Figs. 8-9? If yes, why is then not the same procedure applied to the singlet-pairing amplitudes in Fig. 10?

10) How comes that, while the SU(N) symmetries are directly incorporated in the Tensor Network Ansatz, the conservation of total particle number is not included, and the Authors resort to tuning the average filling via a chemical potential? This might be useful to be explained to the Reader.

11) In Fig. 9, orienting the $\mu$ and $\delta$ axis the same way (i.e., so that left-to-right the population grows / decreases) would probably help the reader a bit.

  • validity: high
  • significance: high
  • originality: good
  • clarity: good
  • formatting: excellent
  • grammar: excellent

Author:  Jheng-Wei Li  on 2021-01-11  [id 1137]

(in reply to Report 1 by Matteo Rizzi on 2020-08-14)
Category:
answer to question

Attached (reply.pdf) is the reply to referees' comments.

Attachment:

reply.pdf

Anonymous on 2021-01-18  [id 1159]

(in reply to Jheng-Wei Li on 2021-01-11 [id 1137])
Category:
answer to question

Please refer to the attached file as our reply to the referees' comments.

Attachment:

reply_BCKhEM9.pdf

---

## Round 3 · Referee Report · Matteo Rizzi · 2021-1-21

Report
I am glad to see that the Authors have taken all questions into serious consideration and have implemented quite a lot of changes in their manuscript to clarify them. Even the answers that did not get translated into the manuscript itself (e.g., because originated by some misunderstanding on my side) are anyway clearly formulated, and will remain publicly visible to the profit of readers with similar doubts.
I do not see any more reason preventing the publication of this work on SciPost, though I am still not sure that it does not qualify better for the Lecture Notes section than for the regular Physics one — but that is almost a detail of taste, and I trust the editorial decision in this respect.

---

## Round 3 · Author Response

Sincerely,
Benedikt Bruognolo, Jheng-Wei Li, Jan von Delft and Andreas Weichselbaum

---

## Round 3 · List of Changes

The principle changes are listed below in response to referees' comments
First Referee:
1. At page 2, the claim “Computational costs scaling as Dα can thus potentially be reduced by a factor of (D/D∗)α” should actually be made a bit milder by taking into account the growing number of smaller objects to be contracted, as correctly discussed later in Sec. 6.1.3. E.g., for a matrix-matrix multiplication it should read (D/D∗)α−1 since there are order of (D∗/D) small blocks to be multiplied with each other. Anyway, nothing to say about the fact that the gain is going to be quite spectacular :-)
We thank the Referee for careful reading. Indeed, the ideal speedup is difficult to reach due to the growing number of smaller objects to be contracted. In particular this concerns small D, whereas the overhead becomes smaller with growing D. Following the suggestion, we adapted our discussion on page 2. However, we disagree with the statement that there are order of (D∗/D) small blocks to be multiplied. In particular, note that the argument of small blocks identically also applies to abelian symmetries, for which D∗/D = 1.
2. At page 4, the statement “In other words, the computational cost to simulate a low- entangled state using a PEPS scales only polynomially with system size.” is misleading at this stage, since it somehow suggests a perfect analogy with the efficiency of MPS algo- rithms. However, as correctly discussed in Sec. 3.3, the contraction of a PEPS “represents an exponentially hard problem” and one should resort to “a variety of approximate schemes to deal with this issue.”
We agreed with this comment. To avoid the confusion, we removed the statement on p. 4.
3. At the bottom of page 11, more quantitative indications and/or criteria to judge the con- vergence would be helpful for beginners. In particular, more quantitative substance to expressions like “typically multiple times”, “a few full steps”, “can significantly increase” would be of great help for the beginner trying to assess the performance of its own simula- tions. Same applies later on in Sec. 3.5.2 when discussing the convergence in terms of the imaginary time steps.
Following the suggestion of the Referee, we modified the sentence accordingly on page 10.
4. At the end of Sec. 3.4, when discussing the convergence in the environment bond dimension χ, I would have expected a small discussion about the extrapolation criteria based on the discarded probability introduced in this specific context by P. Corboz.
We thank the referee for his comment. We added a few sentences at page 17 to address the extrapolation scheme by Corboz, Czarnik, Kapteijins and Tagliacozzo, and by Rader and Läuchli.
5. At the end of Sec. 3.5.3, a sketchy description of the other approaches (e.g., fast-full update, but not only) might be useful, maybe under a further subsection “Alternative Approaches”. One could spare the necessary space, if needed, by dropping some little redundancies in the previous pages (e.g., Fig. 2).
At the end of Sec. 3.5.3, we give a short discussion on the alternative approaches besides full-update. We have now added a subsubsection heading there, 3.5.4 Alternative approaches.
6. I am not really sure why Sec. 4 and 5 about fermionic statistics should be two separate ones, and not merged into a single one :-)
We agreed with the Referee’s comment, and have now merged Sec. 5 into Sec. 4..
7. The notation of Eq.(71) is sub-optimal: dropping the dependence of the Clebsch-Gordan coefficients on the symmetry charges (q,q′,q′′) might implicitly suggest a full decoupling of the factors A and C, which is instead not there.
We thank the Referee for his/her careful reading. We fixed Eqs.(71) and (72) accordingly on pages 33 and 34.
8. I was quite surprised not to find any discussion on how the Authors’ approach deals with non-trivial outer multiplicities, being them coming from SU(2) tensors with rank larger than three or form the symmetry group(s) themselves (like for SU(3)). In the end, they understandably claim that this is what “sets our work apart from that of other iPEPS practitioners”, but they do not provide the reader with any detail, even sketchy, about it. Especially for the beginners, who are declared target of the manuscript, that would be of vital importance, without requiring them to dig into the 75 pages of Ref.[30] at first round of reading. Same applies to the compact notation for the SU(N) labels :-)
Outer multiplicity (OM) is systematically taken care of starting with elementary rank-3 tensors as in the A tensor [see Eq. (71)], or also for the elementary irreducible operators as in Eq. (72). This issue was described in detail in Ref. 30. We have now added a new subsection 5.3 Outer multiplicities, offering a brief summary on outer multiplicities and their implications. OM arises also when building higher-rank tensors via contraction; the consistent treatment of OM for these follows Ref. 31. We also kindly refer the reader to the detailed discussions there. We note that OM increases with tensor rank, leading to a proliferation of blocks. Yet even for abelian symmetries, where the outer multiplicity is always equal to 1, the number of (small) blocks increases exponentially with increasing tensor rank. In either case, the proliferation of small blocks needs to be avoided and can be dealt with efficiently by fusing indices. We describe how to deal with the large number of small objects in Sec. 5.5.3.
9. About the actual data on Hubbard model(s), l wonder whether the extrapolation in (D∗)2 has some deep meaning behind, or is that simply a numerical observation at this stage (I would be perfectly fine with it, if stated). Same applies to the weak (absent?) scaling with χ∗ in panel 8(d): by the way, what is the strange bump in the D∗ = 4 series there? and why is D∗ = 5 not displayed?
The choice of (D∗)2 in Fig. 8(c) is purely empirical (and we now state so), as is the choice of χ∗ in Fig. 8(d). For Fig. 8(d), we are looking for consistent behaviour at large χ∗ limit and we do not think the energy deviation at small χ∗ carries significant meaning.
10. Are site occupation and bond energy extrapolated in a similar way before plotting them in Figs. 8-9? If yes, why is then not the same procedure applied to the singlet-pairing amplitudes in Fig. 10?
No. The site occupations and bond energies in Fig. 9 and the singlet-pairing amplitudes in Fig. 10 are plotted without extrapolation. The extrapolation was used only in Fig. 8(a) and Fig. 8(b), because there we were interested in comparing with QMC results. We have added a clarification in the caption of Fig. 9.
11. How comes that, while the SU(N) symmetries are directly incorporated in the Tensor Network Ansatz, the conservation of total particle number is not included, and the Authors resort to tuning the average filling via a chemical potential? This might be useful to be explained to the Reader.
When implementing symmetries in PEPS, one needs to bear in mind that the tensor network itself must adher to translational invariance. Therefore symmetries cannot pick up a ‘drift’ in the average value of the symmetry labels as one moves through the system. For SU(2), for example, clearly for every multiplet with spin S, Sz varies from −S to +S in a completely symmetric way if SU(2) is preserved. Hence the average value for Sz does not drift away. The argument generalizes to SU(N), since all generators are traceless. In the case of particle- hole symmetry, with charge labels taken relative to half-filling, this also generalizes to particle number. The constraint on half-filling is important here, because otherwise one accumulates particles or holes as one increases block sizes when moving through the tensor network. Conversely, if one enforces translational invariance with charge labels defined relative to half-filling, this would always simulate half-filling, irrespective of the chemical potential chosen in the Hamiltonian. One could play trickery with the U(1) charge labels. Here, however, in order to tune the average particle filling smoothly, we simply relax charge symmetry labels to the minimum requirement, namely tracking charge parity. We have added a brief remark in this regard after discussing the symmetries with the Hamiltonian in Eq. (80). On the upside, this automatically also permits superconducting correlations.
12. In Fig. 9, orienting the μ and δ axis the same way (i.e., so that left-to-right the population grows / decreases) would probably help the reader a bit.
We thank the referee for his suggestion. We have modified Fig. 9 on page 44 accordingly, so that δ and μ axes are aligned now in the same way.
Second Referee:
1. Eq.6: is there a particular motivation to use a ket notation (|σxy >) on the physical leg, rather than just putting a physical index? After Eq (4) the physical index σyx is introduced, thus why not using the physical index instead, which is more common, rather than a ket? That notation may be confusing (also calling the tensor in Eq.6 a rank-4 tensor, rather than a rank-5 tensor may be confusing). The same comment holds also for other figures with labels on the physical leg (Eq(7), Eq(11), . . . )
We agree with the referee’s suggestion. We have fixed Eq. 5, Eq. 6, Eq. 7, Eq. 11 and Eq. 12 accordingly.
2. In Sec 3.5 it would be good to also mention the latest developments in using automatic differentiation to perform a gradient-based energy optimization, see H.-J. Liao, J.-G. Liu, L. Wang, and T. Xiang, Phys. Rev. X 9, 031041 (2019).
We agree with the referee’s suggestion. We have added a small paragraph at page 25.
3. In the study of the spinful two-band Hubbard model, it would indeed be interesting to study larger unit cell sizes (as the authors point out). But I agree that this is probably beyond the scope of the present work.
We agree with the referee that this would indeed be interesting, but leave it for the future.
4. Figure 11(a) and (c): on this scale it is difficult to see which energy is lower. Maybe it would be better to show the difference between the two (or subtracting the mean value), in order to better see which of the two states is lower in energy and by how much.
The simple update method used here gives roughly the same ground state energy for 2 × 2 and 3 × 3 clusters. As the simple update is to crude to resolve the small energy difference, it may be misleading to investigate the energy difference in more detail, hence we refrain from doing so.
5. Minor typos: - on page 40, 2nd line: a verb is missing after "109 individual contractions" - page 41, line 4: "But whem" -> "But when" - page 42, in the middle: "(a) Either" -> "(a) either" - page 46, second paragraph, after Qspace [30]: "fter" -> "After"
We thank the referee for spotting the typos and letting us know. They have been fixed.

---

## Editorial Decision

published